# Hypothesis Testing for Generalized Thurstone Models

**Anuran Makur** [*1 2]   **Japneet Singh** [*2]

## Abstract

In this work, we develop a hypothesis testing framework to determine whether pairwise comparison data is generated by an underlying *generalized Thurstone model* $\mathcal{T}_F$ for a given choice function $F$. While prior work has predominantly focused on parameter estimation and uncertainty quantification for such models, we address the fundamental problem of minimax hypothesis testing for $\mathcal{T}_F$ models. We formulate this testing problem by introducing a notion of separation distance between general pairwise comparison models and the class of $\mathcal{T}_F$ models. We then derive upper and lower bounds on the critical threshold for testing that depend on the topology of the observation graph. For the special case of complete observation graphs, this threshold scales as $\Theta((nk)^{-1/2})$, where $n$ is the number of agents and $k$ is the number of comparisons per pair. Furthermore, we propose a hypothesis test based on our separation distance, construct confidence intervals, establish time-uniform bounds on the probabilities of type I and II errors using reverse martingale techniques, and derive minimax lower bounds using information-theoretic methods. Finally, we validate our results through experiments on synthetic and real-world datasets.

## 1. Introduction

Learning rankings from data is a fundamental problem underlying numerous applications, including recommendation systems (Jannach et al., 2016), sports tournaments (Bradley & Terry, 1952; Cattelan et al., 2013), fine-tuning large language model (LLMs) (Ouyang et al., 2022), and social

choice theory (Luce, 1959; Yellott, Jr., 1977). The class of generalized Thurstone models (GTMs) (Thurstone, 1927; Nelder & Wedderburn, 1972; McCullagh & Nelder, 1989), which fall under the broader framework of random utility models, is a widely adopted framework for ranking agents, items, or choices based on given preference data. GTMs include many other models as special cases, most notably the Bradley-Terry-Luce (BTL) model (Bradley & Terry, 1952; Luce, 1959; McFadden, 1973), which has been widely studied. Given $n$ agents $[n] = \{1, \ldots, n\}$, GTMs can be construed as likelihood models for pairwise comparisons between pairs of agents. In particular, a GTM $\mathcal{T}_F$ assumes that each agent $i$ is endowed with an unknown utility parameter $w_i \in \mathbb{R}$ and the probability that agent $i$ is preferred over agent $j$ (e.g., $i$ beats $j$ in a game) is given by $F(w_i - w_j)$, where $F$ represents a known choice function which is a cumulative distribution function (CDF).

While GTMs have been utilized in many contexts, e.g., (Thurstone, 1927; Elo, 1986), they are parametric models where $n$ utility parameters characterize the model. Indeed, the assumption that pairwise comparison data is governed by a small number of parameters forms the basis of most results on GTMs (Cattelan et al., 2013; Vojnovic & Yun, 2016; Shah et al., 2016; Jadbabaie et al., 2020; 2024). However, such parametric models can sometimes be too restrictive, failing to capture intricacies in real-world applications (Davidson & Marschak, 1974; McLaughlin & Luce, 1965; Tversky, 1972). Notably, GTMs struggle to accommodate context-dependent effects, such as the home-advantage effect observed in sports tournaments (Clarke & Norman, 1995; Morley & Thomas, 2005), where teams may perform differently when playing at home versus away. Furthermore, GTMs assume transitive relationships, which may not hold in real-world datasets. To accurately capture the complex and diverse behaviors observed in real-world data, non-parametric models, e.g., (Chatterjee, 2015; Shah et al., 2016), have been studied as an alternative. This conversation raises an important question: *Given pairwise comparison data, can we determine whether it is comes from a specific GTM?* If it does, then we can rely on the vast GTM literature for learning and interpretation, and if it does not, then we can resort to using other parametric models, such as the Mallows model (Mallows, 1957), or non-parametric models (Seshadri et al., 2020; Mania et al., 2018; Bong et al., 2020).

---

[*]The author ordering is alphabetical. [1]Department of Computer Science, Purdue University, West Lafayette, IN, USA [2]Elmore Family School of Electrical and Computer Engineering, Purdue University, West Lafayette, IN, USA. Correspondence to: Anuran Makur <amakur@purdue.edu>, Japneet Singh <sing1041@purdue.edu>.

*Proceedings of the $42^{nd}$ International Conference on Machine Learning*, Vancouver, Canada. PMLR 267, 2025. Copyright 2025 by the author(s).

Despite extensive research in the area, there is no systematic answer to the above question in the literature, i.e., there is no rigorously analyzed hypothesis test to determine whether given pairwise comparison data conforms to an underlying GTM model. To address this, we study the composite hypothesis testing problem of whether data obeys a GTM $\mathcal{T}_F$ for a given choice function $F$ (which is a special kind of CDF):

$$H_0 : \mathcal{Z} \sim \mathcal{T}_F \text{ for some choice of } w \in \mathcal{W},$$
$$H_1 : \mathcal{Z} \sim \text{pairwise comparison model that is not } \mathcal{T}_F, \quad (1)$$

where $\mathcal{Z}$ denotes the pairwise comparison data, $H_0$ and $H_1$ are the null and alternative hypotheses, respectively, and $\mathcal{W}$ denotes the parameter space for the parameters $w$.

## 1.1. Main Contributions

We analyze the composite hypothesis testing problem outlined in (1). Our main contributions include the following:

1. We frame the hypothesis testing problem in a minimax sense (Section 2) by developing a rigorous notion of separation distance to the class of all GTMs that admits tractable analysis (Section 3, Theorem 3.1).
2. We derive upper and lower bounds on the critical threshold for our test (Section 3, Theorem 3.2 and Proposition 3.5). These bounds exhibit a dependence on the graph induced by the pairwise comparison data (see Table 1) and are tight for complete graphs.
3. We use the separation distance to propose a hypothesis test and establish various theoretical guarantees for our test. Specifically, we prove "time-uniform" type I and type II error probability upper bounds for our test (Section 3, Theorems 3.4 and 3.7), and also provide a minimax lower bound.
4. Additionally, we obtain auxiliary results like $\ell^2$-error bounds on parameter estimation for general pairwise comparison models (Theorem 3.3) and "time-uniform" confidence intervals under the null hypothesis (Proposition 3.8).
5. Finally, we validate our theoretical findings through synthetic and real-world experiments, proposing a data-driven approach to determine the test threshold and using the test to determine different choice functions' fit to the data (Section 4).

## 1.2. Related Literature

The class of GTMs has a rich history in the analysis of preference data. Initially proposed by Thurstone (Thurstone, 1927), these models are widely used in various fields, ranging from psychology (Thurstone, 1931), economics (Marschak, 1974), and more recent applications like aligning LLMs with human preferences (Ouyang et al., 2022).

Table 1: Bounds in this work on critical threshold $\varepsilon_c$ for various induced observation graphs, where $n$ represents the number of agents and $k$ is the number of comparisons per pair of agents.

| Graph Type | Upper Bound | Lower Bound |
|---|---|---|
| Complete graph | $O\left(\dfrac{1}{\sqrt{nk}}\right)$ | $\Omega\left(\dfrac{1}{\sqrt{nk}}\right)$ |
| $d$-Regular graph | $O\left(\dfrac{1}{\sqrt{nk}}\right)$ | $\Omega\left(\dfrac{1}{\sqrt{n^2k}}\right)$ |
| Single cycle | $O\left(\dfrac{1}{\sqrt{nk}}\right)$ | $\Omega\left(\dfrac{1}{\sqrt{n^2k}}\right)$ |
| Toroidal grid | $O\left(\dfrac{1}{\sqrt{nk}}\right)$ | $\Omega\left(\dfrac{1}{\sqrt{n^{7/4}k}}\right)$ |

Early foundational works, e.g., (Thurstone, 1927; Luce, 1959; Yellott, Jr., 1977), explored different cumulative distribution functions $F$ for modeling choice probabilities, including Gaussian (Thurstone, 1927), logistic (Bradley & Terry, 1952), and Laplace (Dawkins, 1969). These models and their extensions underlie popular rating systems, such as Elo in chess (Zermelo, 1929; Elo, 1986) and TrueSkill in video games (Herbrich et al., 2006). Several recent works have actively explored estimation techniques for Thurstone models. For instance, (Vojnovic & Yun, 2016) estimated parameters of Thurstone models when the preference data is derived from general subsets of agents (not specifically pairs), and (Shah et al., 2016) focused on parameter estimation for GTMs and the effect of graph topology on the estimation accuracy.

Furthermore, a significant portion of the literature has focused on parameter (and skill distribution) estimation in the special case of the BTL model, e.g., (Simons & Yao, 1999; Negahban et al., 2017; 2012; Chen et al., 2019; Jadbabaie et al., 2020; 2024), where two prominent algorithms are spectral ranking (Negahban et al., 2017; 2012) and maximum likelihood estimation (Zermelo, 1929; Hunter, 2004). Another related line of work is on uncertainty quantification for estimated parameters (Gao et al., 2023; Han et al., 2024; Fan et al., 2024; Liu et al., 2023). For example, (Gao et al., 2023) established the asymptotic normality of estimated parameters in the BTL model for both spectral ranking and maximum likelihood estimation, and (Han et al., 2024) generalized the asymptotic normality results to a broader class of models, such as GTMs and Mallows model. In a different vein, (Chierichetti et al., 2022) provided sharp sample complexity bounds for A/B testing needed to distinguish similar alternatives.

Despite the extensive work on parameter estimation, relatively few studies have rigorously investigated hypothesis

testing for such parametric models. Notably, (Rastogi et al., 2022) developed two-sample tests for preference data, while (Busa-Fekete et al., 2021) studied hypothesis testing for the Mallows model. (Seshadri & Ugander, 2019) studied lower bounds for testing the independence of irrelevant alternatives (IIA) assumption (i.e., BTL and Plackett-Luce models (Luce, 1959; Plackett, 1975)), and (Makur & Singh, 2023; 2024) developed hypothesis tests for BTL models based on spectral methods. In contrast to these works, we develop hypothesis testing for GTMs using a maximum likelihood framework, complementing the work in (Makur & Singh, 2023; 2024).

## 2. Formal Model and Setup

We begin by introducing a general pairwise comparison model that provides a flexible framework encompassing a broad range of established probabilistic models, including the BTL model (Bradley & Terry, 1952; Luce, 1959; McFadden, 1973), the Thurstone model (Thurstone, 1927), and non-parametric models (Chatterjee, 2015; Shah et al., 2016). In this framework, we consider $n \in \mathbb{N}\backslash\{1\}$ agents (or items or choices) $[n]$ engaged in pairwise comparisons. For agents $i, j \in [n]$ with $i \neq j$, let $p_{ij} \in (0, 1)$ denote the probability that $i$ is preferred over $j$ in an "$i$ vs. $j$" pairwise comparison. This model inherently captures the asymmetric nature of pairwise comparisons, as the outcome of an "$i$ vs. $j$" comparison may differ from that of a "$j$ vs. $i$" comparison. This reflects real-world phenomena like home-advantage that are commonly observed in sports (Clarke & Norman, 1995; Morley & Thomas, 2005).

To model the fact that not all pairwise comparisons may be observed, we assume that we are given an induced observation graph $\mathcal{G} = ([n], \mathcal{E})$, where an edge $(i, j) \in \mathcal{E}$ (with $i \neq j$) exists if and only if comparisons of the form "$i$ vs. $j$" are observed. Let $E \in \{0, 1\}^{n \times n}$ be the adjacency matrix of $\mathcal{G}$, with $E_{ij} = 1$ if $(i, j) \in \mathcal{E}$ and $0$ otherwise. Furthermore, we assume that the edge set $\mathcal{E}$ is symmetric (i.e., $\mathcal{G}$ is undirected), implying that if "$i$ vs. $j$" comparisons are observed, then "$j$ vs. $i$" comparisons are observed as well. Additionally, we assume that $\mathcal{G}$ is connected and is fixed a priori (see Proposition 2.4), independent of the outcomes of the observed pairwise comparisons.

We also let $D \in \mathbb{R}^{n \times n}$ be the diagonal degree matrix with $D_{ii} = \sum_{j=1}^{n} E_{ij}$ for $i \in [n]$, and $L \triangleq D - E$ be the graph Laplacian matrix. $L$ can be expressed as $L = X^{\mathrm{T}}X$, where $X \in \mathbb{R}^{(|\mathcal{E}|/2) \times n}$ is the matrix formed by collecting row vectors $x_{ij} = e_i - e_j$ for $(i, j) \in \mathcal{E}$ and $j > i$, with $e_i$ being the $i$th standard basis vector in $\mathbb{R}^n$. For the Laplacian $L$, we define the semi-norm with respect to $L$ as $\|x\|_L = \sqrt{x^{\mathrm{T}}Lx}$ for all $x \in \mathbb{R}^n$ (which is a semi-norm since $L$ has a zero eigenvalue).

### 2.1. Comparison Models

**Pairwise comparison model:** Given the above preamble, we now formally present general pairwise comparison models (which encompass BTL models (Bradley & Terry, 1952; Luce, 1959; McFadden, 1973), Thurstone models (Thurstone, 1927), non-parametric models (Chatterjee, 2015; Shah et al., 2016), etc., as mentioned above).

**Definition 2.1** (Pairwise Comparison Model)**.** Given an observation graph $\mathcal{G}$ over the agents $[n]$, we refer to the collection of probability parameters $\{p_{ij} : (i, j) \in \mathcal{E}\}$ as a *pairwise comparison model*.

Furthermore, we can represent a pairwise comparison model by a *pairwise comparison matrix* $P \in [0, 1]^{n \times n}$ with

$$P_{ij} \triangleq \begin{cases} p_{ij}, & (i, j) \in \mathcal{E}, \\ 0, & \text{otherwise.} \end{cases} \tag{2}$$

We remark that our ensuing analysis can be easily specialized to a symmetric setting where "$i$ vs. $j$" and "$j$ vs. $i$" comparisons are equivalent. In this case, $E$ is automatically symmetric as assumed. On the other hand, the symmetry assumption on $E$ is needed in asymmetric settings because GTMs inherently treat "$i$ vs. $j$" and "$j$ vs. $i$" comparisons as equivalent, which is not true in general models.

**GTM model:** Next, we describe a GTM for a choice function $F : \mathbb{R} \to [0, 1]$, which is a special kind of CDF (to be explained in the sequel).

**Definition 2.2** (Generalized Thurstone Model)**.** Given an observation graph $\mathcal{G}$, a pairwise comparison model is said to be a *generalized Thurstone model* (GTM) $\mathcal{T}_F$ with choice function $F : \mathbb{R} \to [0, 1]$ if there exists a weight (or utility) vector $w \in \mathcal{W}$ such that:

$$\forall (i, j) \in \mathcal{E}, \ p_{ij} = F(w_i - w_j),$$

where $\mathcal{W} \subseteq \mathbb{R}^n$ is a specified convex parameter space (usually $\mathbb{R}^n$ or a compact hypercube in $\mathbb{R}^n$).

The GTM (Nelder & Wedderburn, 1972; McCullagh & Nelder, 1989) posits that every agent $i$ has a latent utility $w_i$, and uncertainty in the comparison process is modeled by independent and identically distributed (i.i.d.) noise random variables $X_1, \ldots, X_n$ with absolutely continuous CDF $G : \mathbb{R} \to [0, 1]$. The discriminant variables $(w_1 + X_1, \ldots, w_n + X_n)$ formed by combining utilities with the noise random variables are then compared to determine the outcomes of pairwise comparisons. Hence, the probability of preferring agent $i$ over $j$ is given by

$$\mathbb{P}(i \text{ preferred over } j) = \mathbb{P}(w_i + X_i > w_j + X_j)$$
$$= \int_{-\infty}^{\infty} G(y + w_i - w_j)G'(y) \, \mathrm{d}y = F(w_i - w_j). \tag{3}$$

As noted earlier, GTMs also encompass a wide range of known parametric models as special cases, e.g., Thurstone models with $F(t) = \int_{-\infty}^{t} \frac{1}{\sqrt{2\pi}} e^{-x^2/2} \, dx$ (complementary Gaussian CDF) (Thurstone, 1927), BTL models with $F(t) = 1/(1 + e^{-t})$ (sigmoid function, which stems from Gumbel CDFs) (Bradley & Terry, 1952; Luce, 1959; Yellott, Jr., 1977), Dawkins models (Dawkins, 1969), etc. We can also define a pairwise probability matrix $\mathsf{F}(w) \in [0,1]^{n \times n}$ for a GTM $\mathcal{T}_F$ with weight vector $w$ via

$$(\mathsf{F}(w))_{ij} \triangleq \begin{cases} F(w_i - w_j), & (i,j) \in \mathcal{E}, \\ 0, & \text{otherwise.} \end{cases} \quad (4)$$

We next describe the data generation process for GTMs and general pairwise comparison models alike. For any pair $(i,j) \in \mathcal{E}$, define the outcome of the $m$th "$i$ vs. $j$" pairwise comparison between them as the Bernoulli random variable

$$Z_{ij}^m \triangleq \begin{cases} 1, & i \text{ preferred over } j \text{ (with probability } p_{ij}), \\ 0, & j \text{ preferred over } i \text{ (with probability } 1 - p_{ij}), \end{cases} \quad (5)$$

for $m \in [k_{ij}]$, where $k_{ij}$ denotes the number of observed "$i$ vs. $j$" comparisons. The given pairwise comparison data is then a collection of these *independent* Bernoulli variables $\mathcal{Z} \triangleq \{Z_{ij}^m : (i,j) \in \mathcal{E}, \ m \in [k_{i,j}]\}$. For convenience, we also let $Z_{ij} \triangleq \sum_{m=1}^{k_{ij}} Z_{ij}^m$ and $\hat{p}_{ij} \triangleq Z_{ij}/k_{ij}$.

**Parameter estimation for GTM:** To present our testing formulation in the sequel, we explain how the parameters of a $\mathcal{T}_F$ model are estimated given pairwise comparison data $\mathcal{Z}$ (Shah et al., 2016). First, we define the weighted negative log-likelihood function $l : \mathcal{W} \times [0,1]^{|\mathcal{E}|} \to \mathbb{R}_+ \cup \{+\infty\}$ as

$$l(w; \{\hat{p}_{ij} : (i,j) \in \mathcal{E}\}) \triangleq - \sum_{(i,j) \in \mathcal{E}} \hat{p}_{ij} \log(F(w_i - w_j))$$
$$+ (1 - \hat{p}_{ij}) \log(1 - F(w_i - w_j)). \quad (6)$$

Note that this function represents a weighted variant of the typical log-likelihood function used in parameter estimation (Shah et al., 2016; Vojnovic & Yun, 2016). The weights of the $\mathcal{T}_F$ model are estimated by minimizing $l$:

$$\hat{w} \triangleq \underset{w \in \mathcal{W}_b}{\arg\min} \, l(w; \{\hat{p}_{ij} : (i,j) \in \mathcal{E}\}), \quad (7)$$

where the constraint set $\mathcal{W}_b \triangleq \{w \in \mathcal{W} : \|w\|_\infty \leq b, \ w^{\mathrm{T}} \mathbf{1} = 0\}$ for some (universal) constant $b$, $\mathbf{1} \in \mathbb{R}^n$ denotes an all-ones vector, and the constraint $w^{\mathrm{T}} \mathbf{1} = 0$ allows for identifiability of the weights.

## 2.2. Assumptions on Comparison Models

To facilitate the analysis of the hypothesis testing problem in (1), we introduce a simplifying assumption on the class of general pairwise comparison models. We assume that the pairwise probabilities $p_{ij}$ are bounded away from 0 and 1.

**Assumption 2.3** (Dynamic Range). There exists a constant $\delta > 0$ such that for any pairwise comparison model under consideration, $p_{ij} \in [\delta, 1 - \delta]$ for all $(i,j) \in \mathcal{E}$.

Note that under the null hypothesis, the Assumption 2.3 is satisfied by all $\mathcal{T}_F$ models with weights bounded by $F^{-1}(1 - \delta)/2$. Subsequently, we assume that the constant $b$ satisfies $b \geq F^{-1}(1 - \delta)/2$. For any given pairwise comparison model $\{p_{ij} : (i,j) \in \mathcal{E}\}$, define $w^* \in \mathcal{W}_b$ be the weights of a $\mathcal{T}_F$ model that best approximates this pairwise comparison model in the maximum likelihood sense:

$$w^* \triangleq \underset{w \in \mathcal{W}_b}{\arg\min} \, l(w; \{p_{ij} : (i,j) \in \mathcal{E}\}). \quad (8)$$

Finally, we also assume in the sequel that the given choice function $F$ exhibits *strong log-concavity* and has a bounded derivative on $[-2b, 2b]$, i.e., there exist constants $\alpha, \beta > 0$ such that:

$$\forall x \in [-2b, 2b], \quad -\frac{\mathrm{d}^2}{\mathrm{d}x^2} \log(F(x)) \geq \alpha \ \text{ and } \ F'(x) \leq \beta. \quad (9)$$

Several popular GTMs, including the BTL and Thurstone (Case V) models, satisfy both the above assumptions. The following proposition highlights that $w^*$ always exists and is unique for a strongly log-concave function $F$ on $\mathcal{W}_b$.

**Proposition 2.4** (Existence and Uniqueness of Maximum Likelihood). *Suppose the observation graph $\mathcal{G}$ is connected, the choice function $F : \mathbb{R} \to [0,1]$ satisfies (9), and Assumption 2.3 holds. Then, there exists a unique optimal solution $w^* \in \mathcal{W}_b$ satisfying (8).*

The proof is provided in Appendix A.2. It follows from Proposition 2.4 and Gibbs' inequality that when the pairwise comparison model is indeed a $\mathcal{T}_F$ model with weight vector $w$, then we have $w^* = w$.

## 2.3. Minimax Formulation

Given any fixed graph $\mathcal{G}$, separation level $\epsilon > 0$, choice function $F$, and constants $\delta > 0$ and $b \geq F^{-1}(1 - \delta)/2$, define the sets $\mathcal{M}_0$ and $\mathcal{M}_1(\epsilon)$ of $\mathcal{T}_F$ and pairwise comparison models:

$$\mathcal{M}_0 \triangleq \{P : \text{Assumption 2.3 holds and}$$
$$\exists \, w \in \mathcal{W}_b \text{ such that } P = \mathsf{F}(w)\}, \quad (10)$$

$$\mathcal{M}_1(\epsilon) \triangleq \left\{ P : \text{Assumption 2.3 holds and} \right.$$

$$\left. \inf_{w \in \mathcal{W}_b} \frac{1}{n} \|P - \mathsf{F}(w)\|_{\mathrm{F}} \geq \epsilon \right\}, \quad (11)$$

where $\|\cdot\|_{\mathrm{F}}$ denotes Frobenius norm. Now, we formalize the hypothesis testing problem in (1) as:

$$\begin{aligned} H_0 : \ & \mathcal{Z} \sim P \in \mathcal{M}_0, \\ H_1 : \ & \mathcal{Z} \sim P \in \mathcal{M}_1(\epsilon). \end{aligned} \quad (12)$$

We will discuss the separation distance $\inf_{w \in \mathcal{W}_b} \|P - \mathsf{F}(w)\|_{\mathrm{F}}$ later. For now, note that we only test on the set of observed comparisons $\mathcal{E}$ as it is not possible to determine whether the comparisons on $\mathcal{E}^c$ would conform to a $\mathcal{T}_F$ model or some other pairwise comparison model. Next, for any fixed graph $\mathcal{G}$, choice function $F$, and constants $\epsilon, \delta, b$, we define the *minimax risk* as

$$\mathcal{R}(\mathcal{G}, \epsilon) \triangleq \inf_{\phi} \left\{ \underbrace{\sup_{P \in \mathcal{M}_0} \mathbb{P}_{H_0}(\phi(\mathcal{Z}) = 1)}_{\mathcal{Z} \sim P \text{ under } H_0} + \right.$$

$$\left. \underbrace{\sup_{P \in \mathcal{M}_1(\epsilon)} \mathbb{P}_{H_1}(\phi(\mathcal{Z}) = 0)}_{\mathcal{Z} \sim P \text{ under } H_1} \right\}, \quad (13)$$

where the infimum is taken over all randomized decision rules $\phi(\mathcal{Z}) \in \{0, 1\}$ (with 0 corresponding to $H_0$ and 1 to $H_1$), and $\mathbb{P}_{H_0}$ and $\mathbb{P}_{H_1}$ denote the probability measures under hypotheses $H_0$ and $H_1$, respectively. Intuitively, this risk minimizes the sum of the worst-case type I and type II error probabilities. Finally, we define the *critical threshold* of the hypothesis testing problem in (12) as the smallest value of $\epsilon$ for which the minimax risk is bounded by $\frac{1}{2}$ (cf. (Rastogi et al., 2022)):

$$\varepsilon_{\mathsf{c}} \triangleq \inf \left\{ \epsilon > 0 : \mathcal{R}(\mathcal{G}, \epsilon) \le \frac{1}{2} \right\}. \quad (14)$$

Note that the constant $\frac{1}{2}$ here is arbitrary and can be replaced by any constant in $(0, 1)$.

# 3. Main Results

In this section, we present the main results of the paper. We first show that our notion of separation distance can be simplified for analysis, then proceed to bound the critical threshold and minimax risk, and finally, establish type I and II error probability bounds in the sequential setting.

## 3.1. Separation Distance and Test Statistic

Recall that to formalize (12), we defined the *separation distance* of a pairwise comparison model $P$ to the class of $\mathcal{T}_F$ models as $\inf_{w \in \mathcal{W}_b} \|P - \mathsf{F}(w)\|_{\mathrm{F}}$ (for fixed $F$). To make this separation distance more amenable to theoretical analysis, we approximate it in the next theorem with the simpler quantity $\|P - \mathsf{F}(w^*)\|_{\mathrm{F}}$, where $w^*$ is given in (8).

**Theorem 3.1** (Separation Distance to $\mathcal{T}_F$ Models). *Let $P$ be a pairwise comparison matrix satisfying Assumption 2.3. Then, there exists a constant $c_1 > 0$ (that does not depend on $n$) such that the separation distance between $P$ and the class of $\mathcal{T}_F$ models satisfies*

$$c_1 \|P - \mathsf{F}(w^*)\|_{\mathrm{F}} \le \inf_{w \in \mathcal{W}_b} \|P - \mathsf{F}(w)\|_{\mathrm{F}} \le \|P - \mathsf{F}(w^*)\|_{\mathrm{F}},$$

*where $w^*$ is given by (8).*

The proof is provided in Appendix A.3. The upper bound is immediate, and the lower bound utilizes the information-theoretic bounds between $f$-divergences.

**Test statistic:** We now introduce our test statistic based on the approximation derived in Theorem 3.1. First, we partition the observed comparison data $\mathcal{Z}$ into two (roughly) equal parts $\mathcal{Z}_1 = \{Z_{ij}^m : (i, j) \in \mathcal{E}, m \in [\lfloor k_{ij}/2 \rfloor]\}$ and $\mathcal{Z}_2 = \mathcal{Z} \setminus \mathcal{Z}_1$. The first half of the dataset $\mathcal{Z}_1$ is used to estimate the parameters $\hat{w}$ as shown in (7). Then, we use $\mathcal{Z}_2$ to calculate the *test statistic $T$* via

$$T \triangleq \sum_{(i,j) \in \mathcal{E}} \left( \frac{Z_{ij}(Z_{ij} - 1)}{k'_{ij}(k'_{ij} - 1)} + F(\hat{w}_i - \hat{w}_j)^2 \right.$$

$$\left. - 2F(\hat{w}_i - \hat{w}_j) \frac{Z_{ij}}{k'_{ij}} \right) \mathbb{1}_{k'_{ij} > 1}, \quad (15)$$

where $k'_{ij} = k_{ij} - \lfloor k_{ij}/2 \rfloor$, $Z_{ij} = \sum_{m > \lfloor k_{ij}/2 \rfloor} Z_{ij}^m$ is computed as before but using only the samples in $\mathcal{Z}_2$, and $\mathbb{1}_{\mathcal{A}}$ denotes the indicator function on $\mathcal{A}$. For an intuitive explanation of the definition of this statistic, consider the quantity

$$T' = \sum_{(i,j) \in \mathcal{E}} \frac{Z_{ij}(Z_{ij} - 1)}{k'_{ij}(k'_{ij} - 1)} + F(w_i^* - w_j^*)^2 - 2F(w_i^* - w_j^*) \frac{Z_{ij}}{k'_{ij}}, \quad (16)$$

obtained by substituting $w^*$ in place $\hat{w}$ in (15). Then, the expected value of $T'$ is $\|P - F(w^*)\|_{\mathrm{F}}^2$. This is because the expected value of the first term is $p_{ij}^2$, and the last term is $-2F(w_i^* - w_j^*) p_{ij}$. Hence, $T$ is constructed by plugging in $\hat{w}$ in place of $w^*$ in an unbiased estimator of $\|P - \mathsf{F}(w^*)\|_{\mathrm{F}}^2$. Our proposed *hypothesis test thresholds $T$ to determine the unknown hypothesis*; $H_1$ is selected if $T$ exceeds a certain threshold. There is some resemblance between the test statistic $T$ and those in (Makur & Singh, 2023; 2024; Rastogi et al., 2022) since all these statistics are "estimators" for squared Frobenius distances of pairwise comparison models. However, the techniques used to analyze them are very different. For example, our theoretical analysis crucially relies on sample splitting, while the other two do not. We will discuss analytical expressions for the threshold next and a data-driven approach to determining the threshold in Section 4.

## 3.2. Upper Bound on Critical Threshold

In this section, we make the simplifying assumption that $k_{ij} = 2k$ (with $k \in \mathbb{N}$) for all $(i, j) \in \mathcal{E}$. The ensuing theorem, proved in Appendix B, establishes an upper bound on the critical threshold of the hypothesis testing problem defined in (12).

**Theorem 3.2** (Upper Bound on Critical Threshold). *Consider the hypothesis testing problem in (12), and assume that Assumption 2.3 holds and $k \ge 2$. Then, there exists a*

constant $c_2 > 0$ *such that the critical threshold defined in* *(14) is upper bounded by*

$$\varepsilon_{\mathsf{c}}^2 \leq \frac{c_2}{nk}.$$

In our analysis, we select $H_1$ if $T > \gamma \frac{n}{k}$ and $H_0$ otherwise, where $\gamma$ is an appropriate constant independent of $n, k$ (see (37)). The analysis relies on establishing non-trivial error bounds (in $\| \cdot \|_L$ semi-norm) for parameter estimation of $\mathcal{T}_F$ models when the data is generated by a general pairwise comparison model, which is not necessarily a GTM (i.e., deriving error bounds under a potential model mismatch). The error bounds allow us to prove bounds on the mean and variance of the test statistic $T$ under both hypotheses $H_0$ and $H_1$. Then, using Chebyshev's inequality, we can bound the probabilities of error of our test under each of the hypotheses, which induces an upper bound on the critical threshold.

We also note that in the special case where $\mathcal{T}_F$ is a BTL model, our upper bound on $\varepsilon_{\mathsf{c}}$ recovers the bound in (Makur & Singh, 2023; 2024) for complete graphs. But our likelihood-based proof is quite different to the spectral ideas in (Makur & Singh, 2023; 2024). Finally, we present the key error bounds for parameter estimation when data is generated by a general pairwise comparison model needed to prove Theorem 3.2.

**Theorem 3.3** (Error Bounds for Parameter Estimation)**.** *Consider any pairwise comparison model satisfying Assumption 2.3 with $w^*$ given by (8) and $\hat{w}$ constructed according to (7) from data generated by the model. Then, for some constant $c_3 > 0$, the following tail bound holds on the estimation error of $w^*$:*

$$\forall t \geq 1, \ \ \mathbb{P}\left( \|\hat{w} - w^*\|_L^2 \geq \frac{c_3 n \beta^2}{\alpha^2 k F(-2b)^2} \, t \right) \leq e^{-t},$$

*where $\alpha$ is defined in (9). Moreover, for any $p \geq 1$, there exists a $p$-dependent constant $c(p) > 0$ such that the expected $p$th moment of the error is bounded by*

$$\mathbb{E}[\|\hat{w} - w^*\|_L^p] \leq \left( \frac{c(p) n \beta^2}{\alpha^2 k F(-2b)^2} \right)^{\frac{p}{2}}.$$

The proof is provided in Appendix A.4. In the special case where the pairwise comparison model is a GTM, our bounds recover the bounds derived in (Shah et al., 2016, Theorem 3) up to constants. However, our result is much more general because it holds for any pairwise comparison model; this requires a careful formulation and development of the proof techniques.

We remark that these error bounds can be readily converted into $\ell^2$-error bounds using the relation $\|\hat{w} - w^*\|_L^2 \geq$

$\lambda_2(L)\|\hat{w}-w^*\|_2^2$, where $\lambda_2(L)$ is the second smallest eigenvalue of the Laplacian $L$. The connectedness of the graph ensures that $\lambda_2(L) > 0$, and the value of $\lambda_2(L)$ is known for various classes of graphs (cf. (Chung, 1997)).

Moreover, it is worth emphasizing that our error bounds in $\| \cdot \|_L$ semi-norm (in Theorem 3.3) remain valid even when the observation graph is disconnected. Furthermore, since the proof of Theorem 3.2 relies only on these bounds, the results of Theorem 3.3 hold regardless of graph connectivity. When the graph $\mathcal{G}$ is disconnected, solutions $\hat{w}$ and $w^*$ of (7) and (8) may not be unique, but the error $\|\hat{w} - w^*\|_L$ is well-defined as the non-unique component lies in the null space of $L$. Thus, our upper bounds on critical thresholds still hold. This ensures that our testing framework for $\mathcal{T}_F$ models remains applicable even in settings where parameter estimation is infeasible due to a disconnected observation graph, highlighting a *fundamental distinction between testing and parameter estimation* of $\mathcal{T}_F$ models. Moreover, we also note that our results lead to bounds on critical thresholds even when the separation distance is defined in other norms; see Appendix G for more details.

### 3.3. Information-Theoretic Lower Bounds

We now establish information-theoretic lower bounds on the minimax risk and critical threshold for the hypothesis testing problem in (12). For simplicity and analytical tractability, assume that $k_{ij} = k \in \mathbb{N}$ for all $(i,j) \in \mathcal{E}$, and assume that the observation graph $\mathcal{G}$ is *super-Eulerian* (Catlin, 1992), i.e., it has an Eulerian spanning sub-graph $\tilde{\mathcal{G}} = ([n], \tilde{\mathcal{E}})$ so that every vertex of $\tilde{\mathcal{G}}$ has even degree. Then, $\tilde{\mathcal{G}}$ has a *cycle decomposition* $\mathcal{C}$ by Veblen's theorem (Biggs et al., 1976; Seshadri & Ugander, 2019), where $\mathcal{C}$ is a collection of simple cycles $\sigma$ that partitions the undirected edges of $\tilde{\mathcal{G}}$. The ensuing theorem, proved in Appendix C, presents our minimax risk lower bound.

**Theorem 3.4** (Minimax Lower Bound)**.** *Consider the hypothesis testing problem in (12) and assume that the observation graph $\mathcal{G}$ is super-Eulerian with spanning Eulerian sub-graph $\tilde{\mathcal{G}}$. Then, there exists a constant $c_4 > 0$ such that for any $\epsilon > 0$, the minimax risk in (13) is lower bounded by*

$$\mathcal{R}(\mathcal{G}, \epsilon) \geq 1 - \frac{1}{2}\sqrt{\exp\left( \frac{c_4 k^2 n^4 \epsilon^4}{|\tilde{\mathcal{E}}|^2} \sum_{\sigma \in \mathcal{C}} |\sigma|^2 \right) - 1},$$

*where $|\sigma|$ denotes the length of a cycle $\sigma \in \mathcal{C}$, and $\mathcal{C}$ is the cycle decomposition of $\tilde{\mathcal{G}}$.*

Our argument utilizes the *Ingster-Suslina method* (Ingster, 1994; Ingster & Suslina, 2003), which is similar to *Le Cam's method*, but provides a lower bound by considering a cleverly chosen point and a mixture on the parameter space instead of just two points. Our specific construction is inspired by the technique introduced in (Seshadri & Ugander,

2019), which establishes a lower bound for testing the IIA assumption (i.e., BTL models) for Eulerian graph structures. We extend their approach in three ways. First, we generalize their method to accommodate any GTM rather than just the BTL model. Second, we use a different technique based on Theorem 3.1 to lower bound separation distance from the class of $\mathcal{T}_F$ models. Moreover, our work quantifies separation using Frobenius norm instead of sums of total variation (TV) distances. Third, our argument holds for a broader class of graphs, namely, super-Eulerian graphs. Note that the question of algorithmically constructing Eulerian subgraphs of graphs has been widely studied (Haghparast & Kiani, 2019).

The following proposition simplifies Theorem 3.4 to obtain lower bounds on the critical threshold for several classes of graphs.

**Proposition 3.5** (Lower Bounds on Critical Threshold). *Under the assumptions of Theorem 3.4, the following lower bounds hold for the critical threshold defined in (14):*

1. *If $\mathcal{G}$ is a complete graph with odd $n$ vertices, then $\varepsilon_c^2 = \Omega(1/nk)$.*

2. *If $\mathcal{G}$ is a $d$-regular graph with constant $d \geq 2$, then $\varepsilon_c^2 = \Omega(1/n^2 k)$.*

3. *If $\mathcal{G}$ is a single cycle graph with $n$ vertices, then $\varepsilon_c^2 = \Omega(1/n^2 k)$.*

4. *If $\mathcal{G}$ is a two-dimensional $\sqrt{n} \times \sqrt{n}$ toroidal grid on $n$ vertices formed by the Cartesian product of two cycles of length $\sqrt{n}$, then $\varepsilon_c^2 = \Omega(1/n^{7/4} k)$.*

The proof of Proposition 3.5 is provided in Appendix C.1. It involves calculating the number of simple cycles and the individual cycle lengths in the cycle decompositions $\mathcal{C}$. The lower bounds on $\varepsilon_c$ are then obtained from Theorem 3.4. We remark that our minimax upper and lower bounds on $\varepsilon_c$ match for the complete graph case, demonstrating the minimax optimality of the threshold's scaling (up to constant factors). Moreover, they also match with respect to $k$ for other classes of graphs as well. It is worth mentioning that in the special case of BTL models with single cycle graphs, our lower bound on $\varepsilon_c$ improves the high-level scaling behavior in (Seshadri & Ugander, 2019) from $\Omega(1/\sqrt{n^3 k})$ to $\Omega(1/\sqrt{n^2 k})$ (when $\varepsilon_c$ is quantified in terms of Frobenius norm). Lastly, we remark that for single cycle graphs, the gap between the upper and lower bounds in terms of $n$ intuitively holds because our lower bounds become larger when there are more cycles in $\mathcal{C}$, which is only 1 in this case.

### 3.4. Upper Bounds on Probabilities of Type I and II Errors

To complement the minimax risk lower bound in Theorem 3.4, we establish upper bounds on the extremal type I and II error probabilities. We will do this in the *sequential* setting, where data is observed incrementally—a common practical scenario which subsumes the standard fixed sample-size setting, cf. (Manole & Ramdas, 2023; Howard et al., 2020). In the sequential testing framework, at each time step, we observe a single "$i$ vs. $j$" comparison for every $(i, j) \in \mathcal{E}$. (The subsequent analysis can be extended to a general setting where we observe only one comparison for some pair $(i, j) \in \mathcal{E}$ or even a variable number of comparisons at every time step.) At time $k_1 + k$ with $k_1, k \in \mathbb{N}$, we define $T^{k_1, k}$ to be the value of the test statistic $T$ in (15), where comparisons from $k_1$ time-steps have been used to build the dataset $\mathcal{Z}_1$ to estimate parameters using $\hat{w}$, and $k$ time-steps have been used to build the dataset $\mathcal{Z}_2$ to calculate the statistic $T$. Note that $\mathcal{Z}_1$ and $\mathcal{Z}_2$ no longer need to be similar in size. Then, we can decide based on thresholding $T^{k_1, k}$ (see Theorem 3.7) whether to collect more data or stop and reject $H_0$ while controlling the probabilities of error. If the testing process ends without rejecting $H_0$, then we can accept $H_0$. A key observation underlying our analysis is the following *reverse martingale* property (see, e.g., (Howard et al., 2020; Manole & Ramdas, 2023)).

**Proposition 3.6** (Reverse Martingale). *Fix any $k_1 \in \mathbb{N}$, and let $\mathcal{F}_k = \bigotimes_{(i,j)\in\mathcal{E}} \sigma(\sum_{m=k_1+1}^{k_1+k} Z_{ij}^m, Z_{ij}^{k_1+k+1}, Z_{ij}^{k_1+k+2}, \dots)$ be a non-increasing sequence of $\sigma$-algebras, where $\bigotimes$ denotes the product $\sigma$-algebra. Then, the sequence of statistics $\{T^{k_1,k} : k \geq 2\}$ is a reverse martingale with respect to the reverse filtration $\{\mathcal{F}_k : k \geq 2\}$, i.e., for $k \geq 2$, $T^{k_1,k}$ is $\mathcal{F}_k$-measurable and $\mathbb{E}\left[T^{k_1,k}|\mathcal{F}_{k+1}\right] = T^{k_1,k+1}$.*

The proof is presented in Appendix D.1. This observation allows us to develop *time-uniform bounds* in terms of $k$ on the probabilities of type I and II errors, i.e., they hold for all $k$ larger than a constant. The next theorem, proved in Appendix D.3, presents our bounds on the probabilities of type I and II errors.

**Theorem 3.7** (Type I and Type II Error Probability Bounds). *Under the sequential setting discussed above, the following bounds hold on the extremal type I and type II error probabilities. There exist constants $c_5, c_6, c_7, c_8$ such that for all $t \geq 1$, $\nu \in (0, 1/e)$, $k_1 \in \mathbb{N}$, and $\epsilon \geq c_5\sqrt{t}/\sqrt{nk_1}$, we have*

$$\sup_{P \in \mathcal{M}_0} \mathbb{P}_{H_0}\left(\exists k \geq 2, T^{k_1,k} \geq c_6 t \frac{n}{k_1} + \frac{c_7 |\mathcal{E}|^{\frac{1}{2}} \ell_{k,\nu}}{k}\right.$$

$$\left. + c_8 \sqrt{\frac{nt\ell_{k,\nu}}{k_1 k}}\right) \leq \nu + e^{-t},$$

$$\sup_{P \in \mathcal{M}_1(\epsilon)} \mathbb{P}_{H_1}\left(\exists k \geq 2, T^{k_1,k} - \left(D - c_5 t^{\frac{1}{2}} n^{\frac{1}{2}}/k_1^{\frac{1}{2}}\right)^2 \leq \right.$$

$$\left. - \frac{c_7 |\mathcal{E}|^{\frac{1}{2}} \ell_{k,\nu}}{k} - \left(4D + c_8 t^{\frac{1}{2}} n^{\frac{1}{2}}/k_1^{\frac{1}{2}}\right)\sqrt{\frac{\ell_{k,\nu}}{k}}\right) \leq \nu + e^{-t},$$

*where* $D \triangleq \|P - \mathsf{F}(w^*)\|_{\mathrm{F}}$ *and* $\ell_{k,\nu} \triangleq \log(3.5 \log_2(k)^2/\nu)$.

We now make several remarks. Firstly, our error probability bounds encode the scalings of the thresholds to accept or reject $H_0$ (see (37)). Secondly, our bounds hold regardless of how the decision-maker assigns data collected at different time-steps to $\mathcal{Z}_1$ and $\mathcal{Z}_2$. Moreover, they provide insights on how to split the data based on the topology of the observation graph, e.g., the bounds suggest an equal split of the data for complete graphs, whereas for a single cycle, achieving better type I error control requires a larger value of $k_1$. To illustrate this and help parse Theorem 3.7, we present corollaries of Theorem 3.7 for the complete and single cycle graph cases in Appendix E.

Thirdly, our bounds clearly hold in the non-sequential fixed sample-size setting, as we can just fix a particular value of $k$. Hence, adding the two extremal probabilities of error yields upper bounds on the minimax risk. Notably, the proof of Theorem 3.7 requires us to develop a time-uniform version of the well-known Hanson-Wright inequality (Rudelson & Vershynin, 2013) specialized for our setting (see Lemma D.2 in Appendix D.2). Additionally, as an intermediate step in the proof, we also obtain time-uniform *confidence intervals* under the null hypothesis $H_0$, as demonstrated in the following proposition.

**Proposition 3.8** (Confidence Interval for $T^{k_1,k}$)**.** *Suppose* $\hat{w}$ *is estimated as in (7) from the comparisons over* $k_1$ *time-steps. Then, there exists a constant* $c_7 > 0$ *such that for all* $\nu \in (0, 1/e)$ *and* $k_1 \in \mathbb{N}$,

$$\mathbb{P}_{H_0}\bigg(\exists k \geq 2, T^{k_1,k} \geq \|\mathsf{F}(\hat{w}) - \mathsf{F}(w^*)\|_{\mathrm{F}}^2 + c_7 \frac{\sqrt{|\mathcal{E}|\ell_{k,\nu}}}{k}$$
$$+ 4\|\mathsf{F}(\hat{w}) - \mathsf{F}(w^*)\|_{\mathrm{F}}\sqrt{\frac{\ell_{k,\nu}}{k}}\bigg) \leq \nu.$$

Proposition 3.8 is established in Appendix D.2. We remark that the distribution of $\|\mathsf{F}(\hat{w}) - \mathsf{F}(w^*)\|_{\mathrm{F}}$ above can be approximated either by leveraging the asymptotic normality of $\hat{w} - w^*$ (Gao et al., 2023; Han et al., 2024), or by utilizing bootstrapping techniques; this gives $(1 - 2\nu)$ time-uniform confidence intervals. Additionally, the constant $c_7$ here is also the constant in our specialized version of Hanson-Wright inequality (noted above) and can be approximated via simulations for our setting. An empirical investigation into estimating the constant $c_7$ and the subsequent confidence intervals can be found in Appendix F.3.

## 4. Experiments

In this section, we develop a data-driven approach to select the threshold for our test $T$ and conduct simulations to validate our theoretical results on synthetic and real-world datasets. We include several additional experiments in Appendix F to validate our threshold estimation procedure, compare our performance with the spectral method of (Makur & Singh, 2023; 2024) for the BTL model, and provide additional experiments on real-world datasets.

**Estimating the threshold:** Given a pairwise comparison dataset $\mathcal{Z} \triangleq \{Z_{ij}^m : (i,j) \in \mathcal{E}, m \in [k_{ij}]\}$, we employ an empirical-quantile-based approach to determine the critical threshold for our hypothesis testing problem. We generate multiple $\mathcal{T}_F$ models with random skill scores $w \in \mathbb{R}^n$ drawn independently and uniformly in $[-b, b]$ and translated to satisfy $w^{\mathrm{T}}\mathbf{1} = 0$, and simulate $k_{ij}$ "$i$ vs. $j$" comparisons by sampling binomial random variables $\{\tilde{Z}_{ij} \sim \mathrm{Bin}(k_{ij}, F(w_i - w_j))\}_{(i,j)\in\mathcal{E}}$. We then compute the test statistic $T$ for each simulated dataset, repeating the process a sufficient number of times to build a distribution of test statistics. Finally, we extract the 95th percentile value from this distribution as our empirical threshold.

In our first experiment, we investigate the behavior of the thresholds for $T$ based on this empirical-quantile-based approach for various values of $n$ and $k$ and for different graph topologies and $\mathcal{T}_F$ models. We considered values of $n$ ranging from 15 to 55 with intervals of 10, and set $k_{ij} = k$ for all $(i,j) \in \mathcal{E}$ with $k \in \{12, 20\}$, graph topologies including complete graphs, $\lceil\sqrt{n}\rceil \times \lceil\sqrt{n}\rceil$ toroidal grids, and sparse graphs generated from Erdős-Rényi $\mathcal{G}(n,p)$ models with parameter $p = 2\log^2(n)/n$, and $\mathcal{T}_F$ models such as standard Thurstone (Case V) and BTL models. For each choice of parameters, we generated 400 models by randomly sampling weights with $b = F^{-1}(0.98)/2$ and generated synthetic comparison data. The scaled test statistic $k \cdot T/n$ was computed for every parameter choice, and the 95th percentile value of this scaled $T$ was identified as the threshold $\gamma$. Figure 1 plots these 95th percentile values with respect to $n$ for various parameter choices. Notably, the value $\gamma$ remains roughly constant with $n, k$ and model $\mathcal{T}_F$ for complete graphs. For all these cases, the values stabilize approximately to a certain constant, illustrating that the thresholds obtained via the empirical-quantile-based approach follow the same high-level scaling as the theoretical threshold in (37).

In our next experiment, we apply our test to the LMSYS chatbot leaderboard (Chiang et al., 2024), a widely used benchmark for evaluating the performance of LLMs. The dataset contains a collection of pairwise comparisons between various LLMs based on their response to prompts, which are then used to obtain Elo ratings. We retain the directional nature of comparisons, where an "$i$ vs. $j$" comparison indicates model $i$ as the first response and $j$ as the second during the evaluation. We rank the LLMs based on their frequency of appearance in the dataset and perform the test repeatedly on the top-$n$ LLMs in this ordering, with $n$ ranging from 5 to 21 with gaps of 2, for both Thurstone and

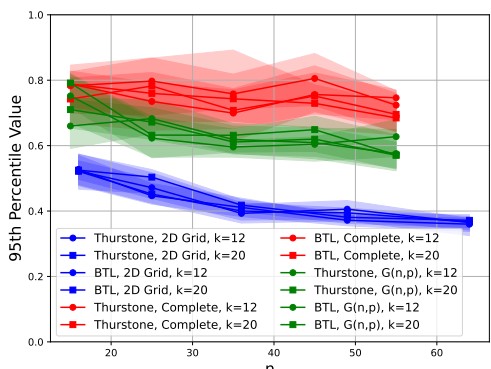

Figure 1: Estimated scaled threshold for various values of $n, k$, graph topologies, and $\mathcal{T}_F$ models.

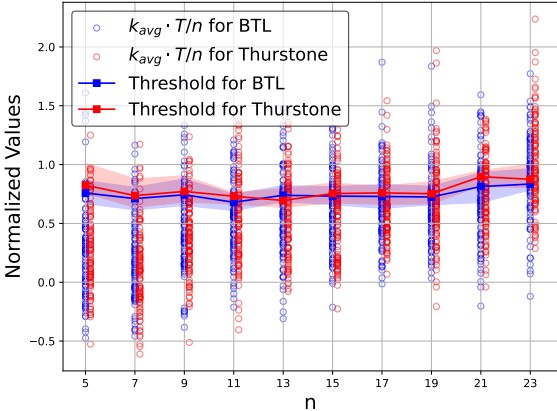

Figure 2: Scaled test statistics and the estimated thresholds evaluated on the LMSYS dataset.

BTL models. For each $n$, we plot the values in Figure 2 of the (scaled) test statistic $k_{\text{avg}} \cdot T/n$ and the obtained (scaled) thresholds using the quantile approach (with the same parameters as above), where $k_{\text{avg}}$ is the average of $k_{ij}$ over all $(i, j) \in \mathcal{E}$. By randomizing over the partitioning of the dataset $\mathcal{Z}$ into $\mathcal{Z}_1$ and $\mathcal{Z}_2$ and computing $T$ each time, we essentially obtain a distribution of $T$ and plot these values in Figure 2 as a scatter plot. The figure highlights that both BTL and Thurstone models perform well in modeling for smaller values of $n$ with only $10\%$ of samples above the threshold, but exhibit significant deviations for larger values of $n$ as around $60\%$ of samples are above the threshold for $n = 21$. The $60\%$ of samples above the threshold can be interpreted as a bootstrapped estimate of the test's power indicating $40\%$ chance that the model lies within $95\%$ statistical deviations from BTL. The figure also shows that deviation from Thurstone increases as $n$ increases, a top-9 batch size provides a statistically accurate fit to the Thurstone model, and the deviations are significant for $n \geq 21$. The results suggest that when using a multi-tier Elo rating system (Brams & Ismail, 2024), a group size of approximately 9 leads to accurate modeling for the top-$n$ models, ranked based on their total available data.

## 5. Conclusion

In this work, we developed a rigorous testing framework to determine whether pairwise comparison data is generated by an underlying generalized Thurstone model with a given choice function $F$. We derived both upper and lower bounds on the critical threshold of our testing problem, which depended on the topology of the observation graph. These bounds were shown to be tight for certain graph classes, such as complete graphs. In addition, we proposed a hypothesis test based on our notion of separation distance and established theoretical guarantees for this test, including time-uniform bounds on type I and II error probabilities as well as a minimax lower bound on the risk of the testing problem. Alongside this, auxiliary results such as error bounds for parameter estimation and confidence intervals under the null hypothesis were derived. To validate our findings, we conducted experiments on both synthetic and real-world datasets and introduced a data-driven approach for determining the test threshold.

This study opens up several avenues for future research. For instance, extending the hypothesis testing framework to handle general multi-way comparisons rather than pairwise comparisons is one such direction. Another direction is developing active testing techniques within the framework of generalized Thurstone models that optimize test performance. Finally, extending the testing and estimation frameworks for such models to dependent data is also an important avenue as real-world data often exhibits correlations that could affect inference results.

## Acknowledgements

This work was supported by the National Science Foundation (NSF) CAREER Award under Grant CCF-2337808.

## Impact Statement

This work is primarily theoretical and does not have immediate societal consequences. However, if practitioners apply our approaches to make decisions in real-world settings, there could be both positive and negative implications. On the one hand, our methods can help determine accurate models for comparison data, such as whether data conforms to a $\mathcal{T}_F$ model, or help detect certain kinds of biases in data. On the other hand, our methods concern ranking models and using any ranking mechanism without due care in the real-world may unintentionally exacerbate existing inequalities, such as unequal access to resources and opportunities.

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

## A. Proofs of Proposition 2.4 and Theorems 3.1 and 3.3

### A.1. Additional Notation and Preliminaries

We begin by introducing some additional notations and discussing some necessary preliminaries that will be used throughout our proofs. To simplify our notation, we use $\hat{l}(w)$ to denote $l(w; \{\hat{p}_{ij} : (i,j) \in \mathcal{E}\})$ where $\hat{p}_{ij}$ are computed based on the partitioned dataset $\mathcal{Z}_1$. Similarly, we use $l^*(w)$ to denote $l(w; \{p_{ij} : (i,j) \in \mathcal{E}\})$ where $p_{ij}$ are the actual underlying pairwise comparison probabilities. When $w = w^*$ or $w = \hat{w}$, we simplify the notation by using $\mathsf{F}$ and $\hat{\mathsf{F}}$ to denote the matrices $\mathsf{F}(w^*)$ and $\mathsf{F}(\hat{w})$ (cf. (4)) respectively, for brevity. We say that a random variable $X$ is $\mu^2$-sub-Gaussian, if it satisfies the condition, $\log(\mathbb{E}[\exp(sX)]) \leq \mu^2 s^2 / 2$ for all $s \in \mathbb{R}$.

Notably, $\hat{w}$ is computed as in (7) even though the data may not conform to an underlying $\mathcal{T}_F$ model. Throughout the appendices, we denote various constants using overlapping labels, such as $c, c_1, c_2, \ldots$ to simplify our notation and facilitate readability. Moreover, we define $\mathcal{E}_+$ to denote the set $\{(i,j) \in \mathcal{E} : j > i\}$.

#### A.1.1. PRELIMINARIES

Recall that, as defined in (8), $w^*$ represents the weights of a $\mathcal{T}_F$ model that best approximates the pairwise comparison model $\{p_{ij}, (i,j) \in \mathcal{E}\}$. Now, we show that any pairwise comparison model can be converted into its skew-symmetric counterpart $\left\{ \frac{p_{ij} + 1 - p_{ji}}{2}, (i,j) \in \mathcal{E} \right\}$ such that both of them share the same optimal weights.

**Lemma A.1** (Skew-symmetrized Model). *For a symmetric edge set $\mathcal{E}$, the pairwise model $\{p_{ij} : (i,j) \in \mathcal{E}\}$ and and its skew-symmetric counterpart $\left\{ \frac{p_{ij} + 1 - p_{ji}}{2} : (i,j) \in \mathcal{E} \right\}$ has the have the same optimal $\mathcal{T}_F$ weights $w^*$ as defined in (8).*

**Proof:** Note that the weighted negative log-likelihood objective can be written as

$$l^*(w) = \underset{w \in \mathcal{W} : w^{\mathsf{T}}\mathbf{1}=0}{\arg\min} - \sum_{(i,j)\in\mathcal{E}} p_{ij}\log(F(w_i - w_j)) + (1 - p_{ij})\log(1 - F(w_i - w_j))$$

$$\overset{\zeta_1}{=} \underset{w \in \mathcal{W} : w^{\mathsf{T}}\mathbf{1}=0}{\arg\min} - \sum_{(i,j)\in\mathcal{E}} p_{ij}\log(1 - F(w_j - w_i)) + (1 - p_{ij})\log(F(w_j - w_i))$$

$$\overset{\zeta_2}{=} \underset{w \in \mathcal{W} : w^{\mathsf{T}}\mathbf{1}=0}{\arg\min} - \sum_{(i,j)\in\mathcal{E}} \left(\frac{p_{ij} + 1 - p_{ji}}{2}\right)\log(F(w_i - w_j)) + \left(\frac{1 - p_{ij} + p_{ji}}{2}\right)\log(F(w_j - w_i))$$

where $\zeta_1$ follows since $F(-x) = 1 - F(x)$ and $\zeta_2$ follows by adding the first two equations and dividing by two. $\square$

Therefore, for any pairwise comparison model $\{p_{ij} : (i,j) \in \mathcal{E}\}$, we can define its skew-symmetrized counterpart $\{q_{ij} : (i,j) \in \mathcal{E}\}$ where

$$\forall (i,j) \in \mathcal{E}, q_{ij} \triangleq \frac{p_{ij} + 1 - p_{ji}}{2}. \tag{17}$$

We call these transformed probabilities $q_{ij}$ as skew-symmetrized probabilities because we have $q_{ij} + q_{ji} = 1$, and thereby this transformation effectively removes any distinctions between "$i$ vs. $j$" and "$j$ vs. $i$" comparisons. Also, note that for any pairwise comparison model satisfying Assumption 2.3, its skew-symmetrized model also satisfies it. In a similar manner, we can define $\hat{q}_{ij} = \frac{\hat{p}_{ij} + 1 - \hat{p}_{ji}}{2}$ as the skew-symmetrized version of the empirical probabilities. With this notation in place, we are ready to state the proof of Proposition 2.4 below.

### A.2. Proof of Proposition 2.4

**Uniqueness:** The uniqueness of $w$ follows directly from the strong log-concavity of $F(\cdot)$. This is because if $v^*, w^* \in \mathcal{W}_b$ are any two non-unique solutions of (8) such that $l^*(v^*) = l^*(w^*)$, then by strong log-concavity of $F$ and the fact that $q_{ij} > 0$ for $(i,j) \in \mathcal{E}$ along with connectedness of graph, for any $\theta \in (0,1)$, we have

$$\theta l^*(v^*) + (1-\theta)l^*(w^*) = -2\theta \sum_{(i,j)\in\mathcal{E}} q_{ij}\log(F(v_i^* - v_j^*)) - 2(1-\theta)\sum_{(i,j)\in\mathcal{E}} q_{ij}\log(F(w_i^* - w_j^*))$$

$$> -2 \sum_{(i,j)\in\mathcal{E}} q_{ij} \log \left( F\big(\theta\big(v_i^* - v_j^*\big) + (1-\theta)\big(w_i^* - w_j^*\big)\big)\right)$$
$$= l^*(\theta v^* + (1-\theta)w^*).$$

This gives a contradiction since $\theta v^* + (1-\theta)w^*$ achieves a higher likelihood (or a lower objective value). The existence of $w^*$ under Assumption 2.3 and finite $b$ follows from the extreme value theorem since a continuous function is being optimized over a compact set. Notably, the existence of $w^*$ also holds for disconnected graphs. As a bonus, we provide a proof of existence even when the parameter $b = \infty$, but for a connected graph $\mathcal{G}$.

**Existence:** Now, we will utilize the connectedness of graph $\mathcal{G}$ and Assumption 2.3 to show the existence of $w^*$. Define a sequence $\{w^{(m)} \in \mathbb{R}^n : m \in \mathbb{N} \cup \{0\}\}$ as

$$w^{(m)} = \operatorname*{argmin}_{\substack{w_1=0:\\ \|w\|_\infty \le m}} l^*(w).$$

Clearly, $w^{(m)}$ exists as the optimization of a convex function $l^*(\cdot)$ is being performed over a compact set. Define the following sets as the components of $w$ that potentially diverge to $\infty$:

$$S_+ = \left\{i \in [n] : \limsup_m \left(w^{(m)}\right)_i = +\infty\right\}, \quad S_- = \left\{i \in [n] : \liminf_m \left(w^{(m)}\right)_i = -\infty\right\}.$$

We will show that $S_+ = S_- = \varnothing$. Notably, if $S_+ \ne \varnothing$, then we consider the partition of $[n]$ as $S_+ \cup S_+^c$. Clearly, $1 \in S_+^c \ne \varnothing$. Since the observation graph $\mathcal{G}$ is connected, for some $i \in S_+$ there exists $j \in S_+^c$ such that $q_{ji} > 0$ (by Assumption 2.3). This implies that $-q_{ji}\log(F(w_j^{(m)} - w_i^{(m)})) \to +\infty$ as $m \to +\infty$. Hence, we can find a constant $A > 0$ such that on the set $\{w_i - w_j \ge A\}$, we have

$$-q_{ji}\log(F(w_i - w_j)) > l^*(w^{(0)}).$$

Equivalently, for any $w$ with $w_i - w_j > A$, we have

$$l^*(w) \ge -q_{ji}\log(F(A)) > l^*(w^{(0)}) \ge l^*(w^{(m)}),$$

where the first inequality follows since each term in $l^*(\cdot)$ is non-negative. Therefore, we must have $w_i^{(m)} \le w_j^{(m)} + A$ for all $k \in \mathbb{N}$. Since $i \in S_+$, it follows that $j \in S_+$ by definition, which contradicts the assumption that $j \in S_+^c$. Hence, we conclude that $S_+ = \emptyset$. A similar argument shows that $S_- = \emptyset$. The fact that $S_+ = S_- = \varnothing$ implies that the sequence $\{w^{(m)} : m \in \mathbb{N} \cup \{0\}\}$ admits a convergent subsequence, which proves the existence of $w^*$. $\qquad\square$

### A.3. Proof of Theorem 3.1

The upper bound is trivial to prove

$$\inf_{w\in\mathcal{W}_b} \|P - \mathsf{F}(w)\|_\mathrm{F} \le \|P - \mathsf{F}(w^*)\|_\mathrm{F} = \|P - \mathsf{F}\|_\mathrm{F},$$

where $\mathsf{F}$ is the pairwise probability matrix associated with the optimal weights. Now to prove the lower bound, observe that

$$\inf_{w\in\mathcal{W}_b} \sum_{(i,j)\in\mathcal{E}} (p_{ij} - F(w_i - w_j))^2$$
$$\ge \inf_{w\in\mathcal{W}_b} F(-2b)(1 - F(-2b)) \sum_{(i,j)\in\mathcal{E}} \frac{(p_{ij} - F(w_i - w_j))^2}{F(w_i - w_j)(1 - F(w_i - w_j))}$$
$$\overset{\zeta_1}{=} c \inf_{w\in\mathcal{W}_b} \sum_{(i,j)\in\mathcal{E}} \chi^2(\mathsf{Bernoulli}(p_{ij})\|\mathsf{Bernoulli}(F(w_i - w_j)))$$
$$\overset{\zeta_2}{\ge} c \inf_{w\in\mathcal{W}_b} \sum_{(i,j)\in\mathcal{E}} D_{\mathsf{KL}}(\mathsf{Bernoulli}(p_{ij})\|\mathsf{Bernoulli}(F(w_i - w_j)))$$
$$= c \inf_{w\in\mathcal{W}_b} \sum_{(i,j)\in\mathcal{E}} p_{ij}\log\left(\frac{p_{ij}}{F(w_i - w_j)}\right) + (1 - p_{ij})\log\left(\frac{1 - p_{ij}}{1 - F(w_i - w_j)}\right)$$

$$\overset{\zeta_3}{=} c \sum_{(i,j)\in\mathcal{E}} p_{ij} \log\left(\frac{p_{ij}}{F(w_i^* - w_j^*)}\right) + (1 - p_{ij})\log\left(\frac{1 - p_{ij}}{1 - F(w_i^* - w_j^*)}\right)$$

$$\overset{\zeta_4}{\geq} 2c \sum_{(i,j)\in\mathcal{E}} \|\mathsf{Bernoulli}(p_{ij}) - \mathsf{Bernoulli}(F(w_i^* - w_j^*))\|_{\mathsf{TV}}^2$$

$$= 2c \sum_{(i,j)\in\mathcal{E}} (p_{ij} - F(w_i^* - w_j^*))^2,$$

where, in $\zeta_1$ we set $c = F(-2b)(1 - F(-2b))$ and $\chi^2(\cdot||\cdot)$ denotes the $\chi^2$-divergence between two Bernoulli random variables and in $\zeta_2$ we utilize the fact that $\chi^2(R||Q) \geq D_{\mathsf{KL}}(R||Q)$ for two distributions $R$ and $Q$ and where $D_{\mathsf{KL}}(\cdot||\cdot)$ denotes the Kullback-Leibler (KL) divergence between two distributions, $\zeta_3$ follows since $w^*$ are the optimal weights maximizing (8) for the $\mathcal{T}_F$ model. Finally, $\zeta_4$ follows by Pinsker's inequality $D_{\mathsf{KL}}(R||P) \geq 2\|R - P\|_{\mathsf{TV}}^2$ and thereby completing the proof. $\qquad\square$

### A.4. Proof of Theorem 3.3

We begin by recalling the definition of $\hat{w} \in \arg\min_{w \in \mathcal{W}_b} \hat{l}(w)$ in terms of the symmetrized probabilities $q_{ij}$ defined in (17) as

$$\hat{l}(w) = -2 \sum_{(i,j)\in\mathcal{E}_+} \hat{q}_{ij} \log F(w_i - w_j) + (1 - \hat{q}_{ij})\log(1 - F(w_i - w_j)).$$

Observe that since $\hat{w}$ is an optimal solution and $w^*$ is a feasible point for the problem in (7), therefore we have $\hat{l}(\hat{w}) \leq \hat{l}(w^*)$. Moreover, since $w^*$ is the optimal solution of a convex function $l^*(w)$, therefore we have the optimality condition $\nabla l^*(w^*)^{\mathrm{T}}(w - w^*) \geq 0$ for all $w \in \mathcal{W}_b$. Now, by subtracting the quantity $\nabla\hat{l}(w^*)^{\mathrm{T}}(\hat{w} - w^*)$ from both sides of $\hat{l}(\hat{w}) \leq \hat{l}(w^*)$ gives

$$\hat{l}(\hat{w}) - \hat{l}(w^*) - \nabla\hat{l}(w^*)^{\mathrm{T}}(\hat{w} - w^*) \leq -\nabla\hat{l}(w^*)^{\mathrm{T}}(\hat{w} - w^*) \tag{18}$$

$$\overset{\zeta_1}{\leq} -(\nabla\hat{l}(w^*) - \nabla l^*(w^*))^{\mathrm{T}}(\hat{w} - w^*)$$

$$\overset{\zeta_2}{\leq} \|\nabla\hat{l}(w^*) - \nabla l^*(w^*)\|_{L^\dagger}\|\hat{w} - w^*\|_L, \tag{19}$$

where $\zeta_1$ follows since $\nabla l^*(w^*)^{\mathrm{T}}(w - w^*) \geq 0$ for all $w \in \mathcal{W}$ and $\zeta_2$ follows from (Shah et al., 2016, Lemma 16) where $\|\cdot\|_L$ is the semi-norm induced by the Laplacian matrix $L$ of graph $\mathcal{G}$ and $L^\dagger$ is the Moore-Penrose pseudoinverse of $L$. Now observe that by the chain rule, the Hessian of $\hat{l}$ is given by

$$\nabla^2\hat{l}(w) = -2 \sum_{(i,j)\in\mathcal{E}_+} \left(\hat{q}_{ij}\frac{\mathrm{d}^2}{\mathrm{d}t^2}\log(F(t))|_{t=w^{\mathrm{T}}x_{ij}} + (1 - \hat{q}_{ij})\frac{\mathrm{d}^2}{\mathrm{d}t^2}\log(1 - F(t))|_{t=w^{\mathrm{T}}x_{ij}}\right)x_{ij}x_{ij}^{\mathrm{T}},$$

Since by our assumption that $F(t)$ is $\alpha$-strongly log-concave on the set $[-2b, 2b]$, this implies $-\frac{\mathrm{d}^2}{\mathrm{d}t^2}\log(F(t)) \geq \alpha$. Moreover, since $F(-t) = 1 - F(t)$, we also have $-\frac{\mathrm{d}^2}{\mathrm{d}t^2}\log(1 - F(t)) \geq \alpha$ for all $t \in [-2b, 2b]$. Therefore, for any $v \in \mathbb{R}^n$ with $v^{\mathrm{T}}\mathbf{1} = 0$, we have

$$v^{\mathrm{T}}\nabla^2\hat{\ell}(w^*)v \geq 2\alpha\|Xv\|_2^2 = 2\alpha\|v\|_L^2.$$

Thus, by definition of strong-convexity, the left side of (18) can be lower bounded by $\alpha\|\hat{w} - w^*\|_L^2$. Therefore, utilizing the bound in (19), we obtain the following inequality

$$\alpha\|\hat{w} - w^*\|_L^2 \leq \|\nabla\hat{l}(w^*) - \nabla l^*(w^*)\|_{L^\dagger}\|\hat{w} - w^*\|_L.$$

Canceling $\|\hat{w} - w^*\|_L$ and squaring both sides leads to the following error bound on $\|\hat{w} - w^*\|_L$ as

$$\|\hat{w} - w^*\|_L^2 \leq \frac{1}{\alpha^2}\|\nabla\hat{l}(w^*) - \nabla l^*(w^*)\|_{L^\dagger}^2. \tag{20}$$

Now, it remains to bound the term $\|\nabla\hat{l}(w^*) - \nabla l^*(w^*)\|_{L^\dagger}$. Note that we can express the respective quantities as:

$$\nabla\hat{l}(w^*) = -2 \sum_{(i,j)\in\mathcal{E}_+} \left(\hat{q}_{ij}\frac{F'(w^{*\mathrm{T}}x_{ij})}{F(w^{*\mathrm{T}}x_{ij})} - (1 - \hat{q}_{ij})\frac{F'(w^{*\mathrm{T}}x_{ij})}{1 - F(w^{*\mathrm{T}}x_{ij})}\right)x_{ij}$$

$$= -2 \sum_{(i,j) \in \mathcal{E}_+} \frac{(\hat{q}_{ij} - F(w_i^* - w_j^*))F'(w_i^* - w_j^*)}{F(w_i^* - w_j^*)(1 - F(w_i^* - w_j^*))} x_{ij}, \tag{21}$$

$$\nabla l^*(w^*) = -2 \sum_{(i,j) \in \mathcal{E}_+} \frac{(q_{ij} - F(w_i^* - w_j^*))F'(w_i^* - w_j^*)}{F(w_i^* - w_j^*)(1 - F(w_i^* - w_j^*))} x_{ij}. \tag{22}$$

Therefore, subtracting the two equations gives

$$\nabla \hat{l}(w^*) - \nabla l^*(w^*) = -2 \sum_{(i,j) \in \mathcal{E}_+} \frac{(\hat{q}_{ij} - q_{ij})F'(w_i^* - w_j^*)}{F(w_i^* - w_j^*)(1 - F(w_i^* - w_j^*))} x_{ij} = -2X^{\mathrm{T}}v, \tag{23}$$

where $v \in \mathbb{R}^{|\mathcal{E}|/2}$ is a vector formed by entries $v_{ij}$ for $(i,j) \in \mathcal{E}_+$. and quantities $v_{ij}$ are defined as

$$v_{ij} \triangleq (\hat{q}_{ij} - q_{ij}) \times \frac{F'(w_i^* - w_j^*)}{F(w_i^* - w_j^*)(1 - F(w_i^* - w_j^*))}.$$

Note that the entries of the vector $v$ are independent and have a mean of zero. Furthermore, we also have:

$$\sup_{x \in [-2b, 2b]} \frac{F'(x)}{F(x)(1 - F(x))} \leq \frac{\beta}{F(-2b)(1 - F(-2b))} \triangleq \tilde{\beta}.$$

Additionally, for any $(i,j) \in \mathcal{E}_+$, an application of the Hoeffding's inequality on $\hat{q}_{ij} - q_{ij}$ yields the following tail bound

$$\forall\, t > 0,\ \mathbb{P}(|\hat{q}_{ij} - q_{ij}| > t) = \mathbb{P}\left( \frac{1}{2k} \left| \sum_{m=1}^{k} (Z_{ij}^m - p_{ij}) + \sum_{m=1}^{k} (Z_{ji}^m - p_{ji}) \right| > t \right)$$
$$\leq 2 \exp(-2kt^2).$$

Consequently, $v$ is a vector whose each entry is independent with zero mean and $\frac{\tilde{\beta}^2}{4k}$-sub-gaussian. Now, observe that we can express $\|\nabla \hat{l}(w^*) - \nabla l^*(w^*)\|_{L^\dagger}^2$ in quadratic form as

$$\|\nabla \hat{l}(w^*) - \nabla l^*(w^*)\|_{L^\dagger}^2 = 4v^{\mathrm{T}}XL^\dagger X^{\mathrm{T}}v. \tag{24}$$

Now, combining (20) and (24) we can upper-bound $\mathbb{E}[\|\hat{w} - w^*\|_L^2]$ as

$$\mathbb{E}[\|\hat{w} - w^*\|_L^2] \leq \frac{1}{\alpha^2} \mathbb{E}[\|\nabla \hat{l}(w^*) - \nabla l^*(w^*)\|_{L^\dagger}^2]$$
$$= \frac{4}{\alpha^2} \mathbb{E}[v^{\mathrm{T}}XL^\dagger X^{\mathrm{T}}v]$$
$$\leq \frac{\tilde{\beta}^2}{k\alpha^2} \mathrm{tr}(XL^\dagger X^{\mathrm{T}}) = \frac{(n-1)\tilde{\beta}^2}{k\alpha^2}, \tag{25}$$

where $\mathrm{tr}$ denotes the trace operator and we have $\mathrm{tr}(XL^\dagger X^{\mathrm{T}}) = \mathrm{tr}(L^\dagger X^{\mathrm{T}}X) = \mathrm{tr}(L^\dagger L) = n - 1$. Hence, by an application of Hanson-Wright inequality (Rudelson & Vershynin, 2013) combined with usage of (20) and (24) as above, we have the following concentration bounds on $\|\hat{w} - w^*\|_L^2$:

$$\forall t > 0, \mathbb{P}\left( \|\hat{w} - w^*\|_L^2 - \frac{n\tilde{\beta}^2}{k\alpha^2} > t \right) \leq 2\exp\left( -c\min\left\{ \frac{t^2 k^2 \alpha^4}{\tilde{\beta}^4 \|XL^\dagger X^{\mathrm{T}}\|_{\mathrm{F}}^2}, \frac{tk\alpha^2}{\tilde{\beta}^2 \|XL^\dagger X^{\mathrm{T}}\|_2} \right\} \right)$$
$$= 2\exp\left( -c\min\left\{ \frac{t^2 k^2 \alpha^4}{\tilde{\beta}^4 (n-1)}, \frac{tk\alpha^2}{\tilde{\beta}^2} \right\} \right).$$

Hence, by a simple calculation, we can conclude that for some constant $c$, we have

$$\text{for all } t \geq 1,\ \mathbb{P}\left( \|\hat{w} - w^*\|_L^2 > t\frac{cn\tilde{\beta}^2}{k\alpha^2} \right) \leq e^{-t}. \tag{26}$$

**Bounding the $p$th moment:** Let $A$ denote the quantity: $A = \sqrt{cn\tilde{\beta}^2/k\alpha^2}$. Now the bound on the pth moment is obtained by integration and utilizing the tail bound in (26) as

$$
\begin{aligned}
\mathbb{E}[\|\hat{w} - w^*\|_L^p] &= p \int_0^\infty t^{p-1} \mathbb{P}(\|\hat{w} - w^*\|_L > t) \, dt \\
&= p \int_0^A t^{p-1} \mathbb{P}(\|\hat{w} - w^*\|_L > t) \, dt + p \int_A^\infty t^{p-1} \mathbb{P}(\|\hat{w} - w^*\|_L > t) \, dt \\
&\leq \int_0^A t^{p-1} \, dt + p \int_1^\infty (At)^{p-1} \mathbb{P}(\|\hat{w} - w^*\|_L > tA) A \, dt \\
&\leq A^p + pA^p \int_0^\infty t^{p-1} e^{-\sqrt{t}} \leq c(p) A^p.
\end{aligned}
$$

Substituting the value of $A$ in the above expression completes the proof. $\qquad\square$

# B. Proof of Theorem 3.2

We begin by recalling the test statistic $T$ from (15) as

$$
T = \sum_{(i,j) \in \mathcal{E}} \frac{Z_{ij}(Z_{ij} - 1)}{k'_{ij}(k'_{ij} - 1)} + F(\hat{w}_i - \hat{w}_j)^2 - \frac{2Z_{ij}}{k'_{ij}} F(\hat{w}_i - \hat{w}_j).
$$

where $\hat{w}$ is calculated based on the data in $\mathcal{Z}_1$ and $Z_{ij}$ are calculated based on the data in $\mathcal{Z}_2$. The expected value of $T$ conditioned on the weights $\hat{w}$ or equivalently conditioned on the data $\mathcal{Z}_1$ is given by

$$
\begin{aligned}
\mathbb{E}[T \mid \mathcal{Z}_1] &= \sum_{(i,j) \in \mathcal{E}} \mathbb{E}\left[\frac{Z_{ij}(Z_{ij} - 1)}{k'_{ij}(k'_{ij} - 1)} \Big| \mathcal{Z}_1\right] + F(\hat{w}_i - \hat{w}_j)^2 - 2\mathbb{E}\left[\frac{Z_{ij}}{k'_{ij}} \Big| Z_1\right] F(\hat{w}_i - \hat{w}_j) \\
&\overset{\zeta}{=} \sum_{(i,j) \in \mathcal{E}} p_{ij}^2 + F(\hat{w}_i - \hat{w}_j)^2 - 2p_{ij} F(\hat{w}_i - \hat{w}_j) \\
&= \sum_{(i,j) \in \mathcal{E}} (p_{ij} - F(\hat{w}_i - \hat{w}_j))^2,
\end{aligned}
\tag{27}
$$

where in $\zeta$ we have utilized the fact that $\mathbb{E}\left[\frac{Z_{ij}(Z_{ij}-1)}{k'_{ij}(k'_{ij}-1)} | \mathcal{Z}_1\right] = p_{ij}^2$. Hence, the expected value of $T$ is given by

$$
\begin{aligned}
\mathbb{E}[T] = \mathbb{E}[\mathbb{E}[T|\mathcal{Z}_1]] &= \sum_{(i,j) \in \mathcal{E}} \mathbb{E}[(p_{ij} - F(\hat{w}_i - \hat{w}_j))^2] \\
&= \sum_{(i,j) \in \mathcal{E}} (p_{ij} - F(w_i^* - w_j^*))^2 + \mathbb{E}[(F(w_i^* - w_j^*) - F(\hat{w}_i - \hat{w}_j))^2] \\
&\quad + 2\mathbb{E}[(p_{ij} - F(w_i^* - w_j^*))(F(w_i^* - w_j^*) - F(\hat{w}_i - \hat{w}_j))] \\
&\leq \|P - \mathsf{F}\|_F^2 + \mathbb{E}[\|\mathsf{F} - \hat{\mathsf{F}}\|_F^2] + 2\|P - \mathsf{F}\|_F \mathbb{E}[\|\mathsf{F} - \hat{\mathsf{F}}\|_F],
\end{aligned}
\tag{28}
$$

where $\mathsf{F}, \hat{\mathsf{F}} \in \mathbb{R}^{n \times n}$ are matrices defined in Appendix A.1. In order to find bounds on estimation error (such as terms like $\mathbb{E}[\|\mathsf{F} - \hat{\mathsf{F}}\|_F^p]$), we will utilize our simplifying assumption that $k_{ij} = 2k$ for all $(i,j) \in \mathcal{E}$. Now the ensuing lemma provides the bounds on the $p$th moments $\mathbb{E}[\|\mathsf{F} - \hat{\mathsf{F}}\|_F^p]$.

**Lemma B.1** ($p$th Moment Bound). *For matrices $\mathsf{F}$ and $\hat{\mathsf{F}}$ defined as in Appendix A.1 and $p \geq 1$, there exists a constant $c_p$, possibly dependent on $p, \alpha, \beta, b$, such that the following bound holds on the pth moment of the Frobenius norm $\mathbb{E}\left[\|\mathsf{F} - \hat{\mathsf{F}}\|_F^p\right]$:*

$$
\mathbb{E}\left[\|\mathsf{F} - \hat{\mathsf{F}}\|_F^p\right] \leq c_p \left(\frac{n}{k}\right)^{p/2}.
$$

*Moreover, there exists a constant $c$ such that we have the following tail bound:*

$$
\forall t \geq 1, \quad \mathbb{P}\left(\|\mathsf{F} - \hat{\mathsf{F}}\|_F^2 \geq t\frac{cn}{k}\right) \leq e^{-t}.
$$

The proof is provided in Appendix B.1. Thus, utilizing Lemma B.1 and (28), we have obtain the following bound for some constant $c_1$ and $c_2$:

$$\mathbb{E}[T] \leq \|P - \mathsf{F}\|_{\mathrm{F}}^2 + c_2 \frac{n}{k} + 2c_1 \sqrt{\frac{n}{k}} \|P - \mathsf{F}\|_{\mathrm{F}}.$$

Let $\mathbb{E}_{H_0}[\cdot]$ and $\mathbb{E}_{H_1}[\cdot]$ denote the expectation operators under hypotheses $H_0$ and $H_1$, respectively. In essence, we have established the following bounds on $\mathbb{E}_{H_0}[T]$:

$$|\mathbb{E}_{H_0}[T]| \leq c_2 \frac{n}{k}, \tag{29}$$

In a similar manner, we can obtain complementary lower bounds on $\mathbb{E}_{H_1}[T]$ (cf. (28)). Consequently, we have the following lower bound on $\mathbb{E}_{H_1}[T]$:

$$\mathbb{E}_{H_1}[T] \geq \|P - \mathsf{F}\|_{\mathrm{F}}^2 - 2c_1 \sqrt{\frac{n}{k}} \|P - \mathsf{F}\|_{\mathrm{F}}. \tag{30}$$

**Bounding variance:** Now we will find bounds on $\mathrm{var}(T)$ under the two hypotheses. For this, we will make use of the law of total variance by conditioning $T$ with respect to $\mathcal{Z}_1$ as

$$\mathrm{var}(T) = \mathbb{E}[\mathrm{var}(T \mid \mathcal{Z}_1)] + \mathrm{var}(\mathbb{E}[T \mid \mathcal{Z}_1]). \tag{31}$$

First, we will examine the term $\mathbb{E}[\mathrm{var}(T|\mathcal{Z}_1)]$. Note that conditioned on $\mathcal{Z}_1$, the term $F(\hat{w}_i - \hat{w}_j)^2$ is constant and does not contribute to $\mathrm{var}(T|\mathcal{Z}_1)$. Moreover, $Z_{ij}$ for $(i,j) \in \mathcal{E}$ are mutually independent, and hence, we can analytically find the expression for $\mathrm{var}(T|\mathcal{Z}_1)$ as

$$\mathrm{var}(T \mid \mathcal{Z}_1) \stackrel{\zeta_1}{=} \sum_{(i,j) \in \mathcal{E}} \mathrm{var}\left(\frac{Z_{ij}(Z_{ij}-1)}{k'_{ij}(k'_{ij}-1)}\right) + 4F(\hat{w}_i - \hat{w}_j)^2 \mathrm{var}\left(\frac{Z_{ij}}{k'_{ij}}\right)$$

$$- 4F(\hat{w}_i - \hat{w}_j)\left(\frac{\mathbb{E}[Z_{ij}^2(Z_{ij}-1)]}{(k'_{ij})^2(k'_{ij}-1)} - \frac{\mathbb{E}[Z_{ij}(Z_{ij}-1)]}{k'_{ij}(k'_{ij}-1)} \frac{\mathbb{E}[Z_{ij}]}{k'_{ij}}\right)$$

$$\stackrel{\zeta_2}{=} \sum_{(i,j) \in \mathcal{E}} \frac{2p_{ij}^2 + 4(k'_{ij}-2)p_{ij}^3 + (6-4k'_{ij})p_{ij}^4}{k'_{ij}(k'_{ij}-1)} + \frac{4F(\hat{w}_i - \hat{w}_j)^2 p_{ij}(1-p_{ij})}{k'_{ij}}$$

$$- 4F(\hat{w}_i - \hat{w}_j)\frac{2(p_{ij}^2 - p_{ij}^3)}{k'_{ij}},$$

where $\zeta_1$ follows from the variance of sum technique and $\zeta_2$ follows from the expressions for the first four moments of Binomial random variables and some basic algebra. Now, in order to bound $\mathbb{E}[\mathrm{var}(T \mid \mathcal{Z}_1)]$, we will substitute all $k'_{ij} = k$ for all $(i,j) \in \mathcal{E}$ and simplify the above expression as:

$$\mathbb{E}[\mathrm{var}(T \mid \mathcal{Z}_1)] = \sum_{(i,j) \in \mathcal{E}} \frac{2p_{ij}^2 - 4p_{ij}^3 + 2p_{ij}^4}{k(k-1)}$$

$$+ p_{ij}(1-p_{ij})\left(\frac{4p_{ij}^2}{k} + \frac{4\mathbb{E}[F^2(\hat{w}_i - \hat{w}_j)]}{k} - \frac{8\mathbb{E}[F(\hat{w}_i - \hat{w}_j)]p_{ij}}{k}\right)$$

$$= \sum_{(i,j) \in \mathcal{E}} \frac{2p_{ij}^2(1-p_{ij})^2}{k(k-1)} + \frac{4p_{ij}(1-p_{ij})}{k}(p_{ij} - \mathbb{E}[F(\hat{w}_i - \hat{w}_j)])^2$$

$$\leq \frac{nd_{\max}}{8k(k-1)} + \frac{1}{k}\|P - \mathbb{E}[\hat{\mathsf{F}}]\|_{\mathrm{F}}^2$$

$$\leq \frac{nd_{\max}}{8k(k-1)} + \frac{1}{k}(\|P - \mathsf{F}\|_{\mathrm{F}} + \|\mathsf{F} - \mathbb{E}[\hat{\mathsf{F}}]\|_{\mathrm{F}})^2$$

$$\leq \frac{nd_{\max}}{8k(k-1)} + \frac{1}{k}(\|P - \mathsf{F}\|_{\mathrm{F}} + \mathbb{E}[\|\mathsf{F} - \hat{\mathsf{F}}\|_{\mathrm{F}}])^2$$

$$\leq \frac{nd_{\max}}{8k(k-1)} + \frac{1}{k}\left(\|P - \mathsf{F}\|_{\mathrm{F}} + c_1\sqrt{\frac{n}{k}}\right)^2, \tag{32}$$

where $d_{\max} = \max_{i \in [n]} \sum_{j \in [n] \setminus i} E_{ij}$ is the maximum degree of the graph, and the last inequality follows from Lemma B.1. Now we will bound the second term of (31), i.e., $\text{var}(\mathbb{E}[T|\mathcal{Z}_1])$. Recall that by (27), we have $\mathbb{E}[T|\mathcal{Z}_1] = \sum_{(i,j) \in \mathcal{E}} (p_{ij} - F(\hat{w}_i - \hat{w}_j))^2$. Therefore, we upper bound $\text{var}(\mathbb{E}[T \mid \mathcal{Z}_1])$ as

$$
\begin{aligned}
\text{var}(\mathbb{E}[T \mid \mathcal{Z}_1]) &= \text{var}\left( \sum_{(i,j) \in \mathcal{E}} (p_{ij} - F(\hat{w}_i - \hat{w}_j))^2 \right) = \text{var}(\|P - \hat{\mathsf{F}}\|_{\mathrm{F}}^2) \\
&= \mathbb{E}[\|P - \hat{\mathsf{F}}\|_{\mathrm{F}}^4] - \mathbb{E}[\|P - \hat{\mathsf{F}}\|_{\mathrm{F}}^2]^2 \\
&\leq \mathbb{E}[(\|P - \mathsf{F}\|_{\mathrm{F}} + \|\mathsf{F} - \hat{\mathsf{F}}\|_{\mathrm{F}})^4] - \mathbb{E}[(\|P - \mathsf{F}\|_{\mathrm{F}} - \|\mathsf{F} - \hat{\mathsf{F}}\|_{\mathrm{F}})^2]^2,
\end{aligned}
$$

where the last inequality follows from the triangle inequality in the first term and the reverse triangle inequality on the second term. Under hypothesis $H_0$, the above expression simplifies trivially as

$$
\text{var}_{H_0}(\mathbb{E}[T \mid \mathcal{Z}_1]) \leq c_4 \left(\frac{n}{k}\right)^2, \tag{33}
$$

where $\text{var}_{H_l}(\cdot)$ denotes the variance under hypothesis $l$ for $l \in \{0, 1\}$. Now, we turn our attention to bounding $\text{var}_{H_1}(\mathbb{E}[T \mid \mathcal{Z}_1])$. This bound can be established through a relatively mechanical process described as follows

$$
\begin{aligned}
\text{var}_{H_1}(\mathbb{E}[T|\mathcal{Z}_1]) &\leq \|P - \mathsf{F}\|_{\mathrm{F}}^4 + 4\|P - \mathsf{F}\|_{\mathrm{F}}^3 \mathbb{E}[\|\mathsf{F} - \hat{\mathsf{F}}\|_{\mathrm{F}}] + 6\|P - \mathsf{F}\|_{\mathrm{F}}^2 \mathbb{E}[\|\mathsf{F} - \hat{\mathsf{F}}\|_{\mathrm{F}}^2] \\
&\quad + 4\|P - \mathsf{F}\|_{\mathrm{F}} \mathbb{E}[\|\mathsf{F} - \hat{\mathsf{F}}\|_{\mathrm{F}}^3] + \mathbb{E}[\|\mathsf{F} - \hat{\mathsf{F}}\|_{\mathrm{F}}^4] \\
&\quad - (\|P - \mathsf{F}\|_{\mathrm{F}}^2 + \mathbb{E}[\|\mathsf{F} - \hat{\mathsf{F}}\|_{\mathrm{F}}^2] - 2\|P - \mathsf{F}\|_{\mathrm{F}} \mathbb{E}[\|\mathsf{F} - \hat{\mathsf{F}}\|_{\mathrm{F}}])^2 \\
&= 8\|P - \mathsf{F}\|_{\mathrm{F}}^3 \mathbb{E}[\|\mathsf{F} - \hat{\mathsf{F}}\|_{\mathrm{F}}] + 4\|P - \mathsf{F}\|_{\mathrm{F}}^2 (\mathbb{E}[\|\mathsf{F} - \hat{\mathsf{F}}\|_{\mathrm{F}}^2] - \mathbb{E}[\|\mathsf{F} - \hat{\mathsf{F}}\|_{\mathrm{F}}]^2) \\
&\quad + 4\|P - \mathsf{F}\|_{\mathrm{F}} (\mathbb{E}[\|\mathsf{F} - \hat{\mathsf{F}}\|_{\mathrm{F}}^3] + \mathbb{E}[\|\mathsf{F} - \hat{\mathsf{F}}\|_{\mathrm{F}}^2] \mathbb{E}[\|\mathsf{F} - \hat{\mathsf{F}}\|_{\mathrm{F}}]) \\
&\quad + \mathbb{E}[\|\mathsf{F} - \hat{\mathsf{F}}\|_{\mathrm{F}}^4] - \mathbb{E}[\|\mathsf{F} - \hat{\mathsf{F}}\|_{\mathrm{F}}^2]^2 \\
&\leq 8c_1\|P - \mathsf{F}\|_{\mathrm{F}}^3 \sqrt{\frac{n}{k}} + 4c_2\|P - \mathsf{F}\|_{\mathrm{F}}^2 \frac{n}{k} \\
&\quad 4(c_3 + c_2 c_1)\|P - \mathsf{F}\|_{\mathrm{F}} \left(\frac{n}{k}\right)^{3/2} + c_4\left(\frac{n}{k}\right)^2
\end{aligned} \tag{34}
$$

Thus, by combining Equations (32) and (33) and (34) we obtain the following bounds on $\text{var}_{H_0}(T)$ and $\text{var}_{H_1}(T)$ based on (31)

$$
\text{var}_{H_0}(\mathbb{E}[T]) \leq \frac{nd_{\max}}{8k(k-1)} + c_1^2 \frac{n}{k^2} + c_4\left(\frac{n}{k}\right)^2 \tag{35}
$$

$$
\text{var}_{H_1}(\mathbb{E}[T]) \leq \frac{nd_{\max}}{8k(k-1)} + \frac{1}{k}\left(\|P - \mathsf{F}\|_{\mathrm{F}} + c_1\sqrt{\frac{n}{k}}\right)^2 + 8c_1\|P - \mathsf{F}\|_{\mathrm{F}}^3 \sqrt{\frac{n}{k}}
$$

$$
+ 4c_2\|P - \mathsf{F}\|_{\mathrm{F}}^2 \frac{n}{k} + 4\tilde{c}_3\|P - \mathsf{F}\|_{\mathrm{F}} \left(\frac{n}{k}\right)^{3/2} + c_4\left(\frac{n}{k}\right)^2. \tag{36}
$$

We define the decision rule as follows: select hypothesis $H_1$ if the test statistic $T$ exceeds the threshold:

$$
\text{Select } H_1 \text{ if} : T > \tilde{\gamma}\frac{n}{k} + c_2\frac{n}{k}, \tag{37}
$$

where $\tilde{\gamma}$ is a suitably chosen constant (selected below). Consequently, we can employ the one-sided Chebyshev's inequality

to bound the probability of error under hypothesis $H_0$, yielding:

$$
\begin{aligned}
\mathbb{P}_{H_0}\left(T > \tilde{\gamma}\frac{n}{k} + c_2\frac{n}{k}\right) &= \mathbb{P}_{H_0}\left(T - \mathbb{E}_{H_0}[T] > \tilde{\gamma}\frac{n}{k} + c_2\frac{n}{k} - \mathbb{E}_{H_0}[T]\right) \\
&\leq \mathbb{P}_{H_0}(T - \mathbb{E}_{H_0}[T] > \tilde{\gamma}\frac{n}{k}) \\
&\leq \frac{\mathrm{var}_{H_0}(T)}{\mathrm{var}_{H_0}(T) + \tilde{\gamma}^2(\frac{n}{k})^2} \\
&\leq \frac{\frac{nd_{\max}}{4k^2} + c_1^2\frac{n}{k^2} + c_4(\frac{n}{k})^2}{\frac{nd_{\max}}{4k^2} + c_1^2\frac{n}{k^2} + c_4(\frac{n}{k})^2 + \tilde{\gamma}^2(\frac{n}{k})^2} \\
&= \frac{\frac{d_{\max}}{4n} + c_1^2\frac{1}{n} + c_4}{\frac{d_{\max}}{4n} + c_1^2\frac{1}{n} + c_4 + \tilde{\gamma}^2} \leq \frac{1}{4},
\end{aligned}
$$

where the last bound holds by the fact that $d_{\max} \leq n$, and for an appropriate constant $\tilde{\gamma} \geq \max\{4\sqrt{c_4}, 4, 4c_1/\sqrt{n}\}$.

Now, we will find an upper bound on the probability of error under hypothesis $H_1$. Observe that an error is made under $H_1$ if the value of the test statistic $T \leq \tilde{\gamma}\frac{n}{k} + c_2\frac{n}{k}$.

$$
\begin{aligned}
\mathbb{P}_{H_1}\left(T \leq \tilde{\gamma}\frac{n}{k} + c_2\frac{n}{k}\right) &\\
&= \mathbb{P}_{H_1}\left(T - \mathbb{E}_{H_1}[T] \leq \tilde{\gamma}\frac{n}{k} + c_2\frac{n}{k} - \mathbb{E}_{H_1}[T]\right) \\
&\overset{\zeta_1}{\leq} \mathbb{P}\left(T - \mathbb{E}_{H_1}[T] \leq \tilde{\gamma}\frac{n}{k} + c_2\frac{n}{k} + 2c_1\sqrt{\frac{n}{k}}\|P - \mathsf{F}\|_{\mathrm{F}} - \|P - \mathsf{F}\|_{\mathrm{F}}^2\right) \\
&\overset{\zeta_2}{\leq} \frac{\mathrm{var}_{H_1}(T)}{\mathrm{var}_{H_1}(T) + (D^2 - \Delta)^2} \overset{\zeta_3}{\leq} \frac{1}{4},
\end{aligned}
$$

where $\zeta_1$ follows from (30), $\zeta_2$ follows by one-sided Chebyshev inequality and defining $D = \|P - \mathsf{F}\|_{\mathrm{F}}$ and $\Delta = \tilde{\gamma}\frac{n}{k} + c_2\frac{n}{k} + 2c_1\sqrt{\frac{n}{k}}D$. The step $\zeta_3$ holds if $4\mathrm{var}_{H_1}(T) \leq (D^2 - \Delta)^2$ or equivalently if $D^2 \geq 2\sqrt{\mathrm{var}_{H_1}(T)} + \Delta$. Using the sub-additivity of $\sqrt{\cdot}$ operator, the following condition necessitates that for this to be true:

$$
\begin{aligned}
D^2 \geq 2\Bigg(&\frac{\sqrt{nd_{\max}}}{2k} + \frac{1}{\sqrt{k}}\left(D + c_1\sqrt{\frac{n}{k}}\right) + 2\sqrt{2c_1}D^{3/2}\left(\frac{n}{k}\right)^{\frac{1}{4}} + 2\sqrt{c_2}D\sqrt{\frac{n}{k}} \\
&+ 2\sqrt{\tilde{c}_3}\sqrt{D}\left(\frac{n}{k}\right)^{\frac{3}{4}} + c_4\frac{n}{k}\Bigg) + \tilde{\gamma}\frac{n}{k} + c_2\frac{n}{k} + 2c_1\sqrt{\frac{n}{k}}D.
\end{aligned}
$$

Substituting $D = a_0\sqrt{\frac{n}{k}}$ in the above expression, for some $a_0$, we obtain:

$$
a_0^2 \geq \frac{\sqrt{d_{\max}}}{\sqrt{n}} + 2(a_0 + c_1)\sqrt{\frac{1}{n}} + 4\sqrt{2c_1}a_0^{3/2} + 4\sqrt{c_2}a_0 + 4\sqrt{2\tilde{c}_3}a_0 + 2c_4 + \tilde{\gamma} + c_2 + 2a_0 c_1.
$$

Now, we can conclude that the above expression is true for some large enough constant $a_0$ independent of $n$ and $k$, thus establishing that for $D = a_0\sqrt{\frac{n}{k}}$ and $a_0$ large enough our decision rule achieves a type I and type II sum error of at most $1/2$. Utilizing Theorem 3.1 we obtain that $\inf_{w \in \mathcal{W}_b}\|P - \mathsf{F}(w)\|_{\mathrm{F}} = \Theta(\|P - \mathsf{F}\|_{\mathrm{F}})$. Combining this fact along with the definition of critical threshold (cf. (14)), we have the following bound on $\varepsilon_{\mathsf{c}}$:

$$
\varepsilon_{\mathsf{c}} \leq O\left(\sqrt{\frac{1}{nk}}\right).
$$

This completes the proof. $\qquad\square$

Notably, from the above result, we have that $\varepsilon_{\mathsf{c}} \to 0$ as $n$ or $k$ goes to infinity. Therefore, for any fixed $n$ and $\epsilon > 0$, our decision rule is guaranteed to achieve a non-trivial minimax risk (strictly less than 1) for any pairwise comparison model $P$ in the class $\mathcal{M}_0$ or $\mathcal{M}_1(\epsilon)$ as long as the number of observed samples for each pair (i.e. $k$) are sufficiently large.

Moreover, there do exist pairwise comparison models $\{p_{ij} : (i,j) \in \mathcal{E}\}$ whose (normalized) separation is constant with $n$. Consequently, for such models, we can argue that for any fixed $k$ and $n$ large enough, our decision rule will achieve a non-trivial minimax risk. One such example of a pairwise comparison model represented by its pairwise comparison matrix (on a complete graph) is

$$P = \left(\frac{1}{2} + \eta\right)(\mathbf{1}\mathbf{1}^{\mathrm{T}} - I), \quad \text{for any } \eta \in \left(0, \frac{1}{2}\right).$$

It is easy to verify that for this comparison model, we must have $\inf_{w \in \mathcal{W}_b} \frac{1}{n}\|P - \mathsf{F}(w)\|_{\mathrm{F}} \geq \eta$. This is because any matrix $\mathsf{F}(w)$ must satisfy the constraint $(\mathsf{F})_{ij} + (\mathsf{F})_{ji} = 1$ for every $i \neq j$, which immediately leads to the lower bound of $\eta$ on the separation distance.

### B.1. Proof of Lemma B.1

Observe that by definition of $\mathsf{F}$ and $\hat{\mathsf{F}}$, we have

$$\sum_{(i,j) \in \mathcal{E}} (F(w_i^* - w_j^*) - F(\hat{w}_i - \hat{w}_j))^2 \leq \beta^2 \sum_{(i,j) \in \mathcal{E}} (|(w_i^* - w_j^*) - (\hat{w}_i - \hat{w}_j)|)^2$$
$$\leq 4\beta^2 \|\hat{w} - w^*\|_L^2. \tag{38}$$

Taking the power $p/2$ on both sides and then taking the expectation, we obtain

$$\mathbb{E}[\|\mathsf{F} - \hat{\mathsf{F}}\|_{\mathrm{F}}^p] \leq 2^p \beta^p \mathbb{E}[\|\hat{w} - w^*\|_L^p] \leq c_p \left(\frac{n}{k}\right)^{p/2},$$

where the last inequality follows by plugging in the bounds on $p$th moment from Theorem 3.3 and absorbing the constants $\alpha, \beta$ in $c_p$. The tail bound follows directly from (38). $\qquad\square$

## C. Proof of Theorem 3.4

Without loss of generality, we assume that the graph $\mathcal{G}$ is Eulerian. If not, we can reduce the problem to an Eulerian graph by considering the largest Eulerian spanning sub-graph $\tilde{\mathcal{G}}$ of $\mathcal{G}$ so that every vertex of $\tilde{\mathcal{G}}$ has even degree, which exists by our assumption that $\mathcal{G}$ is super-Eulerian.

Under the null hypothesis, we assume that the pairwise comparison model $P$ corresponds to equal utilities for all items, i.e., $p_{ij} = \frac{1}{2}, \forall (i,j) \in \mathcal{E}$ and let $\mathbb{P}_0$ denote the probability measure corresponding to this pairwise comparison model $P$. Under the alternative hypothesis, we will set our pairwise comparison model $R$ to be a perturbed version of $P$ (sharing the same observation graph $\mathcal{G}$). Specifically, every perturbation will have the following property:

$$\forall (i,j) \in \mathcal{E}, \ r_{ij} = \frac{1}{2} + \eta b_{ij}, \text{ where } \eta \in \left[0, \frac{1}{2} - \delta\right], \ b_{ij} \in \{-1, 1\}, \ b_{ij} + b_{ji} = 0, \ \forall i \in [n],$$
$$\text{and } \sum_{j:(i,j) \in \mathcal{E}} b_{ij} = 0, \forall i \in [n], \tag{39}$$

where $b_{ij}$ represents the signs of the perturbation by parameter $\eta$. Note that we set $b_{ij} = 0$ for $(i,j) \notin \mathcal{E}$. Let any such sequence of perturbations $b_{ij}$ is represented by a matrix $B \in \{-1, 0, 1\}^{n \times n}$.

As we delve deeper into the perturbation structure, we will carefully select a subset of perturbations $\mathcal{B}$ satisfying the constraints in (39), as well as additional constraints to be specified later

$$\mathcal{B} \subseteq \left\{b_{ij} \in \{-1, 1\} \text{ for } (i,j) \in \mathcal{E} : b_{ij} + b_{ji} = 0, \sum_{j:(i,j) \in \mathcal{E}} b_{ij} = 0, \forall i \in [n]\right\}. \tag{40}$$

Based on this selection, under the alternative hypothesis, let the pairwise comparison model $R$ be generated from a mixture distribution such that $R = P + \eta B$ and $B \sim \text{Unif}(\mathcal{B})$, i.e., $R$ is generated by adding the perturbation sequence selected uniformly at random from $\mathcal{B}$. Let $\mathbb{P}_{\mathcal{B}}$ denote the measure corresponding to the overall mixture distribution.

As we examine the perturbation structure, we make our first observation: for any perturbation $B$ satisfying (39), the corresponding pairwise comparison model $R$ belongs to the class $\mathcal{M}_1(\epsilon)$, for some $\epsilon$ as a function of $\eta$. Specifically, we will show that the perturbation $B$ guarantees a minimum separation distance of $\epsilon$ from the class of $\mathcal{M}_0$.

**Bounding separation:** Our first observation is that any such perturbation $R = P + \eta B$ has a sufficiently large (and more importantly, tractable) separation distance. In order to lower bound this separation distance, we will utilize Theorem 3.1. But first, we need to find the optimal $\mathcal{T}_F$ weights $w^*$ (as in (8)). This is addressed in the following lemma.

**Lemma C.1** (Optimal Weights for Perturbed Matrix). *For the $\mathcal{T}_F$ model and for any $B \in \mathcal{B}$ defined as in (40), the perturbed pairwise comparison matrix $P + \eta B$ has a unique optimal $\mathcal{T}_F$ weights given by $w^* = \mathbf{0}$ (all zeros vector).*

The proof is provided in Appendix C.2. We now utilize Theorem 3.1 to obtain the following lower bound on separation distance as

$$\inf_{w \in \mathcal{W}_b} \|P + \eta B - \mathsf{F}(w)\|_{\mathrm{F}} \geq c_1 \|P + \eta B - \mathsf{F}(\mathbf{0})\|_{\mathrm{F}}$$

$$= c_1 \eta \sqrt{|\mathcal{E}|}$$

where $c_1$ is the lower bound constant in Theorem 3.1 and the last equality follows sinde $(P)_{ij} = (\mathsf{F}(0))_{ij} = 1/2$ for all $(i, j) \in \mathcal{E}$. Therefore, we have $n\epsilon \geq c_1 \eta \sqrt{|\mathcal{E}|}$ by definition of $\mathcal{M}_1(\epsilon)$.

Having established a lower bound on the separation distance for each of the perturbations, our next step is to carefully select a special subset $\mathcal{B}$ of perturbations that allows us to approximate the "degrees of freedom" in the structure of our perturbation set, while also taking into account the constraints imposed by the graph topology.

To this end, we leverage the assumption that our observation graph $\mathcal{G}$ is Eulerian, meaning every node has an even degree. This property enables us to decompose $\mathcal{G}$ into a collection of edge-disjoint cycles, denoted by $\mathcal{C}$. In addition, we introduce a comparison incidence graph $\mathcal{G}_I$, which represents the comparison structure as an undirected bipartite graph. This graph has $n$ item nodes on one side and $|\mathcal{E}|/2$ nodes on the other side, each representing a pairwise comparison $(i, j) \in \mathcal{E}$ for $j > i$. The edges in $\mathcal{G}_I$ connect items to their respective comparison nodes. Since every node in $\mathcal{G}$ has an even degree, this ensures that the incidence graph $\mathcal{G}_I$ is Eulerian, and therefore $\mathcal{G}_I$ also has a cycle decomposition denoted by $\mathcal{C}_I$. Notably, each cycle in $\mathcal{G}_I$ is of even length and we can establish a one-to-one correspondence between the cycles in $\mathcal{C}$ and $\mathcal{C}_I$. Now, we orient the edges in the undirected comparison incidence graph $\mathcal{G}_I$ based on the values $b_{ij}$ in the perturbation $B$. Specifically, we will orient the edges in $\mathcal{G}_I$ as follows: if $b_{ij} = 1$, the edge will point from the item node to the comparison node for pair $(i, j)$, and if $b_{ij} = -1$, the edge will have the opposite direction. The constraints in (40) ensure that each node in $\mathcal{G}_I$ has equal in-degree and out-degree.

To specify the construction of $\mathcal{B}$, we consider any fixed cycle decomposition $\mathcal{C}_I$ (since it may not be unique). Let the number of cycles in the cycle decomposition be denoted by $|\mathcal{C}_I|$. Let $\sigma_i \in \mathcal{C}_I$ represents the $i$th cycle in $\mathcal{C}_I$ and $|\sigma_i|$ denotes the length of this $i$th cycle. Observe that we can independently orient the edges of any cycle $\sigma_i \in \mathcal{C}_I$ in either clockwise or counterclockwise direction, yielding $2^{|\mathcal{C}_I|}$ distinct Eulerian orientations for $\mathcal{G}_I$. We then construct the structured collection of perturbations $\mathcal{B}$ by associating each perturbation with one of the $2^{|\mathcal{C}_I|}$ distinct Eulerian orientations of the cycle decomposition $\mathcal{C}_I$. This establishes a one-to-one correspondence between valid perturbations in $\mathcal{B}$ and distinct Eulerian orientations of $\mathcal{C}_I$. Thus, in summary we define $\mathcal{B}$ correspoding to decomposition $\mathcal{C}_I$ as

$$\mathcal{B} \triangleq \left\{ b_{ij} \in \{-1, 1\} : b_{ij} + b_{ji} = 0, \forall(i, j) \in \mathcal{E}, \sum_{j:(i,j) \in \mathcal{E}} b_{ij} = 0, \forall i \in [n], \right.$$

$$\left. |b_l - b_{l+1}| = 2, \forall l \in \sigma_i, \forall \sigma_i \in \mathcal{C}_I \right\},$$

where, $l$ is used for indexing sequential edges of the cycle $\sigma_i$.

**Bounding risk:** Now, we will utilize the Ingster-Suslina method to compute lower bound on $\mathcal{R}(\mathcal{G}, \epsilon)$ (cf. (13)). The standard testing inequality by Le Cam (Ingster & Suslina, 2003) states that

$$\mathcal{R}(\mathcal{G}, \epsilon) \geq 1 - \|\mathbb{P}_0 - \mathbb{P}_\mathcal{B}\|_{\mathsf{TV}} \geq 1 - \sqrt{\chi^2(\mathbb{P}_\mathcal{B} \| \mathbb{P}_0)}. \tag{41}$$

We calculate the chi-squared divergence $\chi^2(\mathbb{P}_0 \| \mathbb{P}_\mathcal{B})$ by expressing it as an expectation with respect to two independent pairwise models corresponding to permutations $B$ and $B'$ drawn independently and uniformly at random from $\mathcal{B}$ as

$$\chi^2(\mathbb{P}_\mathcal{B} \| \mathbb{P}_0) = \mathbb{E}_{B, B' \sim \mathrm{Unif}(\mathcal{B})} \left[ \int \frac{d\mathbb{P}_B d\mathbb{P}_{B'}}{d\mathbb{P}_0} \right].$$

We will now leverage the tensorization property of $1 + \chi^2(P||Q)$, which enables us to decompose the chi-squared divergence between product distributions into a product of individual divergences. Specifically, for distributions $P_1, Q_1, \ldots, P_n, Q_n$, we have

$$1 + \chi^2\left(\prod_{i=1}^n P_i || \prod_{i=1}^n Q_i\right) = \prod_{i=1}^n (1 + \chi^2(P_i||Q_i)).$$

Consequently, the chi-squared divergence $\chi^2(\mathbb{P}_{\mathcal{B}}||\mathbb{P}_0)$ simplifies as

$$1 + \chi^2(\mathbb{P}_{\mathcal{B}}||\mathbb{P}_0) =$$

$$\mathbb{E}_{B,B'\sim\mathrm{Unif}(\mathcal{B})}\left[ \prod_{(i,j)\in\mathcal{E}} \left( \sum_{m=0}^k \frac{\binom{k}{m}(\frac{1}{2}+\eta b_{ij})^m (\frac{1}{2}-\eta b_{ij})^{k-m} \binom{k}{m}(\frac{1}{2}+\eta b'_{ij})^m (\frac{1}{2}-\eta b'_{ij})^{k-m}}{\binom{k}{m}\left(\frac{1}{2}\right)^k} \right)\right]$$

$$= \mathbb{E}_{B,B'\sim\mathrm{Unif}(\mathcal{B})}\left[ \prod_{(i,j)\in\mathcal{E}} \left( \sum_{m=0}^k \frac{\binom{k}{m}(\frac{1}{2}+\eta b_{ij})^m (\frac{1}{2}-\eta b_{ij})^{k-m} (\frac{1}{2}+\eta b'_{ij})^m (\frac{1}{2}-\eta b'_{ij})^{k-m}}{\left(\frac{1}{2}\right)^k} \right)\right]. \tag{42}$$

We direct our attention to the $(i,j)$th term of the product in (42), for $(i,j)\in\mathcal{E}$ and denote it as $h(b_{ij}, b'_{ij})$

$$h(b_{ij}, b'_{ij}) = \sum_{m=0}^k \frac{\binom{k}{m}(\frac{1}{2}+\eta b_{ij})^m (\frac{1}{2}-\eta b_{ij})^{k-m} (\frac{1}{2}+\eta b'_{ij})^m (\frac{1}{2}-\eta b'_{ij})^{k-m}}{\left(\frac{1}{2}\right)^k}. \tag{43}$$

Now since we have $b_{ij}, b'_{ij} \in \{-1, 1\}$, therefore whenever $b_{ij}$ and $b'_{ij}$ agree, by (43) we have $h(1,1) = h(-1,-1)$. And moreover, we can calculate $h(1,1)$ as

$$h(1,1) = 2^k \sum_{m=0}^k \binom{k}{m} \left(\frac{1}{2}+\eta\right)^{2m} \left(\frac{1}{2}-\eta\right)^{2k-2m}$$

$$= 2^k \sum_{m=0}^k \binom{k}{m} \left(\frac{1}{4}+\eta^2+\eta\right)^m \left(\frac{1}{4}+\eta^2-\eta\right)^{k-m}$$

$$= (1+4\eta^2)^k \sum_{m=0}^k \binom{k}{m} \left(\frac{1}{2}+\frac{\eta}{\frac{1}{2}+2\eta^2}\right)^m \left(\frac{1}{2}-\frac{\eta}{\frac{1}{2}+2\eta^2}\right)^{k-m}$$

$$= (1+4\eta^2)^k.$$

Additionally, by (43) we also have $h(1,-1) = h(-1,1)$ and it simplifies to

$$h(1,-1) = 2^k \sum_{m=0}^k \binom{k}{m} \left(\frac{1}{2}+\eta\right)^{2k} \left(\frac{1}{2}-\eta\right)^{2k} = (1-4\eta^2)^k.$$

For any two perturbations $B, B' \sim \mathrm{Unif}(\mathcal{B})$, let random variables $A_1$ denotes the number of agreements between $B, B'$ respectively, i.e., number of $(i,j)\in\mathcal{E}_+$ where $b_{ij} = b'_{ij}$ in randomly drawn permutation $B$ and $B'$. And similarly let $A_2$ denotes the number of disagreements between $B, B'$ i.e., number of $(i,j)\in\mathcal{E}_+$ where $b_{ij} = -b'_{ij}$. Consequently, we obtain

$$1 + \chi^2(\mathbb{P}_{\mathcal{B}}||\mathbb{P}_0) = \mathbb{E}_{B,B'\sim\mathrm{Unif}(\mathcal{B})}\left[ h(1,1)^{2A_1} h(1,-1)^{2A_2}\right]$$

$$= \mathbb{E}_{B,B'\sim\mathrm{Unif}(\mathcal{B})}\left[ (1+4\eta^2)^{2kA_1} (1-4\eta^2)^{2kA_2}\right]$$

$$\leq \mathbb{E}_{B,B'\sim\mathrm{Unif}(\mathcal{B})}\left[ \exp(8\eta^2 k(A_1 - A_2))\right]. \tag{44}$$

In addition, we define vectors $a, a' \in \{-1, 1\}^{|\mathcal{C}_I|}$ to represent the orientations of the $|\mathcal{C}_I|$ cycles in $\mathcal{G}_I$ induced by $B$. The subsequent calculation will now be used to complete the proof:

$$
\chi^2(\mathbb{P}_\mathcal{B}||\mathbb{P}_0) + 1 \overset{\zeta_1}{\le} \frac{1}{2^{2|\mathcal{C}_I|}} \sum_{B,B'} \exp\big(8\eta^2 k (A_1 - A_2)\big) \overset{\zeta_2}{=} \frac{1}{2^{2|\mathcal{C}_I|}} \sum_{a,a'} \exp\left(8\eta^2 k \sum_{\sigma_i \in \mathcal{C}_I} |\sigma_i| a_i a_i'\right)
$$

$$
= \mathbb{E}\left[\prod_{\sigma_i \in \mathcal{C}_I} \exp\big(8\eta^2 k |\sigma_i| a_i a_i'\big)\right] \overset{\zeta_3}{=} \prod_{\sigma_i \in \mathcal{C}_I} \mathbb{E}\big[\exp\big(8\eta^2 k |\sigma_i| a_i a_i'\big)\big]
$$

$$
= \prod_{\sigma_i \in \mathcal{C}_I} \left(\frac{1}{2} \exp\big(8\eta^2 k |\sigma_i|\big) + \frac{1}{2} \exp\big(-8\eta^2 k |\sigma_i|\big)\right)
$$

$$
\le \prod_{\sigma_i \in \mathcal{C}_I} \left(\exp\big(32\eta^4 k^2 (|\sigma_i|)^2\big)\right) = \exp\left(32\eta^4 k^2 \sum_{\sigma_i \in \mathcal{C}_I} |\sigma_i|^2\right),
$$

where $\zeta_1$ follows from (44) and the fact that there are $2^{|\mathcal{C}_I|}$ perturbations which are sampled uniformly from $\mathcal{B}$, $\zeta_2$ follows from definition of $a_i$ and the fact that number of agreements/disagreements can be represented in terms of the agreements/disagreements of the cycle orientations $a_i, a_i'$. $\zeta_3$ follows from the fact that the orientations of the cycles are independent of one another. Finally, utilizing the fact that $c_1 \eta \sqrt{|\mathcal{E}|} \le n\epsilon$ and by combining the resulting bound along with (41) and the fact that cycle lengths in $\mathcal{G}_I$ are twice the size in $\mathcal{G}$ completes the proof. $\square$

## C.1. Proof of Proposition 3.5

**Part 1:** For a complete graph, the comparison incidence graph $\mathcal{G}_I$ has $n$ item nodes and $\frac{n(n-1)}{2}$ comparison nodes. When $n$ is odd, all nodes have an even degree equal to $n-1$; therefore, $\mathcal{G}$ is Eulerian. Notably, $n$ can take the forms $n = 6m + 1$, $n = 6m + 3$, and $n = 6m + 5$, where $m \in \mathbb{N}$. As established by (Kirkman, 1847), for $n = 6m + 1$ and $n = 6m + 3$, $\mathcal{G}$ can always be decomposed into cycles of length 3. Meanwhile, for $n = 6m + 5$, $\mathcal{G}$ can be decomposed into a cycle of length 4 and remaining cycles of length 3 (Feder & Subi, 2012). Therefore, we have $|\mathcal{E}|^2 = O(n^4)$ and $\sum_{\sigma \in \mathcal{C}} |\sigma|^2 = O(n^2)$, giving $\varepsilon_\mathsf{c}^2 = \Omega(1/nk)$

**Part 2:** Consider a $d$-regular graph with constant even degree $d$. The associated comparison incidence graph has $n$ item nodes and $nd/2$ comparison nodes. By applying (Seshadri & Ugander, 2019, Lemma 9), we can decompose the edge set of the comparison incidence graph into cycles of size at most $\lfloor 2\log_2(n)\rfloor$, with at most $\min\{2n + nd, 4n\} = 4n$ edges remaining. Since the graph is Eulerian, removing cycles does not affect this property. Therefore, the remaining $4n$ edges can be further decomposed into cycles of length at most $2n$ (since cycles can have a maximum length of $2n$, and this reflects a worst-case scenario). This yields $\sum_{\sigma \in \mathcal{C}} |\sigma|^2 = O(n^2)$, which in turn implies $\varepsilon_\mathsf{c}^2 = \Omega(1/n^2 k)$.

**Part 3:** For graphs comprising a single cycle, it is easy to verify that the number of cycles is 1 and the cycle has a length of $n$.

**Part 4:** For a toroidal grid of size $\sqrt{n} \times \sqrt{n}$, we can generate a cycle decompostion of $\mathcal{G}$ consisting of $\sqrt{n}$ horizontal edges and $\sqrt{n}$ vertical edges. Clearly, each of these edges has a length of $\sqrt{n}$. Therefore, $\sum_{\sigma_i \in \mathcal{C}} |\sigma_i|^2 = 2n\sqrt{n}$. And since it is a toroidal grid we have $|\mathcal{E}| = 2n$. Plugging in these values we obtain $\varepsilon_\mathsf{c}^2 = \Omega(1/n^{7/4}k)$. $\square$

## C.2. Proof of Lemma C.1

To find the optimal weights $w^*$, our objective is to solve the following optimization problem with parameter $b$:

$$
l^*(w) = \min_{w \in \mathcal{W}_b} - \sum_{(i,j) \in \mathcal{E}} r_{ij} \log(F(w_i - w_j)) + (1 - r_{ij}) \log(1 - F(w_i - w_j)).
$$

Our initial observation is that, due to the skew-symmetrization of the model $r_{ij} + r_{ji} = 1$, we can express the gradient of $l^*(w)$ as:

$$
(\nabla l^*(w))_i = -2 \sum_{j:(i,j) \in \mathcal{E}} (r_{ij} - F(w_i - w_j)) \times \frac{F'(w_i - w_j)}{F(w_i - w_j)(1 - F(w_i - w_j))}.
$$

Furthermore, the gradient is zero at $w = \mathbf{0}$. To see this, note that for all $i \in [n]$, we have:

$$(\nabla l^*(w))_i|_{w=\mathbf{0}} = -2 \sum_{j:(i,j)\in\mathcal{E}} (\frac{1}{2} + \eta b_{ij} - F(0)) \times \frac{F'(0)}{F(0)(1 - F(0))} = -8\eta F'(0) \sum_{j:(i,j)\in\mathcal{E}} b_{ij} = 0,$$

where the last step is followed by our construction of the perturbation sequence in (39). Considering that the gradient is zero at $w = \mathbf{0}$ and the optimization objective is convex over $\mathcal{W}_b$ (in fact strongly convex over $\mathcal{W}_b$), coupled with the uniqueness of the optimal $\mathcal{T}_F$ weights as indicated by Proposition 2.4, we conclude that $w^* = \mathbf{0}$ is indeed the optimal and unique solution for any $b \geq 0$. $\qquad\square$

## D. Proofs of Time-Uniform Bounds on Probabilities of Type I and II Errors

In this appendix, we will establish bounds on type I and II error probabilities as described in Theorem 3.7. First, we will introduce essential notation to facilitate our analysis and present our proof of Proposition 3.6. Then, we will establish a few auxiliary lemmata such as which are needed to derive the bounds on type I and type II errors. Finally, combining these results, we will proof of Theorem 3.7 in Appendix D.3 and a few corollaries based on these results in Appendix E.

**Additional notation:** We introduce $Y_{ij}^l$ for $l \in \mathbb{N}$ and $(i,j) \in \mathcal{E}$ to denote the observed comparisons that are used for estimating $Z_{ij} = \sum_{l=1}^{k} Z_{ij}^{k_1+l}$, i.e., we let $Y_{ij}^l = Z_{ij}^{k_1+l}$ for $l \in [k]$. Also, define the statistic $\bar{Y}_{ij}^k \triangleq \sum_{m=1}^{k} Y_{ij}^m$. Moreover, define $\mathbf{1}_n$ as an all-ones vector of length $n$ and $I_n$ as the identity matrix of size $n \times n$.

### D.1. Proof of Proposition 3.6

We will focus on the following sequence $\{\tilde{T}_{ij}^k, k \in \mathbb{N} \setminus \{1\}\}$ defined as

$$\tilde{T}_{ij}^k \triangleq \frac{\bar{Y}_{ij}^k(\bar{Y}_{ij}^k - 1)}{k(k-1)} + b_{ij}^2 - 2b_{ij}\frac{\bar{Y}_{ij}^k}{k}.$$

Note that with $b_{ij} = F(\hat{w}_i - \hat{w}_j)$, the term $\tilde{T}_{ij}^k$ reduces to the $(i,j)$th term of the test statistic $T^{k_1,k}$ (based on the notation defined above) and we will now show that it is indeed a reverse martingale. To do this, we will demonstrate that both the terms $\frac{\bar{Y}_{ij}^k(\bar{Y}_{ij}^k-1)}{k(k-1)}$ and $\frac{\bar{Y}_{ij}^k}{k}$ are indeed reverse martingales. First, we focus on the former term. Observe that we can write the product $\bar{Y}_{ij}^k(\bar{Y}_{ij}^k - 1)$ as

$$\frac{\bar{Y}_{ij}^k(\bar{Y}_{ij}^k - 1)}{k(k-1)} = \frac{(\sum_{m=1}^{k} Y_{ij}^m)^2 - \sum_{m=1}^{k} Y_{ij}^m}{k(k-1)} = \frac{1}{k(k-1)} \sum_{l,m=1:l\neq m}^{k} Y_{ij}^l Y_{ij}^m,$$

where the last equality follows because $\sum_{m=1}^{k} Y_{ij}^m(Y_{ij}^m - 1) = 0$ as $Y_{ij}^m \in \{0,1\}$. Also, observe that $\mathbb{E}[Y_{ij}^m Y_{ij}^l \mid \mathcal{F}_{k+1}] = \mathbb{E}[Y_{ij}^m Y_{ij}^r \mid \mathcal{F}_{k+1}]$ for $l \neq m \neq r$ and where $\mathcal{F}_k$ is the canonical reverse filtration defined as the sigma algebra generated by $\mathcal{F}_k = \bigotimes_{(i,j)\in\mathcal{E}} \sigma\left(\frac{\bar{Y}_{ij}^k}{k}, Y_{ij}^{k+1}, Y_{ij}^{k+2}, \dots\right)$. This is because for any set $\mathcal{A} \in \mathcal{F}_{k+1}$ and $l, m, r \in [k]$ and $l \neq m \neq r$, we have

$$\mathbb{E}[Y_{ij}^m Y_{ij}^l \mathbb{1}_{\mathcal{A}}] = \mathbb{E}[Y_{ij}^m Y_{ij}^r \mathbb{1}_{\mathcal{A}}] \text{ and } \mathbb{E}[Y_{ij}^m \mathbb{1}_{\mathcal{A}}] = \mathbb{E}[Y_{ij}^l \mathbb{1}_{\mathcal{A}}].$$

Utilizing the above relations, we can show that $\frac{\bar{Y}_{ij}^k(\bar{Y}_{ij}^k-1)}{k(k-1)}$ is indeed a reverse-martingale as:

$$\mathbb{E}\left[\frac{\bar{Y}_{ij}^k(\bar{Y}_{ij}^k - 1)}{k(k-1)} \mid \mathcal{F}_{k+1}\right] = \mathbb{E}\left[\frac{\sum_{l,m=1:l\neq m}^{k} Y_{ij}^l Y_{ij}^m}{k(k-1)} \mid \mathcal{F}_{k+1}\right] = \frac{\sum_{l,m=1:l\neq m}^{k+1} \mathbb{E}[Y_{ij}^l Y_{ij}^m \mid \mathcal{F}_{k+1}]}{k(k-1)}$$

$$= \mathbb{E}\left[\frac{\sum_{l,m=1:l\neq m}^{k+1} Y_{ij}^l Y_{ij}^m}{k(k+1)} \mid \mathcal{F}_{k+1}\right] = \mathbb{E}\left[\frac{\bar{Y}_{ij}^{k+1}(\bar{Y}_{ij}^{k+1} - 1)}{k(k+1)} \mid \mathcal{F}_{k+1}\right]$$

$$= \frac{\bar{Y}_{ij}^{k+1}(\bar{Y}_{ij}^{k+1} - 1)}{k(k+1)},$$

where the last equality follows since $\bar{Y}_{ij}^{k+1}$ is measurable with respect to $\mathcal{F}_{k+1}$. Similarly, we can also show that $\frac{\bar{Y}_{ij}^k}{k}$ is also a reverse martingale as:

$$
\mathbb{E}\left[\frac{\bar{Y}_{ij}^k}{k}|\mathcal{F}_{k+1}\right] = \frac{\sum_{m=1}^k \mathbb{E}\left[Y_{ij}^m \mid \mathcal{F}_{k+1}\right]}{k} = \frac{\sum_{m=1}^k \mathbb{E}\left[Y_{ij}^m \mid \mathcal{F}_{k+1}\right]}{k+1}
$$

$$
= \mathbb{E}\left[\frac{\bar{Y}_{ij}^{k+1}}{k+1}|\mathcal{F}_{k+1}\right] = \frac{\bar{Y}_{ij}^{k+1}}{k}.
$$

Finally, the proposition follows by the linearity of conditional expectation and substituting $b_{ij} = F(\hat{w}_i - \hat{w}_j)$ as:

$$
\mathbb{E}\left[T^{k_1,k} \mid \mathcal{F}_{k+1}\right] = \mathbb{E}\left[\sum_{(i,j)\in\mathcal{E}} \tilde{T}_{ij}^k \mid \mathcal{F}_{k+1}\right]
$$

$$
= \sum_{(i,j)\in\mathcal{E}} \mathbb{E}\left[\frac{\bar{Y}_{ij}^k(\bar{Y}_{ij}^k - 1)}{k(k-1)} \mid \mathcal{F}_{k+1}\right] + b_{ij}^2 - 2b_{ij}\mathbb{E}\left[\frac{\bar{Y}_{ij}^k}{k} \mid \mathcal{F}_{k+1}\right]
$$

$$
= \sum_{(i,j)\in\mathcal{E}} \frac{\bar{Y}_{ij}^{k+1}(\bar{Y}_{ij}^{k+1} - 1)}{k(k+1)} + b_{ij}^2 - 2b_{ij}\frac{\bar{Y}_{ij}^{k+1}}{k+1} = \sum_{(i,j)\in\mathcal{E}} \tilde{T}_{ij}^{k+1} = T^{k_1,k+1}.
$$

This completes the proof. $\qquad\square$

### D.2. Intermediate Lemmata

In order to derive the proof of Theorem 3.7, we will first prove the following intermediate lemma that gives bounds on type I and type II errors where the threshold is a function of the estimated weights $\hat{w}$. Additionally, the lemma relies on a variant of the Hanson-Wright inequality that is time-uniform (see Lemma D.2) and specialized to our specific setting.

**Lemma D.1** (Conditional Bounds on Type I and II Error Probabilities). *For any $\alpha \in (0, 1]$ and for $\hat{w}$ estimated from the first $k_1$ pairwise comparison for each pair in $\mathcal{E}$, there exist a constant $c$ such that for $\nu \in (0, 1/e)$, the following bounds hold under hypothesis $H_0$ and $H_1$, respectively:*

$$
\mathbb{P}_{H_0}\left(\exists k \geq 2, T^{k_1,k} \geq \|\mathsf{F} - \hat{\mathsf{F}}\|_{\mathrm{F}}^2 + c\frac{\sqrt{|\mathcal{E}|}}{k}\ell_{k,\nu} + 4\frac{\|\mathsf{F} - \hat{\mathsf{F}}\|_{\mathrm{F}}}{\sqrt{k}}\sqrt{\ell_{k,\nu}}\right) \leq \nu,
$$

$$
\mathbb{P}_{H_1}\left(\exists k \geq 2, T^{k_1,k} - \|\mathsf{P} - \mathsf{F}\|_{\mathrm{F}}^2 \geq \|\mathsf{F} - \hat{\mathsf{F}}\|_{\mathrm{F}}^2 - 2\|\mathsf{F} - \hat{\mathsf{F}}\|_{\mathrm{F}}\|\mathsf{P} - \mathsf{F}\|_{\mathrm{F}} - \right.
$$

$$
\left. - c\frac{\sqrt{|\mathcal{E}|}}{k}\ell_{k,\nu} - 4\frac{\|\mathsf{P} - \mathsf{F}\|_{\mathrm{F}} + \|\mathsf{F} - \hat{\mathsf{F}}\|_{\mathrm{F}}}{\sqrt{k}}\sqrt{\ell_{k,\nu}}\right) \leq \nu,
$$

*where $\ell_{k,\nu} = \log\left(3.5\log_2(k)^2/\nu\right)$.*

**Proof:**
**Part 1:** We will first prove the bound under hypothesis $H_0$. Based on the additional notation defined at the beginning of the Appendix, we can simplify the $(i, j)$th term $T_{ij}^{k_1,k}$ of the test statistic $T^{k_1,k}$ as:

$$
T_{ij}^{k_1,k} = \frac{\bar{Y}_{ij}^k(\bar{Y}_{ij}^k - 1)}{k(k-1)} + F(\hat{w}_i - \hat{w}_j)^2 - 2F(\hat{w}_i - \hat{w}_j)\frac{\bar{Y}_{ij}^k}{k}
$$

$$
= \frac{\sum_{l,m=1:l\neq m}^k Y_{ij}^m Y_{ij}^l}{k(k-1)} + F(\hat{w}_i - \hat{w}_j)^2 - 2\frac{F(\hat{w}_i - \hat{w}_j)_{l=1}^m Y_{ij}^m}{k}
$$

$$
\overset{\zeta_1}{=} (y_{ij}^k)^{\mathrm{T}} A^{(k)} y_{ij}^k + \mathbf{1}_k^{\mathrm{T}} A^{(k)} \mathbf{1}_k F(w_i^* - w_j^*)^2 - 2F(w_i^* - w_j^*)\mathbf{1}_k^{\mathrm{T}} A^{(k)} y_{ij}^k +
$$

$$
(F(\hat{w}_i - \hat{w}_j) - F(w_i^* - w_j^*))^2 - 2(F(\hat{w}_i - \hat{w}_j) - F(w_i^* - w_j^*))\left(\frac{\bar{Y}_{ij}^k}{k} - F(w_i^* - w_j^*)\right)
$$

$$\stackrel{\zeta_2}{=} \underbrace{(y_{ij}^k - F(w_i^* - w_j^*)\mathbf{1}_k)^{\mathrm{T}} A^{(k)}(y_{ij}^k - F(w_i^* - w_j^*)\mathbf{1}_k)}_{I_{ij}^{1,k}} + (F(\hat{w}_i - \hat{w}_j) - F(w_i^* - w_j^*))^2$$

$$\underbrace{- 2(F(\hat{w}_i - \hat{w}_j) - F(w_i^* - w_j^*))\left(\frac{\bar{Y}_{ij}^k}{k} - F(w_i^* - w_j^*)\right)}_{I_{ij}^{2,k}},$$

where in $\zeta_1$ we have $A^{(k)} \triangleq \frac{\mathbf{1}_k\mathbf{1}_k^{\mathrm{T}} - I_k}{k(k-1)}$ and $y_{ij}^k \in \mathbb{R}^k$ is a vector such that $y_{ij}^k = [Y_{ij}^1, \ldots, Y_{ij}^k]$ and in $\zeta_2$ the term $I_{ij}^{1,k}$ follows by observing that

$$\begin{aligned}
\left(y_{ij}^k - F(w_i^* - w_j^*)\mathbf{1}_k\right)^{\mathrm{T}} A^{(k)}\left(y_{ij}^k - F(w_i^* - w_j^*)\mathbf{1}_k\right) = \\
(y_{ij}^k)^{\mathrm{T}} A^{(k)} y_{ij}^k + \mathbf{1}_k^{\mathrm{T}} A^{(k)} \mathbf{1}_k F(w_i^* - w_j^*)^2 - 2p_{ij}\mathbf{1}_k^{\mathrm{T}} A^{(k)} y_{ij}^k. \quad (45)
\end{aligned}$$

Now, we will upper bound the quadratic variation term $\sum_{(i,j)\in\mathcal{E}} I_{ij}^{1,k}$. For this we will utilize Lemma D.2 to obtain the tail bounds for any $\nu \in (0, 1/e)$ to obtain (for some constant $c$):

$$\mathbb{P}\left(\exists k \geq 2 : \sum_{(i,j)\in\mathcal{E}} I_{ij}^{1,k} > c\frac{\sqrt{|\mathcal{E}|}}{k}\ell_{k,\nu}\right) \leq \frac{\nu}{2}.$$

It is straightforward to show that $\sum_{(i,j)\in\mathcal{E}} I_{ij}^{2,k}$ is $(4\|\mathsf{F} - \hat{\mathsf{F}}\|_{\mathrm{F}}^2/k)$-sub-gaussian. Therefore, by utilizing a time-uniform version of Hoeffding inequality (Manole & Ramdas, 2023, Corollary 8) for the user-defined $h(k) = (\pi k)^2/6$ (also used for the stitching argument in the proof of Lemma D.2), we obtain for any $\nu \in (0, 1)$

$$\mathbb{P}\left(\exists k \geq 2 : \sum_{(i,j)\in\mathcal{E}} I_{ij}^{2,k} > 2\|\mathsf{F} - \hat{\mathsf{F}}\|_{\mathrm{F}}\sqrt{2\frac{\log(h(\log_2(k))) + \log(2/\nu)}{k/2}}\right) \leq \frac{\nu}{2}.$$

Combining the two tail bounds and a simple calculation completes the proof for type I error.

**Part 2:** Observe that under hypothesis $H_1$ we have

$$\begin{aligned}
T_{ij}^k - (p_{ij} - F(w_i^* - w_j^*))^2 &= \frac{\bar{Y}_{ij}^k(\bar{Y}_{ij}^k - 1)}{k(k-1)} - p_{ij}^2 + F(\hat{w}_i - \hat{w}_j)^2 - F(w_i^* - w_j^*)^2 \\
&\quad + 2\left(F(w_i^* - w_j^*)p_{ij} - F(\hat{w}_i - \hat{w}_j)\frac{\bar{Y}_{ij}^k}{k}\right) \\
&\stackrel{\zeta_1}{=} (y_{ij}^k)^{\mathrm{T}} A^{(k)} y_{ij}^k - \mathbf{1}_k^{\mathrm{T}} A^{(k)} \mathbf{1}_k p_{ij}^2 + F(\hat{w}_i - \hat{w}_j)^2 - F(w_i^* - w_j^*)^2 \\
&\quad + 2p_{ij}(F(w_i^* - w_j^*) - F(\hat{w}_i - \hat{w}_j)) + 2\left(p_{ij} - \frac{\bar{Y}_{ij}^k}{k}\right)F(\hat{w}_i - \hat{w}_j), \quad (46)
\end{aligned}$$

where in $\zeta_1$ we have $A^k \triangleq \frac{\mathbf{1}_k\mathbf{1}_k^{\mathrm{T}} - I_k}{k(k-1)}$ and $y_{ij}^k \in \mathbb{R}^k$ is a vector $y_{ij}^k = [y'_{ij}, \ldots, y_{ij}^k]$ as before. Now, observe that the term

$$\begin{aligned}
(y_{ij}^k)^{\mathrm{T}} A^{(k)} y_{ij}^k - \mathbf{1}_k^{\mathrm{T}} A^{(k)} \mathbf{1}_k p_{ij}^2 &= \left(y_{ij}^k - p_{ij}\mathbf{1}_k\right)^{\mathrm{T}} A^{(k)}\left(y_{ij}^k - p_{ij}\mathbf{1}_k\right) \\
&\quad + 2p_{ij}\left(y_{ij}^k - p_{ij}\mathbf{1}_k\right)^{\mathrm{T}} A^{(k)} \mathbf{1}_k \\
&= \left(y_{ij}^k - p_{ij}\mathbf{1}_k\right)^{\mathrm{T}} A^{(k)}\left(y_{ij}^k - p_{ij}\mathbf{1}_k\right) + 2p_{ij}\left(\frac{\bar{Y}_{ij}^k}{k} - p_{ij}\right).
\end{aligned}$$

Substituting the above bound in (46), we obtain

$$T_{ij}^{k_1,k} - (p_{ij} - F(w_i^* - w_j^*))^2 = \underbrace{\left(y_{ij}^k - p_{ij}\mathbf{1}_k\right)^{\mathrm{T}} A^{(k)} \left(y_{ij}^k - p_{ij}\mathbf{1}_k\right)}_{I_{ij}^{3,k}}$$

$$+ \underbrace{2(p_{ij} - F(\hat{w}_i - \hat{w}_j))\left(\frac{\bar{Y}_{ij}^k}{k} - p_{ij}\right)}_{I_{ij}^{4,k}}$$

$$+ \underbrace{(F(\hat{w}_i - \hat{w}_j) - F(w_i^* - w_j^*))(F(\hat{w}_i - \hat{w}_j) + F(w_i^* - w_j^*) - 2p_{ij})}_{I_{ij}^{5,k}}.$$

Now the term $\sum_{(i,j)\in\mathcal{E}} I_{ij}^{3,k}$ is bounded by utilizing Lemma D.2 to obtain the tail bounds for some constant $c$

$$\mathbb{P}\left(\exists k \geq 2 : \sum_{(i,j)\in\mathcal{E}} I_{ij}^{3,k} < -c\frac{\sqrt{|\mathcal{E}|}}{k}\ell_{k,\nu}\right) \leq \frac{\nu}{2}.$$

And the term $\sum_{(i,j)\in\mathcal{E}} I_{ij}^{4,k}$ is bounded by utilizing adaptive Hoeffding's inequality (Manole & Ramdas, 2023, Corollary 8) to obtain the tail bounds

$$\mathbb{P}\left(\exists k \geq 2 : \sum_{(i,j)\in\mathcal{E}} I_{ij}^{4,k} < -4\|P - \hat{\mathsf{F}}\|_{\mathrm{F}}\sqrt{\frac{\ell_{k,\nu}}{k}}\right) \leq \frac{\nu}{2}.$$

An application of the triangle inequality to the above equation yields

$$\mathbb{P}\left(\exists k \geq 2 : \sum_{(i,j)\in\mathcal{E}} I_{ij}^{4,k} < -4(\|P - \mathsf{F}\|_{\mathrm{F}} + \|\mathsf{F} - \hat{\mathsf{F}}\|_{\mathrm{F}})\sqrt{\frac{\ell_{k,\nu}}{k}}\right) \leq \frac{\nu}{2}.$$

Finally, the term $\sum_{(i,j)\in\mathcal{E}} I_{ij}^{5,k}$ is bounded using Cauchy-Schwarz inequality as:

$$\sum_{(i,j)\in\mathcal{E}} I_{ij}^{5,k} = \sum_{(i,j)\in\mathcal{E}} (F(\hat{w}_i - \hat{w}_j) - F(w_i^* - w_j^*))^2$$

$$+ \sum_{(i,j)\in\mathcal{E}} 2(F(\hat{w}_i - \hat{w}_j) - F(w_i^* - w_j^*))(F(w_i^* - w_j^*) - p_{ij})$$

$$\geq \|\mathsf{F} - \hat{\mathsf{F}}\|_{\mathrm{F}}^2 - 2\|\mathsf{F} - \hat{\mathsf{F}}\|_{\mathrm{F}}\|\mathsf{F} - P\|_{\mathrm{F}}.$$

Combining the three bounds for $I_{ij}^{3,k}, I_{ij}^{4,k}, I_{ij}^{5,k}$ completes the proof for type II error. $\qquad\square$

**Lemma D.2** (Time-Uniform Quadratic Bound). *For any element $v$ in a finite set $\mathcal{V}$, consider a sequence of independent random variables $x_v^{(1)}, x_v^{(2)}, x_v^{(3)}, \ldots$ such that $x_v^{(l)} \sim \mathsf{Bernoulli}(p_v)$ for $l \in \mathbb{N}$ and some $p_v \in (0,1)$. Define the random vector $\bar{x}_v^{(k)} = \left(x_v^{(1)}, x_v^{(2)}, \ldots, x_v^{(k)}\right) \in \mathbb{R}^k$ and let $\{A^{(k)} \in \mathbb{R}^{k\times k} : k \in \mathbb{N}\}$ be a sequence of matrices such that $A^{(k)} = (\mathbf{1}_k\mathbf{1}_k^{\mathrm{T}} - I_k)/(k(k-1))$. Then, there exists a constant $c > 0$ such that for all $\nu \in (0, 1/e)$, we have*

$$\mathbb{P}\left(\exists k \geq 2, \sum_{v\in\mathcal{V}} (\bar{x}_v^{(k)} - p_v\mathbf{1}_k)^{\mathrm{T}} A^{(k)}(\bar{x}_v^{(k)} - p_v\mathbf{1}_k) > c\frac{\sqrt{|\mathcal{V}|}}{k}\log\left(3.5\log_2^2(k)/\nu\right)\right) \leq \nu.$$

**Proof:** Denote $s_v^{(k)} = (\bar{x}_v^{(k)} - p_v\mathbf{1}_k)^{\mathrm{T}} A^{(k)}(\bar{x}_v^{(k)} - p_v\mathbf{1}_k)$. Recall that we had shown that $\{s_v^{(k)} : k \in \mathbb{N} \setminus \{1\}\}$ forms a reverse-martingale with respect to canonical reverse filtration $\{\sigma\left(\sum_{m=1}^k x_v^{(m)}, x_v^{(k+1)}, x_v^{(k+2)}, \ldots\right) : k \in \mathbb{N} \setminus \{1\}\}$, i.e., for $k \geq 2$, we have

$$\mathbb{E}\left[(\bar{x}_v^{(k)} - p_v\mathbf{1}_k)^{\mathrm{T}} A^{(k)}(\bar{x}_v^{(k)} - p_v\mathbf{1}_k)|\mathcal{F}_{k+1}\right] = (\bar{x}_v^{(k+1)} - p_v\mathbf{1}_{k+1})^{\mathrm{T}} A^{(k+1)}(\bar{x}_v^{(k+1)} - p_v\mathbf{1}_{k+1}).$$

This fact follows from an expansion of the corresponding terms (similar to (45)), and then the proof follows as in Appendix D.1. Now, for any $v \in \mathcal{V}$, we define a richer class of filtration known as exchangeable filtration $\{\tilde{\mathcal{F}}_k^v : k \in \mathbb{N} \setminus \{1\}\}$ (Durrett, 2019), which denotes the $\sigma$-algebra generated by all real-valued Borel-measurable functions of $x_v^{(1)}, x_v^{(2)}, x_v^{(3)}, \ldots$ which are permutation-symmetric in the first $k$ arguments. It follows directly that $s_v^{(k)}$ is also a reverse-martingale with respect to $\tilde{\mathcal{F}}_k^v$. Therefore, by (Manole & Ramdas, 2023, Theorem 4), we have that the mapping $x \to \exp(\lambda x)$ for $\lambda \in (0, \infty)$, when applied to $s_v^{(k)}$, yields a reverse-submartingale with respect to filtration $\tilde{\mathcal{F}}_k^v$. Define the product $\sigma$-algebra $\tilde{\mathcal{F}}_k = \bigotimes_{v \in \mathcal{V}} \tilde{\mathcal{F}}_k^v$. Now, for any $k_0 \geq 2$ and $k_0 \in \mathbb{N}$, we have that

$$\mathbb{P}\left(\exists k \geq k_0 : \sum_{v \in \mathcal{V}} s_v^{(k)} \geq u\right) = \mathbb{P}\left(\exists k \geq k_0 : e^{\lambda \sum_{v \in \mathcal{V}} s_v^{(k)}} \geq e^{\lambda u}\right)$$

$$\leq \frac{\mathbb{E}\left[\exp\left(\lambda \sum_{v \in \mathcal{V}} s_v^{(k_0)}\right)\right]}{e^{\lambda u}}.$$

where the last step follows from Ville's inequality for nonnegative reverse submartingales (Manole & Ramdas, 2023, Theorem 2). Also note that $s_v^{(k)} \leq 1$ for all $k$ with probability 1, therefore $\mathbb{E}[e^{\lambda s_v^{(k)}}]$ always exists. Now note that

$$\sum_{v \in \mathcal{V}} (\bar{x}_v^{(k)} - p_v \mathbf{1}_k)^{\mathrm{T}} A^{(k)} (\bar{x}_v^{(k)} - p_v \mathbf{1}_k) = \sum_{v \in \mathcal{V}} \sum_{i,j \in [k]: i \neq j} \frac{(x_v^{(i)} - p_v)(x_v^{(j)} - p_v)}{k(k-1)}$$

$$= (\bar{x}_{\mathcal{V}}^{(k)})^{\mathrm{T}} A_{\mathcal{V}}^{(k)} (\bar{x}_{\mathcal{V}}^{(k)}),$$

where $(\bar{x}_{\mathcal{V}}^{(k)})^{\mathrm{T}} = [(\bar{x}_{v_1}^{(k)} - \mathbf{1}_k p_{v_1})^{\mathrm{T}}, \ldots, (\bar{x}_{v_{|\mathcal{V}|}}^{(k)} - \mathbf{1}_k p_{v_{|\mathcal{V}|}})^{\mathrm{T}}]$ is a vector in $\mathbb{R}^{k|\mathcal{V}|}$ formed by concatenating the vectors $\bar{x}_{v_i}^{(k)}$ for all $v_i \in \mathcal{V}$ and $i \in [|\mathcal{V}|]$. And $A_{\mathcal{V}}^{(k)} \in \mathbb{R}^{|\mathcal{V}|k \times |\mathcal{V}|k}$ formed by stacking the matrices $A_v^{(k)}$ as a diagonal block structure. Now by (Rudelson & Vershynin, 2013, Theorem 1.1), we have that there exists constants $c, c'$ such that we have

$$\mathbb{E}\left[e^{\lambda \sum_{v \in \mathcal{V}} s_v^{(k_0)}}\right] \leq \exp(-c\lambda^2 \|A_{\mathcal{V}}^{(k_0)}\|_{\mathrm{F}}^2) \text{ for } \lambda \leq c' / \|A_{\mathcal{V}}^{(k_0)}\|_2,$$

where $\|\cdot\|_2$ denotes the spectral norm. Therefore, we have that

$$\mathbb{P}\left(\exists k \geq k_0 : \sum_{v \in \mathcal{V}} s_v^{(k)} \geq u\right) \leq \exp(-\lambda u + c\lambda^2 \|A_{\mathcal{V}}^{(k_0)}\|_{\mathrm{F}}^2) \text{ for } \lambda \leq c' / \|A_{\mathcal{V}}^{(k_0)}\|_2.$$

Now, by optimizing over $\lambda$ and for some constant $\bar{c}$, we can conclude that

$$\mathbb{P}\left(\exists k \geq k_0 : \sum_{v \in \mathcal{V}} s_v^{(k)} \geq u\right) \leq \exp\left(-\bar{c} \min\left\{\frac{u^2}{\|A_{\mathcal{V}}^{(k_0)}\|_{\mathrm{F}}^2}, \frac{u}{\|A_{\mathcal{V}}^{(k_0)}\|_2}\right\}\right). \tag{47}$$

By the block structure of $A_{\mathcal{V}}^{(k_0)}$, we have the following

$$\|A_{\mathcal{V}}^{(k_0)}\|_{\mathrm{F}}^2 = \frac{|\mathcal{V}|}{k_0(k_0 - 1)}, \text{ and } \|A_{\mathcal{V}}^{(k_0)}\|_2 = \max_{v \in \mathcal{V}} \|A_v^{(k_0)}\|_2 = \frac{1}{k_0}.$$

Substituting, the above values in (47) we obtain that for all $\nu \in (0, 1/e)$, we have the following result for some constant $\tilde{c}$:

$$\mathbb{P}\left(\exists k \geq k_0 : \sum_{v \in \mathcal{V}} s_v^{(k)} \geq \tilde{c} \frac{\sqrt{|\mathcal{V}|} \log(1/\nu)}{k_0}\right) \leq \nu$$

The rest of the proof follows by a stitching argument (Zhao et al., 2016) and is provided below for completeness. For any

$\nu \in (0, 1/e)$, define a function $h(k) = \frac{(\pi k)^2}{6}$. Now, observe that

$$\mathbb{P}\left(\exists k \geq 2 : \sum_{v \in \mathcal{V}} s_v^{(k)} \geq \tilde{c}\frac{\sqrt{|\mathcal{V}|}}{k} \log\left(\frac{h(\log_2 k)}{\nu}\right)\right)$$

$$\leq \sum_{l=1}^{\infty} \mathbb{P}\left(\exists k \geq 2^l : \sum_{v \in \mathcal{V}} s_v^{(k)} \geq \tilde{c}\frac{\sqrt{|\mathcal{V}|}}{2^l} \log\left(\frac{h(l)}{\nu}\right)\right)$$

$$\leq \sum_{l=1}^{\infty} \frac{\nu}{h(l)} = \nu.$$

Finally, the statement in the lemma follows by a simple calculation. $\qquad\square$

### D.3. Proof of Theorem 3.7

Fix $t \geq 1$, and recall that from Lemma B.1, for some constant $c_0$ we have that with probability at most $e^{-t}$, $\|\mathsf{F} - \hat{\mathsf{F}}\|_{\mathrm{F}} \geq \sqrt{c_0 tn/k_1}$. This implies that with probability at least $1 - e^{-t}$, we have

$$\|\mathsf{F} - \hat{\mathsf{F}}\|_{\mathrm{F}}^2 + c\frac{\sqrt{|\mathcal{E}|}}{k}\ell_{k,\nu} + 4\frac{\|\mathsf{F} - \hat{\mathsf{F}}\|_{\mathrm{F}}}{\sqrt{k}}\sqrt{\ell_{k,\nu}} \leq \frac{c_0 tn}{k_1} + c\frac{|\mathcal{E}|^{\frac{1}{2}}}{k}\ell_{k,\nu} + 4\sqrt{\frac{c_0 tn\ell_{k,\nu}}{kk_1}}.$$

Therefore, by using a basic union-bound argument to the bound under null hypothesis in Lemma D.1, we have

$$\mathbb{P}_{H_0}\left(\exists k \geq 2, T^{k_1,k} \geq \frac{c_0 tn}{k_1} + c\frac{|\mathcal{E}|^{\frac{1}{2}}}{k}\ell_{k,\nu} + 4\sqrt{\frac{c_0 tn\ell_{k,\nu}}{kk_1}}\right) \leq \nu + e^{-t}.$$

The bound on type II error follows by a similar argument. First, using basic algebra, and using $D$ to denote $\|P - \mathsf{F}\|_{\mathrm{F}}$, we obtain from the bound under $H_1$ in Lemma D.1 that:

$$\mathbb{P}_{H_1}\left(\exists k \geq 2, T^{k_1,k} \geq (D - \|\mathsf{F} - \hat{\mathsf{F}}\|_{\mathrm{F}})^2 - c\frac{|\mathcal{E}|^{\frac{1}{2}}}{k}\ell_{k,\nu} - 4\frac{(D + \|\mathsf{F} - \hat{\mathsf{F}}\|_{\mathrm{F}})}{\sqrt{k}}\sqrt{\ell_{k,\nu}}\right) \leq \nu.$$

Now utilizing the fact that the $D \geq \sqrt{c_0 tn/k_1}$ and the same union bound technique as above, we obtain that

$$\mathbb{P}_{H_1}\left(\exists k \geq 2, T^{k_1,k} - \left(D - \sqrt{\frac{c_0 tn}{k_1}}\right)^2 \leq -\frac{c|\mathcal{E}|^{\frac{1}{2}}\ell_{k,\nu}}{k}\right.$$

$$\left. - 4\left(\frac{D}{\sqrt{k}} + \sqrt{\frac{c_0 tn}{k_1 k}}\right)\sqrt{\ell_{k,\nu}}\right) \leq \nu + e^{-t}.$$

Since the above bounds for any pairwise comparison matrix $P$ satisfying Assumption 2.3 and since $n\epsilon = \Theta(D)$ by Theorem 3.1, hence we can taking supremum with respect to pairwise comparison $P$ in class $\mathcal{M}_0, \mathcal{M}_1(\sqrt{\tilde{c}_0 t/(nk_1)})$ on type I and type II error bounds respectively, and thus completing the proof of the theorem. $\qquad\square$

## E. Simplified Expressions for Type I and II Error Probability Bounds

We obtain the following corollaries by plugging in the parameter values in Theorem 3.7 for a complete graph and a single cycle on $n$ nodes.

**Corollary E.1** (Type I and II Error Probability Bounds for Complete Graph). *For the setting in Theorem 3.7 assume that we have a complete graph on $n$ nodes, then there exists (different) constants $c_1, c_2, c_3, c_4, c_5 > 0$ such that for $\epsilon \geq c_3/\sqrt{nk_1}$, such that for all $\nu \in (0, 1/e)$ and $t \geq 1$, we have*

$$\mathbb{P}_{H_0}\left(\exists k \geq 2, T^{k_1,k} \geq \frac{c_1 tn}{k_1} + \frac{c_2 n\ell_{k,\nu}}{k} + c_3\sqrt{\frac{nt\ell_{k,\nu}}{kk_1}}\right) \leq \nu + e^{-t},$$

$$\mathbb{P}_{H_1}\left(\exists k \geq 2, T^{k_1,k} \geq \left(D - c_4\sqrt{\frac{tn}{k_1}}\right)^2 - \frac{c_2 n\ell_{k,\nu}}{k} - \left(4\frac{D}{\sqrt{k}} + c_5\sqrt{\frac{tn}{kk_1}}\right)\sqrt{\ell_{k,\nu}}\right) \leq \nu + e^{-t}.$$

**Corollary E.2** (Type I and II Error Probability Bounds for Single Cycle Graph). *For the setting in Theorem 3.7, assume that we have a single cycle graph on $n$ nodes, then there exists (different) constants $c_1, c_2, c_3, c_4, c_5 > 0$ such that for $\epsilon \geq c_3/\sqrt{nk_1}$, such that for all $\nu \in (0, 1/e)$ and $t \geq 1$, we have*

$$\mathbb{P}_{H_0}\left(\exists k \geq 2, T^{k_1, k} \geq \frac{c_1 tn}{k_1} + \frac{c_2\sqrt{n}\ell_{k,\nu}}{k} + c_3\sqrt{\frac{nt\ell_{k,\nu}}{kk_1}}\right) \leq \nu + e^{-t},$$

$$\mathbb{P}_{H_1}\left(\exists k \geq 2, T^{k_1, k} \geq \left(D - c_4\sqrt{\frac{tn}{k_1}}\right)^2 - \frac{c_2\sqrt{n}\ell_{k,\nu}}{k} - \left(4\frac{D}{\sqrt{k}} + c_5\sqrt{\frac{tn}{kk_1}}\right)\sqrt{\ell_{k,\nu}}\right) \leq \nu + e^{-t}.$$

# F. Experimental Details and Additional Experiments

In this appendix, we begin by providing additional details of the experiments corresponding to Section 4 in Appendix F.1. Furthermore, we perform several additional experiments in Appendix F.2 to support and extend our main findings. These include: (i) performance comparisons with the spectral method of (Makur & Singh, 2023) for the BTL model (see Appendix F.2.1), (ii) analysis of type I and II errors for different thresholding strategies (see Appendix F.2.2) and empirical validation of our threshold estimation approach under different settings (e.g., clustered versus randomly sampled skill scores), and (iii) experiments on real-world datasets (see Appendix F.2.3). Finally, we illustrate a method of computing confidence intervals in Appendix F.3 based on the expression in Proposition 3.8.

## F.1. Additional Details for Experiments

Below we provide additional details on the experimental setup and methodology for the experiments in Section 4.

**Error bars and estimation of $\hat{w}$:** To estimate the 95th quantile of the test statistics $T$, we used 400 samples. The error bars were based on the two-sided distribution-free conservative estimates presented in (Hahn & Meeker, 1991). Specifically, for the 95th quantile, the upper 96% confidence interval was computed as the 97th quantile of the computed tests $T$, and the lower confidence interval was computed as the 92.5% quantile. Moreover, in all of our experiments, the estimation of $\hat{w}$ was performed using a standard gradient descent algorithm with a learning rate of 0.01 and for a maximum of 3000 iterations, or until the norm of the gradient was less than $10^{-5}$.

**Testing on LMSYS dataset:** In our experiment on the LMSYS dataset, we used a maximum of 200 samples per pair and discarded pairs with fewer than 30 observed comparisons to reduce the imbalance in the data across pairs. The observation graph was a complete graph except for the larger values of $n$, which had a few edges missing.

**Computational overhead of empirical quantile approach:** While computing the threshold using the empirical quantile approach can be expensive due to repeated simulations, we remark that the procedure is highly parallelizable and is fast even with a naïve implementation (without any parallelization); indeed, each of our experiments can be done within 5 minutes on a normal CPU for $n$ as large as 60. Additionally, to speed up computing $\hat{w}$ on the simulated dataset, one can initialize the iterates at the optimal values that generated the simulated dataset and then apply a few iterations of gradient descent. Moreover, when all $k_{ij}$ are equal to $k$ (or roughly the same), our simulation results in Figure 2 suggest that $0.75n/k$ and $0.4n/k$ are good approximations of the thresholds for complete graphs and 2D grids, respectively. On the other hand, Appendix F.3 provides a faster asymptotic alternative based on Proposition 3.8.

## F.2. Additional Experiments

### F.2.1. COMPARISONS TO BASELINE METHODS

We now conduct experiments to compare the performance of our test with the test proposed in (Makur & Singh, 2023) by fixing the choice function $F$ to be the logistic function (i.e., BTL model). We assume that the observation graph is complete and define the pairwise comparison matrix $P$ under the null and alternative hypotheses as the one used to prove the lower bound in (39) and (40). We set $\eta = 0.06$ and $k_{ij} = 10$ for all $i \neq j$. We evaluate the empirical type I and II error probabilities, averaged over 400 trials, for three testing methods: our proposed likelihood-based statistic with sample splitting (Max. Likelihood), the same statistic without any sample splitting (Max. Likelihood2), and the spectral approach from (Makur & Singh, 2023) (Spectral). The threshold for all three methods is determined using the empirical quantile approach as detailed in Section 4. Our results in Figure 3 indicate that the proposed method performs comparably to the spectral approach, while Max. Likelihood2 achieves even lower type I and II sum error rates. Various parameters, such as the number of simulated datasets and construction of error bars, are the same as those used in Section 4.

### F.2.2. COMPARISON OF EMPIRICAL AND ANALYTICAL THRESHOLDS

To evaluate the reliability of our empirical-quantile-based threshold, we compare the empirical-quantile-based threshold with the "optimal" theoretical threshold in a well-crafted setting. Specifically, we consider the same setting as in Appendix F.2.1 and set the theoretical threshold as $\eta^2 n(n-1)/2$ across a range of values of $n$. Specifically, we vary $\eta \in [0.08, 0.16]$ and $n \in \{15, 25, 35, 45\}$, while fixing $k_{ij} = 10$ for all $i \neq j$. For each configuration, we compute type I and II error probabilities using the empirical-quantile-based threshold (cf. Section 4) and the theoretical thresholds averaged over 400 trials. As shown in Figure 4, the empirical threshold achieves type I error rates that closely track the nominal 0.05 level and achieves nearly identical type I and II sum-error rates as that of the analytical threshold. This suggests that the empirical threshold, despite being derived from random samples of skill scores, provides performance comparable to that of the theoretically optimal threshold.

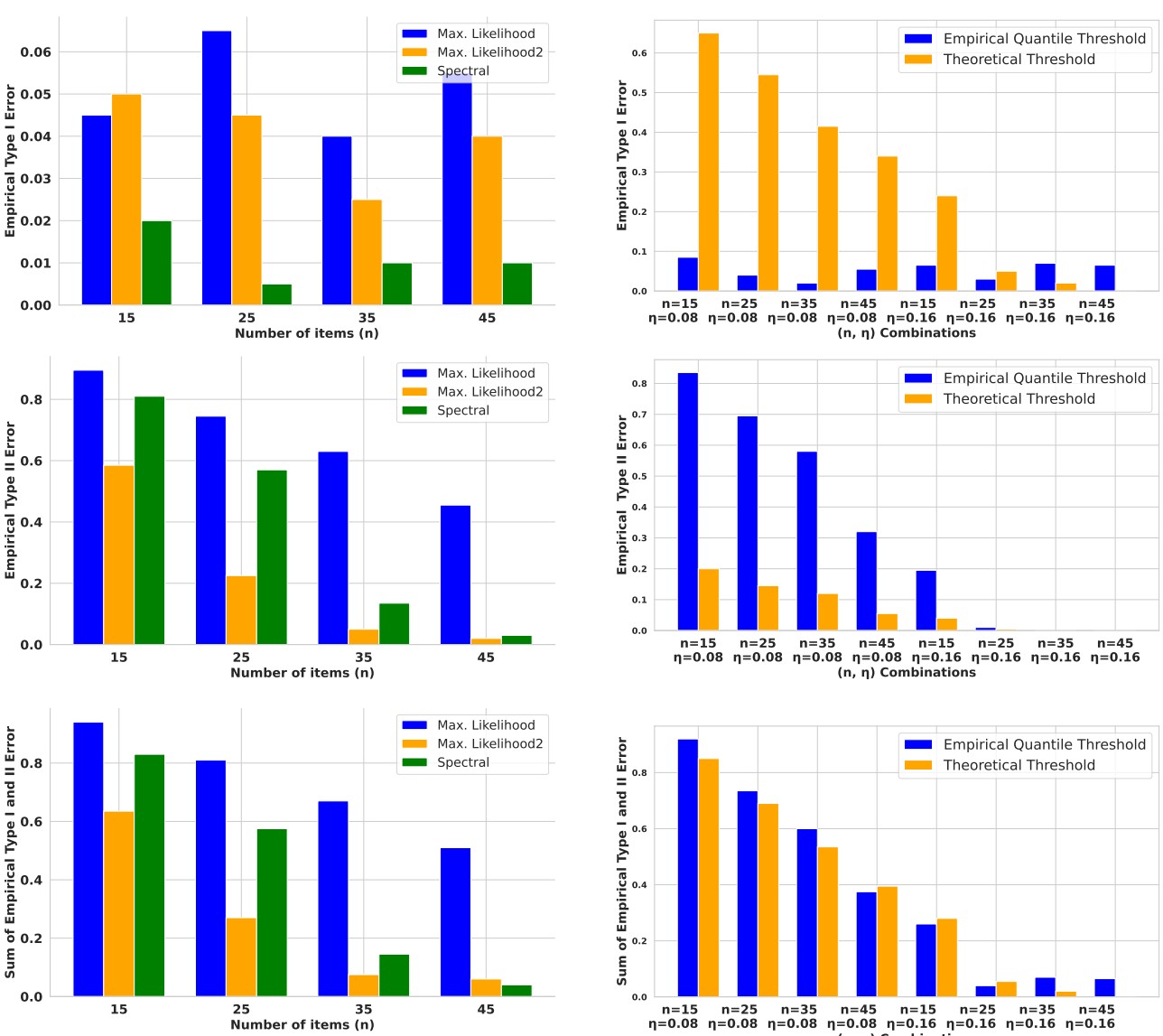

Figure 3: Performance comparison of spectral method in (Makur & Singh, 2023; 2024) and our proposed method with (Max. Likelihood) and without partitioning (Max. Likelihood2) of dataset.

Figure 4: Empirical type I and II error rates along with their sum for the Empirical Quantile method compared to a well-crafted threshold.

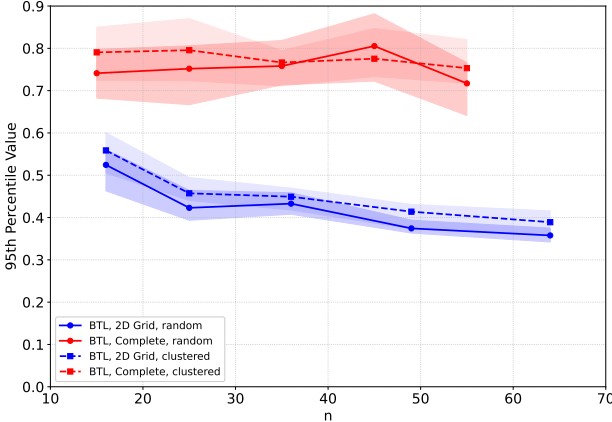
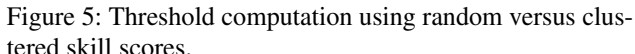

Figure 5: Threshold computation using random versus clustered skill scores.

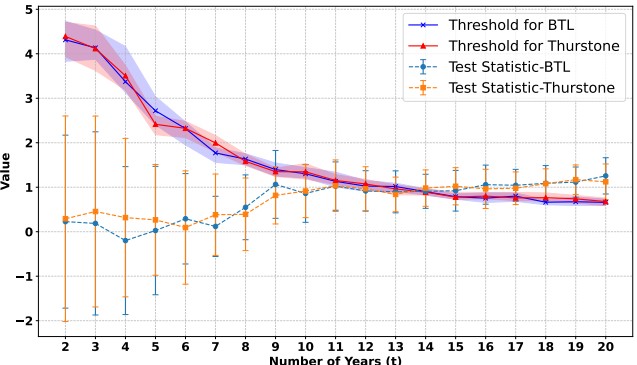

Figure 6: Scaled test statistic $n \cdot T$ for BTL and Thurstone models computed on NBA dataset on cumulative data of $t$ recent years and thresholds computed using empirical-quantile-based approach.

To address potential coverage concerns arising from the use of uniformly sampled skill scores in our empirical quantile thresholding approach, we conduct an additional experiment to test the sensitivity of the threshold. In our main setup, scores are sampled uniformly from the interval $[-b, b]$, which may not fully reflect more structured or clustered configurations. To evaluate the impact of this assumption, we construct a clustered distribution by assigning half of the skill scores uniformly at random in the range $[-0.7, -0.4]$ and setting the remaining half to be their exact negatives, thereby introducing a symmetric bimodal structure. As shown in Figure 5, the empirical thresholds derived from the uniform and clustered settings are nearly identical.

### F.2.3. ADDITIONAL EXPERIMENTS ON REAL-WORLD DATASET

In addition to our LMSYS dataset analysis, we apply our testing procedure to historical NBA match outcomes using the publicly available dataset from Kaggle (Lauga, 2023). We focus on games played until the 2022 season and restrict attention to the 12 teams with the highest number of games played since 2002. Each data point corresponds to a comparison between a home and an away team, and we evaluate our hypothesis test on cumulative game data aggregated over a rolling window of $t$ recent years. As shown in Figure 6, for small values of $t$ (approximately the first 10 seasons), the data is consistent with the BTL and Thurstone models. However, as $t$ increases, the hypothesis is increasingly rejected, suggesting that a single latent skill score per team fails to adequately represent team strength over long horizons of time.

### F.3. Confidence Intervals under Null Hypothesis

In this subsection, we will discuss a method to approximately calculate the confidence intervals under the null hypothesis, with a focus on the BTL model. While our discussion is specific to the BTL model, it can be easily generalized to other Thurstone models. Specifically, our goal is to approximately calculate the constants in Proposition 3.8 and as well as approximate the distribution of $\|\hat{\mathsf{F}} - \mathsf{F}\|_{\mathrm{F}}$. For the former, we will estimate the constants by conducting some simulation, while for the latter, we will be utilizing Gaussian approximation based on the asymptotic normality of $\hat{w} - w^*$, which was proved for the BTL model (Gao et al., 2023, Proposition 4.1). Similar results have been established for the general Thurstone models as in (Han et al., 2024).

**Estimating constant $c_7$ in Proposition 3.8:** To estimate the constant, we plot several trajectories of the normalized stochastic process $\frac{\sqrt{k}}{|\mathcal{V}|} \sum_{v \in \mathcal{V}} (\bar{x}_v^{(k)} - p_v \mathbf{1}_k)^{\mathrm{T}} A^{(k)} (\bar{x}_v^{(k)} - p_v \mathbf{1}_k)$ with $\bar{x}_v^{(k)}$ generated as in Lemma D.2 and the $p_v$ selected uniformly at random from $(0, 1)$. The values of $|\mathcal{V}|$ varied from 10 to 100 with gaps of 10. Figure 7 plots the various trajectories of the (normalized) stochastic process as a function of $k$ and also plots the 95th quantile for the stochastic process for all $k \in [100]$. The figure suggests that $c_7 \approx 0.45$ is a good approximation to the value of $c_7$ (in the fixed sample setting).

**Estimating the quantile of $\|\mathsf{F} - \hat{\mathsf{F}}\|_{\mathrm{F}}$:** In order to estimate $\|\mathsf{F} - \hat{\mathsf{F}}\|_{\mathrm{F}}$, we will utilize the asymptotic normality of vector $\Delta = \hat{w} - w^*$. Let $\hat{w}$ be computed as in (7), then under mild regularity conditions, it was shown in (Gao et al., 2023,

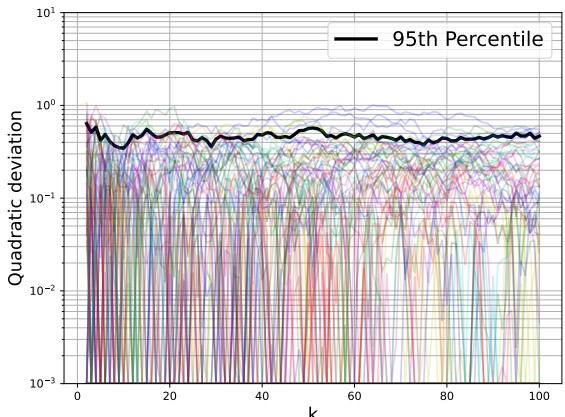

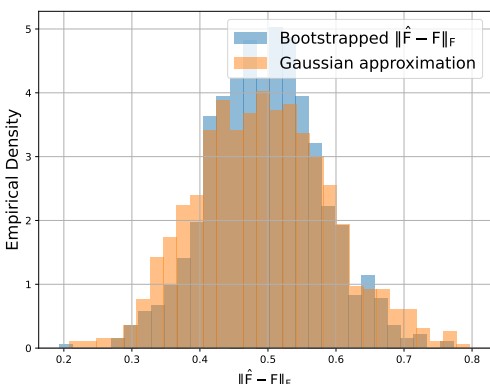

Figure 7: Plot of various trajectories of stochastic process in Lemma D.2.

Figure 8: Histogram of $\|\mathsf{F} - \hat{\mathsf{F}}\|_{\mathrm{F}}^2$ based on bootstrapping versus asymptotic approximation.

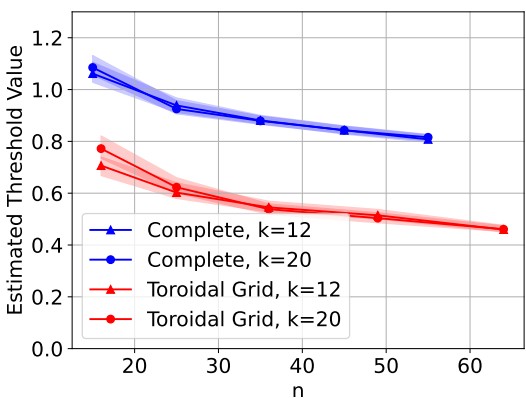

Figure 9: Estimated threshold based on Proposition 3.8.

Proposition 4.1) that

$$(\rho_1(\hat{w})(\hat{w}_1 - w_1^*), \ldots, \rho_n(\hat{w})(\hat{w}_n - \hat{w}_n^*)) \xrightarrow{d} \mathcal{N}(0, I_k),$$

where $\rho_i(\hat{w}) = \sqrt{k \sum_{j:(i,j) \in \mathcal{E}} F'(\hat{w}_i - \hat{w}_j)}$ and $\xrightarrow{d}$ denotes the convergence in distribution. We will utilize this asymptotic normality result to approximate the distribution of $\|\hat{\mathsf{F}} - \mathsf{F}\|_{\mathrm{F}}$ using the delta method as:

$$\begin{aligned}
\|\mathsf{F} - \hat{\mathsf{F}}\|_{\mathrm{F}}^2 &= \sum_{(i,j) \in \mathcal{E}} \left( F\left(w_i^* - w_j^*\right) - F(\hat{w}_i - \hat{w}_j) \right)^2 \\
&\approx \sum_{(i,j) \in \mathcal{E}} F'(\hat{w}_i - \hat{w}_j)^2 \left( \left(w_i^* - \hat{w}_i\right) - \left(w_j^* - \hat{w}_j\right) \right)^2 \\
&= \sum_{(i,j)} F'(\hat{w}_i - \hat{w}_j)^2 (\Delta_i - \Delta j)^2 = 2\Delta^{\mathrm{T}} L_F(\hat{w})\Delta,
\end{aligned}$$

where we define $L_F(w)$ to be the following matrix

$$(L_F(w))_{ij} = \begin{cases} -F'(w_i - w_j)^2 & \text{if } (i,j) \in \mathcal{E} \\ \sum_{j:(i,j) \in \mathcal{E}} F'(w_i - w_j)^2 & \text{if } i = j \\ 0 & \text{otherwise.} \end{cases}$$

Since $\Delta$ is asymptotically normal (as $k \to \infty$), therefore we approximate the distribution of $\|\mathsf{F} - \hat{\mathsf{F}}\|_{\mathrm{F}}^2$ with distribution

of $2\Delta^{\mathrm{T}}(\hat{w})L_F(\hat{w})\Delta(\hat{w})$ where $\Delta_i(\hat{w}) \sim \mathcal{N}\left(0, \frac{1}{k\sum_{j:(i,j)\in\mathcal{E}}F'(\hat{w}_i-\hat{w}_j)}\right)$. In Figure 8, we plot the empirical distribution of $\|F(\hat{w}) - F(w^*)\|_F$ calculated by randomizing over the choice of partitioning of $\mathcal{Z}$ into $\mathcal{Z}_1$ and $\mathcal{Z}_2$. We also plot its asymptotic approximation, i.e., the empirical distribution of $2\Delta^{\mathrm{T}}(\hat{w})L_F(\hat{w})\Delta(\hat{w})$. Clearly, as can be seen in Figure 8, our asymptotic approximation does indeed well approximate the empirical distribution even for a small number of samples. Finally, based on the the 95th percentile of the empirical distribution of $2\Delta^{\mathrm{T}}(\hat{w})L_F(\hat{w})\Delta(\hat{w})$, we compute the expression for $c_7 = 0.45$ and plot the estimated confidence intervals in Figure 9. It is worth noting that the estimated threshold values for complete graphs converge towards the theoretical value of $0.8$, while those for toroidal grids approach $0.4$. These findings are consistent with the threshold values computed via the empirical quantile method presented in Figure 1 for the respective graph topologies. Together, these observations suggest that while the exact distribution of $T$ is difficult to characterize, its tail can be asymptotically approximated using a quadratic function of Gaussian random variables via Proposition 3.8.

## G. Testing for $\mathcal{T}_F$ Models in TV and Spectral Norms

In Section 2, we framed the hypothesis testing problem using the Frobenius norm due to its analytical tractability, particularly in the context of maximum-likelihood-type estimators. There is also significant precedent for employing quadratic distance measures in classical statistical testing—for instance, the $\chi^2$-test essentially utilizes a squared weighted $\ell_2$-distance. However, in some applications, alternative notions of separation may be more desirable. One such example is the TV distance, which has appealing properties, such as Le Cam's relation and the data processing inequality (cf. (Makur, 2019; Makur & Zheng, 2020)).

Recall that in (11), we defined the separation distance from the class of $\mathcal{T}_F$ models using the Frobenius norm. Alternatively, for any pairwise comparison matrix $P \in [0,1]^{n\times n}$, we can define the TV separation distance from the model class $\mathcal{T}_F$ as

$$\mathsf{TV}(P, \mathcal{T}_F) \triangleq \inf_{w\in\mathcal{W}_b} \frac{1}{|\mathcal{E}|} \sum_{(i,j)\in\mathcal{E}} |p_{ij} - F(w_i - w_j)|.$$

Using $\mathsf{TV}(P, \mathcal{T}_F) \geq \epsilon$ in (11) leads to an equivalent hypothesis testing problem, but with the separation measured in TV distance. Utilizing standard norm-equivalence inequalities, such as $\|x\|_1 \leq \sqrt{n}\|x\|_2$ for $x \in \mathbb{R}^n$, we can relate this to the Frobenius norm as

$$\mathsf{TV}(P, \mathcal{T}_F) \leq \inf_{w\in\mathcal{W}_b} \frac{1}{\sqrt{|\mathcal{E}|}}\|P - F(w)\|_{\mathrm{F}}.$$

Therefore, if the TV separation satisfies $\mathsf{TV}(P, \mathcal{T}_F) \geq \epsilon$, it follows that

$$\inf_{w\in\mathcal{W}_b} \frac{1}{n}\|P - F(w)\|_{\mathrm{F}} \geq \frac{\epsilon}{n}\sqrt{|\mathcal{E}|}. \tag{48}$$

Recall from the minimax formulation in (12) that our test distinguishes between the null and alternative hypotheses with minimax risk at most $1/2$, provided the Frobenius separation exceeds $c/\sqrt{nk}$ (see Theorem 3.2). Utilizing the inequality (48), this implies that our test retains minimax risk at most $1/2$ whenever

$$\frac{\epsilon}{n}\sqrt{|\mathcal{E}|} \geq \frac{c}{\sqrt{nk}},$$

which implies that the TV separation distance $\mathsf{TV}(P, \mathcal{T}_F) \geq c\sqrt{n/(|\mathcal{E}|k)}$ is sufficient for testing (yielding an upper bound on critical threshold for TV distance). Notably, for complete graphs, this reduces to the critical threshold of $O(1/\sqrt{nk})$, which also matches the lower bound established by (Seshadri & Ugander, 2019) for the same notion of TV separation distance.

Finally, similar arguments allow our upper bounds to be extended to the spectral norm, using the inequality $\|A\|_2 \leq \|A\|_{\mathrm{F}}$ for any matrix $A$. Consequently, our test also guarantees a small minimax risk under spectral norm separation. However, obtaining corresponding lower bounds under the spectral norm remains an open problem.

