# OpenReview forum: "Hypothesis Testing for Generalized Thurstone Models"
_ICML.cc/2025/Conference — ICML 2025 poster_

### Official Review · Reviewer_gx7K · 2025-03-06

**Overall Recommendation:** 4

**Summary:**

In this paper, the authors look at the problem of hypothesis testing of generalized thurstone models. The later models are used to model ranking among several entities based on pairwise comparisons. Extensive research has been done to learn the parameters of such a model. However, an important questions is given a set of empirical pairwise comparisons does the observations correspond to an underlying GTM? This hypothesis question apparently has not been looked so far. The authors initiate a formal theoretical study of this testing problem and came up with a formalism and test statistic which is the main contribution of the paper.

**Claims And Evidence:**

The main claim is that the proposed hypothesis questions is useful in checking whether a given set of samples of pairwise comparisons correspond to a GTM or not. They also give experimental evidence to support the theoretical guarantees.

**Essential References Not Discussed:**

NA

**Experimental Designs Or Analyses:**

The experiments were conducted on both synthetic and benchmark data. For both cases it shows that the proposed methods works well to determine the test threshold using a data-driven approach.

**Methods And Evaluation Criteria:**

Yes

**Other Comments Or Suggestions:**

NA

**Other Strengths And Weaknesses:**

I think the proposed testing problem is novel and the techniques proposed are sound.

**Questions For Authors:**

What happens if we define the testing problem with respect to other matrix norms such as the spectral norm?

**Relation To Broader Scientific Literature:**

This works initiates the hypothesis testing of GTMs. However some special cases appeared in the literature before. The proposed method generalized beyond these special cases.

**Theoretical Claims:**

Yes, in my opinion. The formalism of the testing problem is as follows

A good weight vector w must be orthogonal to the all ones vector and must have bounded values. The hypothesis testing problem distinguishes the following two cases.
- there exists a good weight vector w such that the probability of pairwise comparisons follow a cumulative probability according to some distribution F with respect to the difference of w values
- the pairwise comparisons is eps far in Frobenius norm from any such good weight vector

I didn't check the proof but here are the main theoretical claims.

- we roughly need n/eps samples to answer the above testing problem
- the upper bound is supported with a suitable lower bound based on LeCam method
- confidence intervals have been derived for the test statistic

---

> ### Author Rebuttal · Authors · 2025-03-31
>
> Thank you for your insightful feedback and for dedicating time to review our paper! We respond to the specific questions as follows:
>
> **Testing problem with respect to other norms**: We believe the Frobenius norm is a natural choice for our problem as it allows tractable analysis of maximum-likelihood-type methods. There is some precedent to utilizing such quadratic distances in hypothesis testing, such as the classical chi-squared test. Another popular choice for separation distance is the sum of total variation (TV) distances. Using the equivalence of norms ($||x||_1 \leq \sqrt{n} ||x||_2$), our results can be translated to TV distance, and they again turn out to be tight for complete graphs. For the TV separation distance $TV(P, \mathcal{T}_F)$, a simple calculation gives:
>
> $TV(P, \mathcal{T}_F) =$
>
> $\inf_{w \in \mathcal{W}_b }$
>
> $\sum_{(i,j) \in \mathcal{E}}$
> $\frac{1}{|\mathcal{E}|} |p_{ij} - F(w_i - w_j)| \leq  \inf_{w \in \mathcal{W}_b } \frac{1}{\sqrt{|\mathcal{E}|} } ||P - F(w)||_F$
>
> Recall that our test can distinguish whether $P = F(w)$ for some $w \in \mathcal{W}_b$ or
>
>  $\inf_{w \in \mathcal{W}_b} \frac{1}{n}  ||P - F(w)||_F \geq c/\sqrt{nk}$
> with a minimax risk at most 1/2. If $TV(P, \mathcal{T}_F) \geq \epsilon$, then using above equation, we can conclude that
>
> $$\inf_{w \in \mathcal{W}_b } \frac{1}{n} ||P - F(w)||_F \geq  \frac{\epsilon}{n} \sqrt{|\mathcal{E}| }$$
>
> Thus our test has small minimax risk if  $\frac{\epsilon}{n} \sqrt{|\mathcal{E}| } \geq c/\sqrt{nk}$, which is equivalent to $\epsilon \geq c\sqrt{n/(|\mathcal{E}| k)}$. This implies that the critical threshold for our test for the quantity $TV(P, \mathcal{T}_F)$ is  $O(\sqrt{n/(|\mathcal{E}| k)})$ which reduces to $O(1/\sqrt{nk})$ for complete graphs. Interestingly, this bound matches the lower bound in [Seshadri & Ugander, 2020] for complete graphs, derived for the same notion of TV separation distance.
>
> Moreover, using the inequality $||A||_2 \leq ||A||_F$ for any matrix $A$ and the same argument as above, we can translate our upper bounds to the spectral norm. However, we don't have lower bounds for the spectral norm.
> We will incorporate these comments into the manuscript.

---

### Official Review · Reviewer_3Nsw · 2025-03-07

**Overall Recommendation:** 3

**Summary:**

Covers “Generalized Thurstone models,” in which each player has a utility, and the probability of winning is a function of the difference in utilities – but this choice function is an arbitrary CDF. Nice motivation to observe that GTMs do not capture certain types of choice dynamics, so asking whether data are consistent with any GTM is a useful question. The general question regarding whether *any* GTM is consistent with a dataset is very interesting, and also technically intricate -- this is an excellent question to ask, and the authors make significant progress.

## update after rebuttal

Note that my questions below have been addressed in the authors' rebuttal.

**Claims And Evidence:**

The abstract claims that the results are "validated" through experiments on synthetic and real-world datasets. I don't think is quite accurate. It seems that, instead, the synthetic experiments (basically just one in the main paper body) show empirically how the test statistic is distributed for a particular distribution over in-class tournaments, while the real-world datasets (also just one) also pursue the same methodology to estimate the test statistic, and show that the statistic is distributed roughly as expected for GTM models for the most popular models, but not for comparisons that involve less popular models. This is a suggestive finding, but does not seem to be explored in more detail.

**Essential References Not Discussed:**

Perhaps consider this reference at PNAS on sample complexity where your assumption of winning probabilities bounded away from {0,1} does not hold: https://www.pnas.org/doi/10.1073/pnas.2202116119

**Experimental Designs Or Analyses:**

See below for some comments.

**Methods And Evaluation Criteria:**

I feel that the empirical evaluation is less thorough than the theoretical parts of the paper. There are some interesting findings, but I don't come away with a strong understanding of applying the techniques developed here to a broad range of settings. See some more specific points in the general comments and questions below.

**Other Comments Or Suggestions:**

Slight confusion: equation 1) speaks to a specific choice function F, while main contribution 1) speaks to the distance to the family of GTMs. At this point in the draft, I was uncertain whether the focus was examining a null hypothesis regarding a single GTM, versus the entire family. This became clear later on.

My summary and some comments on the theoretical section is given above under "Theoretical Claims."

Here are some comments, questions, and thoughts regarding the Experiments section:

“Random skill scores” – does this mean uniform in [-b,b]? This is a good strawman, but not a broad coverage of what happens in settings where utilities are not drawn uniformly (eg, what happens if there are clusters of utilities, etc). Could the authors please comment about this assumption? I discuss next that the synthetic experiments are quite tightly tied to this assumption, as it induces the distribution over test statistic values (unless I am misunderstanding).

The first experiment as I understand it measures the distribution of the test statistic induced by the very specific underlying model (random skill scores) – I’m not sure how to read the finding from this experiment, as I can’t imagine an actual experimental setting in which this is a good probabilistic model of reality…? Related to this, on the idea of “repeating the process a sufficient number of times to build a distribution of test statistics” – is anything known about the true distribution of the test statistic for a GTM?

Perhaps my biggest question on the experiments is that I seem to be missing something high level. I don’t see the connection between the empirical test statistics and the various upper and lower bounds from the rest of the paper. Could the authors please clarify this, and whether anything more can be done experimentally to tighten the connection of the theoretical results to the questions that practitioners might face?

Related to this, there are no experiments that test whether the empirical quantile approach can actually distinguish between, say, a random skill scores GTM model, and a related model chosen to be a certain known distance from any GTM model. Is it possible to perform such an experiment? It’s hard to know how to interpret

For the LMSYS results, what does it mean that 60% of values are above the threshold, beyond the fact that the estimation procedure underestimates the critical value? Is this an issue of the increased variance of estimating p_{ij} due to sparsity of results for each (i,j) pair? Or, as I read it, is the idea that somehow less popular language models do *not* have a fundamental "quality" that can be used to estimate their likelihood of generating a better answer than a competitor? If the latter, this seems like a somewhat startling finding, and would benefit from more discussion (and analysis), as it now speaks more centrally to LLM testing, an issue of great importance right now.

Note: Figure 2 caption references LYMSYS (as does later text) -- is this supposed to be LMSYS? Otherwise, please clarify.

**Other Strengths And Weaknesses:**

Please see below under comments and suggestions

**Questions For Authors:**

In addition to the questions above, two more things:

1. What is the intuition behind your test statistic? Is there some reason to think this is the best or approximately best (ie, variance-minimizing) test statistic for GTM or specific choice functions?

2. Are there other simplifications that would allow considering your bounds relative to other standard statistical tests? Perhaps a step function choice function?

**Relation To Broader Scientific Literature:**

The authors do a good job of laying out the landscape for this area.

**Theoretical Claims:**

I did not check proofs. For the supplementary material in particular, I just scanned at a very high level.

Here is my rough understanding of the flow of the theoretical parts of the main paper body, with a few notes for suggestions inlined:
1. Model: each directed comparison (i,j) is either present or not. If present, the probability of winning is known fully. The graph is fixed in advance, not dependent on outcomes.
2. Note: Some confusion in notation: Definition 2.2 places no restrictions on F beyond R → [0,1], but then in Equation 3 it’s defined in terms of noise distribution G.
3. Pairwise comparison data is then (Equation 5) defined as bernoulli random variables parameterized by the winning probability of a competition.
4. The authors then define likelihood of observations, assume winning probabilities are bounded away from 0 and 1. The estimation problem is given with a bound b of magnitude of weights – seems unnatural, and shouldn’t be necessary with this assumption on winning probabilities, but later sections make clear that there are some technical issues for which this is necessary. Question for authors: is this restriction necessary because of the techniques, or is it fundamental?
5. Add some assumptions on strong log-concavity and bounded derivatives for F – this guarantees a unique solution for the likelihood estimation problem, and holds for some common GTMs.
6. Note: Section 2.3: why are the constants "universal" when they depend on F? Assuming it’s just \delta,\epsilon that are universal? Possible to clarify this in the text?
7. The authors then define their testing problem to differentiate between H_0, in which the winning probabilities come from some utilities under F, and H_1, in which any utilities result in a winning matrix bounded away from the actual one. They then show this is approximated by the frobenius norm difference between the true winning matrix, and the one under the likelihood-maximizing utilities.
8. From here, they define a test statistic, and ask what properties of the graph and observations are needed for the “minimax risk” (a measure of the probability of type-1 and type-2 errors) to be bounded by a constant.
9. They study the “critical threshold,” which is the distance lower bound in the H_1 null hypothesis. They show first there is an upper bound like c/sqrt(nk) on the critical threshold, where n is #players, and k is (half) number of games per observed pair. Their test uses half the data to estimate the utilities, and the other half to compute the statistic.
10. Their test holds even when the observation graph is disconnected – but I wonder what happens with their utility estimation procedure since it’s no longer identified? Possible to comment on this?
11. Next they consider lower bounds on the critical threshold. The lower bound is based on Seshadri&Ugander, but extends to GTM and more general observation graphs. These seem to be tight to within constants for complete graphs.
12. They also consider some upper bounds on type 1 and 2 error probabilities when comparisons arrive by “rounds” of games, each round containing a bernoulli draw per edge. I have to say, I’m not confident how to interpret the results of Theorem 7, including how tight the resulting confidence intervals might be. Perhaps the authors could include some more discussion about this -- I'd love to hear what is possible in the rebuttal.

---

> ### Author Rebuttal · Authors · 2025-03-31
>
> Thank you for detailed feedback. Due to 5000 character limit, our responses are concise. We will incorporate the addressed points and new experiments in the final version
>
> **Experimental Concerns**
> * Experiments not explored in detail: Our focus was on theoretical aspects of the testing problem. To clarify, our first synthetic experiment shows that the threshold obtained via the empirical quantile approach follows the same high-level scaling as the theoretical threshold in Eq 37 (see Fig1). We also showed how to obtain a threshold using an asymptotic approximation in App F.2. **To address concerns, we added new experiments on real & synthetic data (see response to Reviewer smfn)**
> * Interpreting Exp 1 findings: See response above
> * Broader application: Validating the BTL model is crucial due to its wide applications(eg. in RLHF), where flexible ranking models are preferred. We believe our test could extend to RLHF where BTL scores are given by neural network/linear models.
> * Connection b/w theory and practice: Our results on critical threshold and confidence intervals (CI), which follow from Type I & II error bounds, connect to our 1st synthetic experiment and App F.2 (see above).
> * Validating empirical quantile approach: We have designed an experiment to validate its effectiveness in a synthetic setting. See Exp2 in response to Reviewer smfn
> * Interpretation of LMSYS Results: The 60% above-threshold rate is a bootstrapped estimate of the test's power indicating ~40% chance that the model lies within 95% statistical deviations from BTL. The figure shows that as n increases, deviation from Thurstone increases, a top-9 batch size provides a statistically accurate fit to the Thurstone model, and the deviations are significant for $n\geq21$.
> * Sparsity of $p_{ij}$: There is no sparsity as the graph is nearly complete
>
> **Necessity of Bound b**: This restriction arises from our techniques for bounding separation distance. While we believe the bound b can be removed for certain cases (eg.complete graph) the analysis is significantly more complex. For general graphs, it is unclear whether b is fundamental; this is an interesting future problem. Additionally, assuming bounded weights is standard in literature on parametric models [Shah et al. 2016].
>
> **Non-uniqueness of w in Disconnected Graph**: Our analysis uses error bounds in Laplacian semi-norm. When $G$ is disconnected, solutions $w^*,\hat{w}$ of Eq 7,8 may not be unique, but the error $||\hat{w}-w^*||_L$ is well-defined as the non-unique component lies in the null space of $L$. Thus, our upper bounds on critical thresholds still hold. Additionally, our lower bounds do not assume connectivity and hold for disconnected super-Eulerian graphs.
>
> **Interpretation of Thm. 7 & Tightness of CIs**:
> * Thm. 7 can be reduced from the sequential setting to the standard testing where data is available upfront.
> * It implies Type I & II errors decay exponentially when separation distance $\gg1/\sqrt{nk}$.
> * The bounds apply to any partition of data into $\mathcal{Z}_1$ and $\mathcal{Z}_2$, offering guidance on split-size based on graph topology (e.g., equal split for complete graphs, larger $k_1$ for better type I control in cycle graphs). See Appendix E for details
> * To see tightness of CIs, we can compare Figs 1 and 5 for complete and grid graphs, with scaled threshold values of ~0.75 & ~0.45.
>
> **Random Skill Scores**: Yes, the scores are drawn independently and uniformly in [-b,b] and translated to satisfy $w^T1=0$. While we acknowledge concerns about coverage with random sampling, in the context of finding thresholds, our experiments show that the resulting threshold exhibits the same scaling as theory predicts and achieves good empirical type I and II errors compared to a well-crafted threshold in a synthetic setting (see Exp2).
>
> **Notational Confusion**:
> * Definition 2.2: Sure, we will remark that F is a special CDF and add examples.
> * Universal constant: Once graph $G$ choice function $F$, parameters $\delta,b$ are fixed, all the constants are universal as they depend only on these entities. We will remove 'universal' to avoid confusion. Note:$\epsilon$ is not constant as it can depend on $n,k$
> * LYMSYS/LMSYS: Yes, it is LMSYS
>
> **Test statistic**:
> * Distribution: The distribution of $T$ is indeed hard to characterize, but using Prop. 3.8, we can asymptotically approximate its tail by a quadratic function of Gaussian random variables (see App F.2)
> * Intuition: See response to Reviewer smfn. Our statistic may not be optimal in the sense you propose, but it is analytically tractable and does characterize the critical threshold.
>
> **Essential References**: We will discuss it in final version.
>
> **Other statistical tests** Our problem is a minimax composite hypothesis testing problem, which lacks a non-asymptotic theory in general. The only direct comparison of our bounds is with [Makur, Singh 2023] in the BTL case; both results have the same critical threshold scaling.

---

> > ### Comment · Reviewer_3Nsw · 2025-04-03
> >
> > Thank you for your detailed responses, they are very helpful

---

### Official Review · Reviewer_smfn · 2025-03-08

**Overall Recommendation:** 3

**Summary:**

This work develops a hypothesis testing framework to determine whether pairwise comparison data follows a generalized Thurstone model for a given choice function, introducing a minimax separation distance to quantify deviations from such models. The study establishes theoretical bounds on the critical threshold based on the observation graph's topology, proposes a hypothesis test with confidence intervals, and establish time-uniform bounds on type I and II errors using reverse martingale techniques.

**Claims And Evidence:**

My primary concern is the definition of the test statistic in Equation (15), as the paper lacks an intuitive explanation for its formalization. Furthermore, the relationship between this test statistic and those proposed in Rastogi et al. (2022) (Equation 8) and Makur & Singh (2023) (Equation 11) remains unclear, **requiring further clarification regarding its advantages in both theoretical analysis and experimental performance.**

References:

Rastogi, C., Balakrishnan, S., Shah, N. B., and Singh, A. Two-sample testing on ranked preference data and the role of modeling assumptions. Journal of Machine Learning Research, 23(225):1–48, 2022.

Makur, A. and Singh, J. Testing for the bradley-terry-luce model. In 2023 IEEE International Symposium on Information Theory (ISIT), pp. 1390–1395, 2023. doi:10.1109/ISIT54713.2023.10206450.

**Essential References Not Discussed:**

N/A

**Experimental Designs Or Analyses:**

Lack of comparisons with state-of-the-art methods. All experiments present only test statistics and thresholds; explicitly reporting key evaluation metrics such as test power and Type-I error would be highly appreciated.

**Methods And Evaluation Criteria:**

See Claims And Evidence part.

**Other Comments Or Suggestions:**

N/A

**Other Strengths And Weaknesses:**

N/A

**Questions For Authors:**

N/A

**Relation To Broader Scientific Literature:**

N/A

**Theoretical Claims:**

The paper establishes a strong theoretical framework with rigorous proofs.

---

> ### Author Rebuttal · Authors · 2025-04-01
>
> Thank you for your insightful feedback and for dedicating time to review our paper! We respond to the specific questions as follows:
>
> **Intuitive explanation for definition of test statistic**: In addition to our existing discussion after Eq 15 in the paper, we provide here with an additional intuitive explanation: Consider the statistic
>
> $$T^{\prime} =  \sum_{(i,j)\in \mathcal{E}}\frac{Z_{ij}(Z_{ij}-1)}{k_{ij}'(k_{ij}'-1)}+F(w_i^*-w_j^*)^{2}-2F(w_i^* -w_j^*)\frac{Z_{ij}}{k_{ij}' }$$
> obtained by substituting $w^*$ in place $\hat{w}$ in Eq. 15. Then, the expected value of $T'$ is $||P- F(w^*) ||_F^2$.
>
> This is because the expected value of the first term is $p_{ij}^2$, and the last term is $-2 F(w_i^* - w_j^*) p_{ij}$. Hence, $T$ is constructed by plugging in $\hat{w}$ in place of $w^*$ in the unbiased estimator $T^{\prime}$ of $||P- F(w^*)||_F^2$.
> We will add additional clarification in the manuscript.
>
> **Relationship between this test statistic and those in [Rastogi et al. 2022] and [Makur, Singh 2023]**:
> * The test statistic proposed in [Rastogi et al. 2022] is for a two-sample testing problem (i.e., testing whether two sets of samples are drawn from the same pairwise comparison model or not) which is a very different problem to that in our work.
> * The test statistic in [Makur, Singh 2023] is for testing the BTL model, which is a special case of our model. Furthermore, their test statistic is based on spectral techniques, while ours is based on maximum likelihood techniques and is applicable for general Thurstone models
> * There is some resemblance between the three test statistics since all three statistics are ``estimators” for squared Frobenius distances of pairwise comparison models. However, the techniques used to analyze them are very different. For example, our theoretical analysis crucially relies on sample splitting, while the other two do not. We will expand on this discussion in the final version. We comment on the experimental concerns below.
>
> **Comparisons with state-of-the-art methods/Additional Experiments**: Except [Makur, Singh 2023], we are not aware of any work that addresses the same minimax problem as ours for a general Thurstonian model in a non-asymptotic regime. The tests based on likelihood ratio do not consider the general hypothesis testing problem considered in this work. However, we have performed following additional experiments for the BTL model. The figures for these experiments can be found at http://anonymous.4open.science/r/something-E6CB/ALLFigures.pdf
>
> 1. **Performance Comparison with [Makur, Singh 2023] (Exp1)**: We assume the observation graph to be complete and define the pairwise comparison matrix $P$ under the null and alternative hypothesis as the one used to prove the lower bound in Eq. 39. We set $\eta = 0.06$ and $k = 10$. We evaluate the empirical Type I and Type II errors for three methods: the proposed test statistic based on maximum likelihood estimation (Max. Likelihood), the same test statistic without sample splitting (Max. Likelihood2), and the spectral method from [Makur, Singh 2023] (Spectral). Our results in Fig. 1 indicate that the proposed method performs comparably to the spectral approach, while Max. Likelihood2 achieves even lower error. The threshold for all three methods is determined using the empirical quantile approach.
>
> 2. **Type I and II Errors: Empirical Quantile vs. Optimal Threshold (Exp2)**: We extend our previous experiment by comparing the empirical quantile threshold with the "optimal" threshold $\eta^2n(n-1)/2$. We evaluate Type I and II errors for $\eta\in[0.08,0.16], n\in[15,25,35,45], k=10$. Fig. 2 shows that the empirical quantile approach performs similarly to the optimal threshold, with Type I error control close to the nominal 0.05 level, despite the threshold being computed from randomly sampled skill scores.
> To address Reviewer 3Nsw concerns regarding threshold computation using clustering, we also compare thresholds derived from random and clustered skill scores, where clustering is based on assigning half the player's scores randomly in [-0.7,-0.4] and the rest with their exact negative. Fig 3 shows that both approaches yield nearly identical thresholds.
>
> 3. **Experiment on real-world datasets**: In addition to our LMSYS dataset experiment, we also analyzed the NBA dataset from https://www.kaggle.com/datasets/nathanlauga/nba-games/data. Using data from 2022 onward, we applied our test to the 12 teams with the most matches since 2002. Each comparison involved a home and away team, and we tested on cumulative data over t recent years. Fig. 4 shows that for the first ~10 years, the BTL and Thurstone models fit well, but for larger intervals, the hypothesis is rejected, as a single BTL score cannot capture team strength over extended periods.
>
> Various parameters, such as number of simulated datasets, are the same as those used in our paper for previous experiments. We will include these experiments in the final version

---

> > ### Comment · Reviewer_smfn · 2025-04-05
> >
> > Thank you for your response. The answers have addressed my questions and concerns. I will proceed to increase my rate.

---

### Official Review · Reviewer_MqJu · 2025-03-12

**Overall Recommendation:** 3

**Summary:**

The paper addresses the problem of hypothesis testing for whether a given pairwise comparison dataset follows a Generalized Thurstone Model (GTM), which is formally stated in equation (12). It proposes a test statistic along with a corresponding testing threshold that matches the lower bound on the critical threshold $\epsilon_c$ in the case of a complete observation graph, as defined in equation (14), thereby establishing its minimax optimality in this setting.

Additionally, the paper derives information-theoretic lower bounds on $\epsilon_c$ for different graph types, as shown in Table 1 and stated in Proposition 3.5. It also provides time-uniform bounds on Type I and Type II errors in a sequential testing framework, where at each time step, a single comparison is observed, as presented in Theorem 3.7. Furthermore, the paper constructs confidence intervals for the test statistic under the null hypothesis and validates the theoretical results through experiments on both synthetic and real datasets.

**Claims And Evidence:**

yes

**Essential References Not Discussed:**

no

**Experimental Designs Or Analyses:**

yes, see the comment above in the methods evaluation and criteria

**Methods And Evaluation Criteria:**

Yes, however, I think the experiment section could be improved by adding comparisons to other methods that perform hypothesis testing for Thurstone models, particularly in terms of Type I and Type II errors. For instance, some of the methods mentioned in the last paragraph of Section 1.2 could be included as baselines, even if they only apply to specific models, to provide a more informative comparison.

**Other Comments Or Suggestions:**

For completeness, it would be helpful to provide the expression for $F$ that was used in the experiments for the BTL and Thurstone models.

**Other Strengths And Weaknesses:**

Strengths:

- The paper addresses an interesting problem within a general framework that encompasses many Generalized Thurstone Models.
- The paper’s theoretical results are promising and well-developed, providing a thorough investigation into the minimax optimality of the proposed test. Additionally, it offers bounds within the sequential testing framework.


Weaknesses:

- I believe the experimental study could be enhanced by providing more informative results and comparing the proposed test with other methods.
- The approach used in Section 4 to estimate the threshold—by generating numerous simulated datasets under the null hypothesis to compute the testing threshold—can be computationally expensive.

**Questions For Authors:**

Is there a specific motivation for using the weighted log-likelihood in equation (6) instead of the typical log-likelihood, which I assume would involve $Z_{ij}$ instead of $p_{ij}$?

**Relation To Broader Scientific Literature:**

The key contribution is the proposed hypothesis test for the Generalized Thurstone Model using a maximum likelihood approach. This complements previous work in (1), which developed hypothesis tests for the Bradley-Terry-Luce (BTL) model based on spectral methods.


---

(1) Makur, A., and Singh, J. (2023). "Testing for the Bradley-Terry-Luce Model." In IEEE International Symposium on Information Theory (ISIT).

**Theoretical Claims:**

I checked the general ideas in the proofs but I haven't verified each step.

---

> ### Author Rebuttal · Authors · 2025-03-31
>
> Thank you for your insightful feedback and for dedicating time to review our paper! We respond to the specific questions as follows:
>
> **Comparisons to other methods for testing of Thurstone models**: Minimax testing for generalized Thurstone models (for a fixed choice function $F$) has not been studied much in the literature. The case of logistic $F$, which corresponds to BTL models, is the only case that has baselines [Makur, Singh 2023]. So, it is difficult to provide accurate baselines for general functions $F$. However, to address this concern, we have included an additional experiment to compare the proposed method (Max. Likelihood) with [Makur, Singh 2023] for complete graphs (See response to Reviewer smfn for details). Our results show that the proposed method has a similar performance to the spectral method in [Makur, Singh 2023]. We have also added other experiments as outlined in the response to Reviewer smfn. We will add details of all experiments to the manuscript to make it more informative.
>
>
> **Estimating threshold can be computationally expensive**: Yes, we agree that the computational complexity of estimating threshold will depend on the number of simulated datasets generated. However, we remark that:
> 1) This procedure is efficiently parallelizable.
> 2) Even with a naive implementation (without any parallelization), the running time is small as each of our experiments can be done within 5 minutes on a normal CPU for $n$ as large as $60$. To speedup computing $\hat{w}$ on the simulated dataset, one can initialize the iterates at the optimal values that generated the simulated dataset, followed by a few iterations of gradient descent.
> 3) Moreover, when all $k_{ij}$ are equal (or roughly the same), our simulation results in Figure 1 suggest that $0.75n/k$ as the threshold for complete graphs and $0.4n/k$ for 2D grid is a good approximation.
> 4) Using an asymptotic approximation and an intermediate result (Proposition 3.8), we can approximate the threshold as detailed in Appendix F.2. Approximating the threshold in this way is much faster than the empirical quantile approach.
>
>
> **Motivation for using the weighted log-likelihood**: We chose to use weighted negative log-likelihood for the following reasons:
> 1) To define the minimax composite hypothesis testing problem, we first introduced a separation distance. For analytical purposes, we chose to represent this separation distance using the (unweighted) Frobenius norm. The weighted likelihood expression in Eq. 8 arises when we relate the separation distance in Frobenius norm to the cross-entropy term, as detailed in the Proof of Theorem 1. As a result, our definitions in Eq. 6,7 naturally inherited the weighted property.
> 2) If we were to use the unweighted version of maximum likelihood, we would need to employ a weighted Frobenius norm with weights $k_{ij}$ to define separation distance, which did not seem a natural choice to us. Alternatively, we would need to add an extra assumption that different $k_{ij}$ are within constant factors of each other, which we also chose not to do.
>
>
>
> **Expression for F for the BTL and Thurstone models**: We remark that we have used $F(t) = \frac{1}{1 + e^{-t} }$ (sigmoid function) for the BTL model and $F(t) = \int_{-\infty}^t  \frac{1}{\sqrt{2\pi}} e^{-x^2/2} dx$ (complementary-CDF of normal) for the standard Thurstone model. We will include this information in the revised manuscript.

---

> > ### Comment · Reviewer_MqJu · 2025-04-07
> >
> > Thank you for your detailed response, and for running the additional experiments.

---

### Decision · Program_Chairs · 2025-05-01

**Decision:**

Accept (poster)

**Comment:**

**Summary.**

The topic of this paper is goodness-of-fit testing for pairwise comparison models.
Specifically, the objective is to test whether pairwise comparison data was generated by a Generalized Thurstone Model (null hypothesis; henceforth “GTM”) or by a pairwise comparison model which is not GTM (composite alternative hypothesis).

In a GTM, it is assumed that each agent $i$ is assigned a latent utility value $w_i$ and the probability $p_{ij}$ that agent $i$ compares favorably with $j$ (or is "preferred") is given by $F(w_j-w_i)$ where $F$ (the so-called “choice function”) is a CDF, whereas in a more general pairwise comparison model, the $p_{ij}$ probabilities may be unstructured and are recorded in a matrix.

Under a selected notion of distance to the sub-class of GTMs and chosen minimax formulation (where the risk is the sum of the type I and type II errors), the authors derive lower bounds on the so-called critical threshold of the problem (14) which depend on the topology of the underlying observation graph. Conversely, they propose and analyze a test statistic to obtain upper bounds on this critical threshold. In particular, their results are tight when the underlying graph is complete. They further derive time-uniform bounds on the error probabilities in a sequential testing framework and provide an experimental evaluation of their test.

**Strengths.**

* The problem had not been considered before in its generality (only the more restricted BTL model was previously considered in this framework).
* The problem is technically difficult and leaves questions for further investigation to the community.
* They perform comparably to Makur & Singh, 2023 for the more restricted BTL models.
* Sufficient technical novelty (MqJu, 3Nsw) compared to Makur & Singh, 2023.

**Weaknesses.**

* Not clear how much interest there is for non-BTL models in practice (3Nsw).
* The results are not tight for most observation graphs.

**Discussion and reviewer consensus.**

The author response successfully addressed the reviewers’ concerns.
However, while the referees acknowledged its contributions, none seemed to be particularly excited about the paper.

**Additional remarks.**

Originally, the referees had numerous concerns:

* Experiments are insufficient: no baselines, missing relevant evaluation metrics (MqJu, smfn, 3Nsw).
* Computing the testing threshold is expensive (MqJu).
* There is no intuition for the propose test statistics (smfn).

However, the authors have done an honest effort in their response, and ran additional experiments. As a result, the above concerns were dispelled.

**Overall evaluation.**

I trust that the authors would be able to make the promised changes in their final version. Due to its theoretical nature, I do not think that remaining experimental limitations of the paper warrant rejection; quite on the opposite, I think the authors went the extra mile. I am recommending this paper for acceptance.